# Strong Correlations Induce Cause Only Predictions in Transformer Training

**Haihan Zhang[1], Yimu Zhang[1], Cong Fang[1,2],** *
[1] State Key Lab of General AI, School of Intelligence Science and Technology, Peking University
[2] Institute for Artificial Intelligence, Peking University
{zhanghaihan}@stu.pku.edu.cn

## Abstract

We revisit when Transformers can prioritize causes over spurious effects by viewing the problem through data correlation strength and the implicit regularization of gradient descent. We identify a phenomenon called Correlation Crowding-Out (CCO) arising from the training dynamics of Transformers. Specifically, under strongly correlated causal features, gradient descent filters out spurious cues and converges to a predictor that relies almost exclusively on the causes. Theoretically, using a simplified Transformer model trained on data from a minimal causal chain, we introduce a Dominant-coordinate condition that characterizes when CCO arises and explain its mechanism as a coupling of "occupation" and "crowding-out". "Occupation" denotes the rapid growth of weights aligned with the dominant causal direction while non-dominant directions remain small. "Crowding-out" denotes the attention logits align with separation directions favoring the causal branch, suppressing descendants. We provide convergence guarantees for both the optimization trajectory and generalization. Our empirical results on simulated and real examples across various tasks including vision and natural language demonstrate the procedure. Together, these results show that, under suitable conditions, standard training alone can induce cause only prediction.

## 1 Introduction

Whether data-driven models can extract causal invariances from observational data and thereby deliver robust predictions has long been a central hope in AI (Pearl, 2009; Peters et al., 2016; Arjovsky et al., 2019; Schölkopf et al., 2021; Fan et al., 2024). Yet models trained by empirical risk minimization are often prone to shortcut learning (Geirhos et al., 2020; Shah et al., 2020; Ye et al., 2024), indiscriminately exploiting any correlation, including spurious cues unrelated to the true causal mechanisms (Sagawa et al., 2020; Qiu et al., 2023). This pattern is widely documented across modalities and tasks (Geirhos et al., 2019; McCoy et al., 2019; Li et al., 2025). The rise of Transformers and LLMs sharpens this tension: these systems can sometimes rely on shallow artifacts (Bender et al., 2021; Tang et al., 2023; Du et al., 2023; Varma et al., 2024; Jin et al., 2024; Gui & Ji, 2025), yet they also produce answers that appear strikingly logical and robust in certain scenarios (Brown et al., 2020; Wei et al., 2022; Kojima et al., 2022; Yuan et al., 2024). Recent theory offers partial clues for why Transformers sometimes appear causal, but does not yet answer the cause only generalization question. On stylized in-context tasks, Transformers trained on Markov Chain sequences can recover parent sets and estimate transition probabilities in-context (Edelman et al., 2024; Nichani et al., 2024; D'Angelo et al., 2025). These results suggest how attention might reconstruct graph edges from observational sequences, but they rely on designed ICL setups rather than generic pipelines with spurious features and do not show when spurious information is suppressed at both train and test time. In parallel, large margin analyses show that gradient descent (GD) pushes query–key parameters toward max-margin separators (Tarzanagh et al., 2023; Ataee Tarzanagh et al., 2023; Vasudeva et al., 2025). While this suggests separation can emerge during training, it does not characterize how such separation filters out spurious features or yields cause only risk guarantees. This landscape motivates a basic question:

---

* Corresponding author

*When and through what mechanism can Transformer training produce predictors*
*that rely on causes while ignoring spurious effects?*

We answer this by uncovering and analyzing Correlation Crowding-Out (CCO). CCO is a training phenomenon in which, under a uniform dominance gap where a causal feature is more strongly associated with the target than any competing spurious feature, GD drives Transformer to progressively suppress spurious features and converge to a predictor that relies almost exclusively on the causal feature. Crucially, the dominance condition does not require spurious features to be weak: many can remain highly correlated with the target and may even surpass non-dominant causal coordinates. What matters is a persistent margin favoring the dominant causal direction.

Remarkably, strong causal correlation in the data *alone* does not guarantee cause only prediction for generic estimators. In Example 25, even under a dominance gap, population least squares retains a constant fraction of a spurious feature. Thus, CCO is not a corollary of data dominance; it hinges on optimization induced implicit regularization that actively crowds out spurious features. This occurs without explicit invariance penalties (Arjovsky et al., 2019; Shapiro, 2017; Fan et al., 2024) or multi-environment training (Peters et al., 2016; Fan et al., 2024; Xu et al., 2024).

Our perspective complements existing analyses of correlation driven learning dynamics. Prior work has shown that neural networks exhibit a simplicity bias, often preferring features that are highly correlated with the label or easier to fit (Belkin et al., 2019; Moayeri et al., 2022; Morwani et al., 2023; Qiu et al., 2023; Xue et al., 2023; Yang et al., 2024). When spurious features are more predictive or less complex, they tend to dominate early training, delaying or even entirely inhibiting the learning of causal features (Shah et al., 2020; Yang et al., 2024). These studies underscore that correlation strength and feature complexity critically shape learning trajectories, and they reinforce the notion that deep models are vulnerable to superficial shortcuts. In contrast, we focus on the opposite regime: when the causal features themselves dominate in predictiveness. We formalize CCO as the mirror image of shortcut learning. Intuitively, if a causal feature explains the target with overwhelming strength, the model has little incentive to rely on weaker spurious cues.

Building on this premise, we demonstrate CCO empirically and provide a theoretical account of its mechanism. To theoretically understand this behavior, we analyze a simplified two-layer Transformer trained on data from a causal chain ($\mathbf{x} \to y \to \mathbf{z}$) generative process. Our theory provides a Dominant-Coordinate Condition on the data, which quantifies how strong the $\mathbf{x}$-$y$ correlation must be for CCO to occur. Under this condition, the training dynamics unfold in two coupled phases. In the first "occupation" phase, within the Transformer's feed-forward sublayer, the weight vector that aligns with the dominant causal coordinate in $\mathbf{x}$ grows rapidly to a stable magnitude, while weights in other directions remain small. This expansion makes the causal direction salient and establishes it as the primary signal driving the predictions. Next comes the "crowding-out" phase: the Transformer's attention mechanism gradually shifts its query-key alignment toward the max-margin separator between the transformed causal and spurious features (roughly, $\tilde{\mathbf{x}} - \tilde{\mathbf{z}}$). Consequently, the attention weights concentrate almost entirely on the causal $\mathbf{x}$ branch, effectively gating out the spurious $\mathbf{z}$ branch. Through this two-phase process, GD steers the model toward a cause only solution without any specialized regularization for invariance.

By elucidating the mechanism behind CCO, we contributes a more nuanced perspective on Transformer's generalization: while spurious shortcuts are a serious and pervasive concern, there exist regimes in which strong causal signals can turn GD into an ally for causal learning. In such regimes, the implicit regularization of GD yield cause only generalization, even in the absence of multiple training environments or explicit causal objectives.

## 1.1 OUR CONTRIBUTION

- We introduce and formalize the new phenomenon CCO.
- We elucidate CCO's mechanism with both theory and experiments.

## 2 RELATED WORK

**Spurious Correlations and Invariance Learning.** Across vision, language, and ERM-trained deep models including modern Transformers and LLMs—readily latch onto shortcut cues and spu-

rious correlations, leading to brittle generalization under shift (Geirhos et al., 2019; 2020; Zhou et al., 2021; Du et al., 2021; Tang et al., 2023; Du et al., 2023; Yuan et al., 2024). A major theoretical response is invariance learning: instead of trusting raw correlations, one seeks mechanisms stable across environments. Two canonical frameworks are Invariant Causal Prediction (Peters et al., 2016; Meinshausen et al., 2016), which tests for subsets of covariates that render the conditional law of the target invariant across interventions or environments, and Invariant Risk Minimization (Arjovsky et al., 2019), which encourages representations that admit a single optimal classifier across environments. Both lines have spurred extensive follow-ups and critiques clarifying assumptions, identifiability, and practical limitations (Ghassami et al., 2017; Heinze-Deml et al., 2018; Pfister et al., 2019; Rothenhäusler et al., 2019; 2021; Rosenfeld et al., 2021; Lin et al., 2022b; Kamath et al., 2021; Lu et al., 2021a; Zhou et al., 2022; Lin et al., 2022a). In parallel, distributionally robust optimization offers a complementary lens by minimizing worst-case (group) risk under distributional shifts (Shapiro, 2017; Sagawa et al., 2020; Duchi & Namkoong, 2021; Gao et al., 2024). More recently, Environment-Invariant Linear Least Squares and variants show that, when cross-environment heterogeneity is sufficiently strong, a regularized least-squares estimator can recover invariant features with generalization guarantees while quantifying heterogeneity (Fan et al., 2024; Gu et al., 2025a; Xu et al., 2024; Gu et al., 2025b). Most invariance methods posit either explicit regularizer or environment partitions; comparatively less is known about when standard GD on a Transformer will, by its own dynamics, yield an cause only predictor. Our work targets precisely this gap.

**Implicit Bias.** Implicit bias refers to the tendency of (S) GD, even without explicit regularization, to select solutions with special structure and generalization properties, widely regarded as a key to the success of over-parameterized models. Such as for logistic regression, (S) GD converges in direction to the max-margin classifier (Soudry et al., 2018; Ji & Telgarsky, 2019; Wu et al., 2025; Cai et al., 2025); in over-parameterized linear models, (S) GD can display benign overfitting (Zou et al., 2021; Wu et al., 2022), double descent (Lu et al., 2023; Zhang et al., 2025), and scaling laws (Bordelon et al., 2024; Lin et al., 2024); and for quadratically parameterized models, (S) GD implicitly favors low-complexity solutions and exhibits incremental learning (Li et al., 2018; Vaskevicius et al., 2019; Woodworth et al., 2020; HaoChen et al., 2021; Li et al., 2021; Jin et al., 2023; Xu et al., 2024). Turning to Transformers, a growing theory literature dissects how attention evolves under GD. For single-head ViTs, GD is shown to concentrate attention on label-relevant tokens, yielding progressively sparse maps (Jelassi et al., 2022; Li et al., 2023a). These results clarify which inputs receive mass under training induced anisotropy, but they are agnostic to causal structure. On stylized in-context tasks, Transformers trained on Markov chain sequences learn the set of parent tokens and estimate transition probabilities in-context; related mechanistic work on induction heads explains how attention circuits implement dependency tracking and copying behaviors (Lu et al., 2021b; Olsson et al., 2022; Li et al., 2023b; Edelman et al., 2024; Nichani et al., 2024; D'Angelo et al., 2025). These analyses are posed in designed ICL setups and do not address under generic training with spurious descendants when and why GD yields a cause only predictor. A complementary line shows that GD on attention pushes query–key parameters toward max-margin separators, establishing that separation can emerge during training; yet this does not identify which side of the margin corresponds to causal versus spurious directions, nor when separation suffices for cause only generalization (Tarzanagh et al., 2023; Ataee Tarzanagh et al., 2023; Vasudeva et al., 2025).

## 3 CCO: A PHENOMENOLOGY IN TRANSFORMER TRAINING

CCO refers to a training phenomenon in Transformers whereby, if there exists a dominant causal feature whose association with the target exceeds that of any competing spurious feature by a uniform gap, GD learns a predictor that progressively suppresses spurious features and relies almost exclusively on the causal one. Crucially, this dominance condition does not require spurious features to be weak: many can remain highly correlated with the target and some may even surpass non-dominant causal features. What matters is a persistent gap favoring the dominant causal direction.

CCO unfolds through two coupled effects:

(I) Occupation (early rise): within representation and prediction layers (e.g., embeddings, feedforward blocks, attention heads), weights aligned with a dominant, highly predictive causal feature grow rapidly to a stable, large scale, while spurious features aligned directions remain small, rendering the causal signal salient to the optimizer.

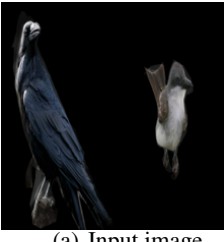 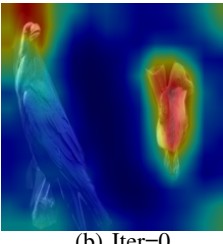 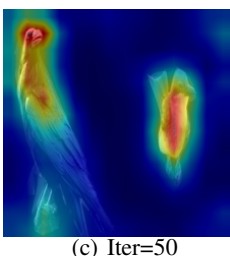 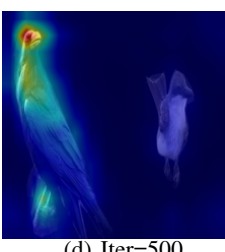

|           |           |            |             |
|-----------|-----------|------------|-------------|
| (a) Input image | (b) Iter=0 | (c) Iter=50 | (d) Iter=500 |

Figure 1: This figure shows how attention shifts during ViT training on a foreground–foreground causal disentanglement task. (a) is the input image. Early in training (b, iter 0), attention is diffuse across left bird (causal feature) and right bird (spurious feature). As training proceeds (c, iter 50), attention weight rises on the left bird illustrating the occupation phase. By (d, iter 500), attention is concentrated almost entirely on the left bird, with the right bird and background receiving near zero weight, illustrating the crowding-out phase.

(II) Crowding-out (attention selection): multi-head attention progressively aligns its logits with separation directions that prefer causal over spurious features (e.g., larger query-key margins for causal tokens), concentrates attention mass on causal features, and suppresses spurious features.

We verify the above phenomenon in Fig 1. During ViT training, the attention map first show a rapid growth in left bird (causal feature) in the occupation stage, while the attention on right bird (spurious feature) remain small growth. Then in the crowding-out stage, the attention allocated to the left bird significantly surpasses the attention to the right bird, with the right bird's attention appearing very faint in the attention map.

## 3.1 WHY CCO ARISES

(I) Intrinsic strong causal correlation: CCO emerges when the data exhibits a property whereby the correlation between a causal feature and the target is consistently stronger than that of any spurious feature. This is common rather than contrived: many real datasets can be viewed as mixtures of latent environments in which causal–target relationships remain relatively stable, whereas spurious features oscillate across environments. When pooled, these oscillations destructively interfere, reducing spurious–target correlation relative to causal–target correlation. Equivalently, causal features concentrate stable signal, while spurious features disperse unstable variance, making the dominant causal direction statistically more salient.

(II) Implicit regularization of GD in Transformers: early strong-signal directions (Occupation) steer gradients toward causal features, inducing a directional bias in the learned representation. Attention then transduces this bias into selection (Crowding-out), assigning higher weight to causal features and down-weighting spurious ones, thereby approaching an invariant, cause only solution without explicit invariance penalties.

## 3.2 STRONG CAUSAL CORRELATION ALONE DOESN'T ENSURE CAUSE ONLY

It is important to stress that strong causal correlation in the data does *not* by itself *alone* guarantee cause only prediction for generic estimators. In Example 25, we show that even under a dominant causal correlation, population linear regression retains a constant fraction of the spurious features, remaining using spurious features to predict noise. This demonstrates two points: (a) strong causal alignment alone does not ensure spurious suppression; and (b) CCO is *not* a trivial corollary of data dominance but instead relies on the *implicit regularization* induced by Transformers and GD.

## 4 THEORETICALLY ANALYSIS OF CORRELATION CROWDING-OUT

### 4.1 PROBLEM SETUP

We provide a theoretical explanation for CCO by analyzing a specialized Transformer module trained on data generated by the causal chain $\mathbf{x} \to y \to \mathbf{z}$. When the dominant feature of $\mathbf{x}$ exhibits

sufficiently strong association with $y$, the implicit regularization of GD leads the learned predictor to filter out $\mathbf{z}$ and rely almost exclusively on $\mathbf{x}$.

### 4.1.1 DATA GENERATIVE PROCESS

We consider the causal chain $\mathbf{x} \to y \to \mathbf{z}$, where $\mathbf{x}, \mathbf{z} \in \mathbb{R}^d$ are vector covariates and $y \in \mathbb{R}$ is a scalar response. The response $y$ is a sparse quadratic signal in $\mathbf{x}$:

$$y = \mathbf{x}^\top (\mathbf{w}^*)^{\odot 2} + \epsilon,$$

with noise $\epsilon \perp \mathbf{x}$, $\mathbb{E}[\epsilon] = 0$, and $\mathrm{Var}(\epsilon) = \sigma^2$. The descendant $\mathbf{z}$ depends on $y$ via an $L$-Lipschitz function $f : \mathbb{R} \to \mathbb{R}^d$ and additive noise $\boldsymbol{\xi} \in \mathbb{R}^d$,

$$\mathbf{z} = f(y) + \boldsymbol{\xi}, \quad \boldsymbol{\xi} \perp y.$$

We assume the moment and boundedness conditions:

$$\mathbf{H} := \mathbb{E}[\mathbf{x}\mathbf{x}^\top] = \begin{bmatrix} a & \mathbf{0} \\ \mathbf{0} & \mathbf{I}_{d-1} \end{bmatrix}, \quad \mathbb{E}[\mathbf{x} + \mathbf{z}] = \boldsymbol{\zeta}, \quad \mathrm{Var}(\mathbf{x} + \mathbf{z}) = \Sigma,$$

and almost surely $\sup_{1 \le j \le d} |\mathbf{x}_j| \le B_{\mathbf{x}}$, $\sup_{2 \le j \le d} |\mathbf{x}_j| \le B'_{\mathbf{x}}$, $|\epsilon| \le B_\epsilon$, $\sup_{1 \le j \le d} |\boldsymbol{\xi}_j| \le B_{\boldsymbol{\xi}}$, $\sup_{1 \le j \le d} |\mathbf{z}^i_j| \le \|f(0)\|_\infty + L(rB_{\mathbf{x}} + B_\epsilon) + B_{\boldsymbol{\xi}} := B_{\mathbf{z}}$. The ground truth $\mathbf{w}^*$ is sparse and binary: $\mathbf{w}^*_j \in \{0, 1\}$, $\mathbf{w}^*_1 = 1$, and $|\mathrm{supp}(\mathbf{w}^*)| \le r$. We observe i.i.d. samples $\{(\mathbf{x}^i, y^i, \mathbf{z}^i)\}_{i=1}^n$ from $(\mathbf{x}, y, \mathbf{z})$.

The chain $\mathbf{x} \to y \to \mathbf{z}$ is a minimal DAG that that captures the key trade-off behind CCO: a causal parent $\mathbf{x}$ that determines $y$, versus a spurious descendant $\mathbf{z}$ is induced by $y$. This reduction is purposeful and representative. For example, in sentiment analysis, content features $\mathbf{x} \to$ sentiment label or rating $y \to$ label derived auxiliary fields generated downstream $\mathbf{z}$ (Gururangan et al., 2018). So that $\mathbf{z}$ is a descendant induced spurious correlate of $y$ while $\mathbf{x}$ carries the causal signal.

In this pattern, descendants furnish alluring but non invariant shortcuts, a phenomenon widely documented across deep learning (Geirhos et al., 2020). By positing one dominant, highly $y$-predictive direction in $\mathbf{x}$ while allowing $\mathbf{z}$ to be strongly, yet non causally correlated with $y$. Thus, the $\mathbf{x} \to y \to \mathbf{z}$ pattern offers a principled, portable abstraction: it is simple enough for precise analysis yet representative of broader scenarios where CCO is expected to emerge.

### 4.1.2 MODEL ARCHITECTURE

We adopt a two-key attention architecture and augment inputs with fixed positional encodings $\mathbf{s}_1, \mathbf{s}_2 \in \mathbb{R}^M$:

$$\tilde{\mathbf{x}}^i = \begin{bmatrix} \mathbf{s}_1 \\ \mathbf{x}^i \end{bmatrix}, \qquad \tilde{\mathbf{z}}^i = \begin{bmatrix} \mathbf{s}_2 \\ \mathbf{z}^i \end{bmatrix} \in \mathbb{R}^{M+d}.$$

We parameterize the *query* as the gating vector $\mathbf{q}^t := \tilde{\mathbf{v}}^t \in \mathbb{R}^{M+d}$, take the *keys* as $\mathbf{k}^i_x := \tilde{\mathbf{x}}^i$ and $\mathbf{k}^i_z := \tilde{\mathbf{z}}^i$, and the *values* as $\mathbf{v}^i_x := \tilde{\mathbf{x}}^i$ and $\mathbf{v}^i_z := \tilde{\mathbf{z}}^i$.

**Two-key Attention.** Define the logits

$$\ell^t_{x,i} = (\mathbf{q}^t)^\top \mathbf{k}^i_x, \qquad \ell^t_{z,i} = (\mathbf{q}^t)^\top \mathbf{k}^i_z,$$

and weights

$$\alpha^t_{x,i} = \frac{e^{\ell^t_{x,i}}}{e^{\ell^t_{x,i}} + e^{\ell^t_{z,i}}}, \qquad \alpha^t_{z,i} = 1 - \alpha^t_{x,i}.$$

By softmax translation invariance,

$$\alpha^t_{x,i} = \sigma\big((\mathbf{q}^t)^\top (\mathbf{k}^i_x - \mathbf{k}^i_z)\big) = \sigma\big((\tilde{\mathbf{v}}^t)^\top (\tilde{\mathbf{x}}^i - \tilde{\mathbf{z}}^i)\big) =: p^t_i.$$

The attention output (per sample) is

$$\hat{\mathbf{h}}^{i,t} = \alpha^t_{x,i} \mathbf{v}^i_x + \alpha^t_{z,i} \mathbf{v}^i_z = p^t_i \tilde{\mathbf{x}}^i + (1 - p^t_i) \tilde{\mathbf{z}}^i.$$

---

**Algorithm 1** GD on the two-key attention model

---

1: **Input:** $\{(\mathbf{x}^i, y^i, \mathbf{z}^i)\}_{i=1}^n$, encodings $\mathbf{s}_1, \mathbf{s}_2$, stepsizes $\{\eta_t\}, \{\beta_t\}$, initialization scale $\alpha$, iterations $T$.

2: **Positional Encoding:** $\tilde{\mathbf{x}}^i = \begin{bmatrix} \mathbf{s}_1 \\ \mathbf{x}^i \end{bmatrix}$, $\tilde{\mathbf{z}}^i = \begin{bmatrix} \mathbf{s}_2 \\ \mathbf{z}^i \end{bmatrix}$.

3: **Init:** $\tilde{\mathbf{w}}^0 = \begin{bmatrix} \mathbf{0} \\ \alpha \mathbf{I}_d \end{bmatrix}$, $\tilde{\mathbf{v}}^0 = \mathbf{0}_{M+d}$.

4: **for** $t = 0, 1, \ldots, T-1$ **do**

5:     **for** $i = 1$ **to** $n$ **do**

6:         $p_i^t \leftarrow \sigma\big((\tilde{\mathbf{v}}^t)^\top (\tilde{\mathbf{x}}^i - \tilde{\mathbf{z}}^i)\big)$, $\hat{y}^{i,t} \leftarrow \big(p_i^t \tilde{\mathbf{x}}^i + (1-p_i^t)\tilde{\mathbf{z}}^i\big)^\top (\tilde{\mathbf{w}}^t)^{\odot 2}$, $r_i^t \leftarrow \hat{y}^{i,t} - y^i$

7:     $\tilde{\mathbf{w}}^{t+1} \leftarrow \tilde{\mathbf{w}}^t - \frac{\eta_t}{n} \sum_{i=1}^n r_i^t \big(p_i^t \tilde{\mathbf{x}}^i + (1-p_i^t)\tilde{\mathbf{z}}^i\big) \odot \tilde{\mathbf{w}}^t$

8:     $\tilde{\mathbf{v}}^{t+1} \leftarrow \tilde{\mathbf{v}}^t - \frac{\beta_t}{n} \sum_{i=1}^n r_i^t p_i^t (1-p_i^t)(\tilde{\mathbf{x}}^i - \tilde{\mathbf{z}}^i)^\top \big((\tilde{\mathbf{w}}^t)^{\odot 2}\big)(\tilde{\mathbf{x}}^i - \tilde{\mathbf{z}}^i)$

9: **Return:** $(\tilde{\mathbf{w}}^{t+1}, \tilde{\mathbf{v}}^{t+1})$.

---

**Squared-parameter Head and Loss.** We predict with a quadratic parameterization feed-forward layer:

$$\hat{y}^{i,t} = \big(\hat{\mathbf{h}}^{i,t}\big)^\top (\tilde{\mathbf{w}}^t)^{\odot 2} = \sum_{j=1}^{M+d} \big(\tilde{\mathbf{w}}_j^t\big)^2 \hat{\mathbf{h}}_j^{i,t}, \qquad \mathcal{L}_n(\tilde{\mathbf{w}}, \tilde{\mathbf{v}}) = \frac{1}{2n} \sum_{i=1}^n (\hat{y}^i - y^i)^2.$$

This quadratic parameterization feed-forward layer can be seen as a special diagonal neural network, essentially a position wise FFN that provides anisotropic multiplicative gains and thus retains feature learning capacity through the attention mixed representation. This parameterization can be further generalized by $\hat{y}^{i,t} = \big(\hat{\mathbf{h}}^{i,t}\big)^\top \big((\tilde{\mathbf{w}}^{+,t})^{\odot 2} - (\tilde{\mathbf{w}}^{-,t})^{\odot 2}\big)$.

GD on the two-key attention model is summarized in Algorithm 1.

Our module is exactly a single-head dot-product attention applied per sample with two keys/values, one for the cause path and one for the descendant path. It is the special case of a Transformer attention block where $W_Q, W_K, W_V$ are identity projections, so the query is the learned gating direction $\tilde{\mathbf{v}}$, and the two tokens are $\tilde{\mathbf{x}}$ and $\tilde{\mathbf{z}}$. This reduction keeps the softmax competition geometry and the value mixing mechanism intact while stripping away projection layers that would obscure the optimization dynamics. The quadratic parameterization head is a diagonal, position wise FFN that provides nonnegative per-coordinate gains. Studying this minimal attention–FFN pair is theoretically meaningful: it isolates the allocation dynamics behind the implicit bias we analyze, preserving the key nonlinearities (softmax and multiplicative gains) that produce CCO.

The distinct fixed encodings $\mathbf{s}_1 \neq \mathbf{s}_2$ attach branch identity to keys and values and inject a sample-independent margin $(\tilde{\mathbf{v}}^t)^\top(\mathbf{s}_1 - \mathbf{s}_2)$ into the logit difference. When $\mathbf{x}_i$ and $\mathbf{z}_i$ are weakly separated early in training, the offset $(\tilde{\mathbf{v}}^t)^\top(\mathbf{s}_1 - \mathbf{s}_2)$ prevents the gate from collapsing to $1/2$ and ensures a non-degenerate gradient, thereby guaranteeing identifiability of branches and stable training dynamics. This mirrors the role of positional embeddings in Transformers.

### 4.1.3 DOMINANT-COORDINATE CONDITION.

We characterize which patterns of strong correlation are sufficient for CCO to emerge. The two conditions below formalize (i) a population-level dominance of one causal coordinate and (ii) a per-sample margin along that coordinate.

Define $s_j := \mathbb{E}\big[(\mathbf{x}^\top (\mathbf{w}^*)^{\odot 2})(\mathbf{x}_j + \mathbf{z}_j)\big]$ measures the cross-moment between response $y$ and the combined coordinate $\mathbf{x}_j + \mathbf{z}_j$. The adjustment $\mu_j := \mathbb{E}[\epsilon(\mathbf{x}_j + \mathbf{z}_j)]$ accounts for noise leakage. $s_j^{\text{eff}} := s_j + \mu_j$ is the effective signal which governs the drift of gradient updates. We also write $m_j := \mathbb{E}\big[(\mathbf{x}_j + \mathbf{z}_j)^2\big] = \Sigma_{jj} + \zeta_j^2$ and $m_{kj} := \mathbb{E}\big[(\mathbf{x}_k + \mathbf{z}_k)(\mathbf{x}_j + \mathbf{z}_j)\big] = \Sigma_{kj} + \zeta_k\zeta_j$ which capture the second-moment scales of the combined features.

**Condition 1.** *The effective signal satisfies that $s_1^{\text{eff}} > \frac{2m_1}{15} + \max_{j>1}\big(4\big|s_j^{\text{eff}}\big| + \frac{m_{1j}}{8}\big)$.*

Condition 1 requires effective signal the dominant feature is sufficiently strong to exceed that of other competitor by a uniform gap. The assumption is mild, it allows strong descendant induced correlations on other coordinates but prevents the dominant causal direction from being overwhelmed. Under Condition 1, the GD dynamics preferentially amplify the squared weight on the dominant coordinate, creating the occupancy that initiates CCO.

**Condition 2.** *There exist constant $\tau_1, \tau_2 > 0$ such that for every sample $i = 1, \ldots, n$: (i) Nontrivial gap:* $|\mathbf{x}_1^i - \mathbf{z}_1^i| \geq \tau_1$. *(ii) Sign stability:* $\text{sgn}(\mathbf{x}_1^i - \mathbf{z}_1^i) = \text{sgn}(\mathbf{x}_1^i)$. *(iii) Dominant-coordinate margin lower bound:* $\frac{3}{4}|\mathbf{x}_1^i| \geq (r-1)B_{\mathbf{x}}' + B_\epsilon + \tau_2$.

In combination with Condition 1, Condition 2 guarantees that GD on the gate parameter $\tilde{\mathbf{v}}^t$ towards the max-margin solution on $\{\tilde{\mathbf{x}}_i - \tilde{\mathbf{z}}_i\}_{i=1}^n$ drives $p_i^t \to 1$ and thereby squeezes out the descendant branch. In short, Condition 1 ensures occupancy, whereas Condition 2 ensures crowding out, completing the CCO mechanism.

These two conditions are satisfiable in bounded, Lipschitz settings. Importantly, as detailed in Example 26, they do not exclude the empirically relevant regime where some non-dominant causal coordinates are less correlated with $y$ than descendant coordinates: it can happen that for some $j > 1$ with $\mathbf{w}_j^* = 1$, $\text{Cov}(\mathbf{x}_j, y) < \text{Cov}(\mathbf{z}_j, y)$.

## 4.2 Main Result

We next formalize when and how CCO emerges in our two-key attention model. Under the Dominant-coordinate condition, the first theorem provides a mechanistic account of CCO during training. The second theorem provides a generalization guarantee: with high probability, the learned predictor filters out the descendant $\mathbf{z}$, relies almost exclusively on the causal $\mathbf{x}$, and attains test risk near the cause only level.

**Theorem 1** (CCO's Mechanism). *Under Condition 1 and Condition 2, consider GD with initialization scale $\alpha = \frac{\sqrt{\sigma^2 \log d/n}}{d^3}$ and the following stepsize schedule: (i) For $1 \leq t \leq T_1^* := \min\{t \in \mathbb{N} : \mathbf{w}_1^t \geq \frac{1}{4}\}$, set $\eta_t \equiv \eta$ and $\beta_t \equiv 0$. (ii) For $T_1^* < t \leq T_1^* + T_2^*$, with $T_2^* \asymp \exp\left(\sqrt{\|\mathbf{s}\|_2^2 + d(B_{\mathbf{x}} + B_\xi)^2}\right)$, set $\eta_t \equiv 0$ and $\beta_t \equiv \beta$. (iii) For $T_1^* + T_2^* < t \leq T_1^* + T_2^* + T_3^* =: T^*$, set $\eta_t \equiv \eta$ and $\beta_t \equiv 0$ with $T_3^* \asymp \frac{1}{\eta} \log\left(\frac{n}{\sigma^2 \log(dr)}\right)$. Then, with probability at least $1 - \frac{1}{d^2}$, the squared-parameter head satisfies*

$$\left|\mathbf{w}_i^{T^*} - \mathbf{w}_i^*\right| \lesssim \frac{\sigma\sqrt{\log d}}{\sqrt{n}} \text{ for } i \in \text{supp}(\mathbf{w}^*), \qquad \left|\mathbf{w}_i^{T^*} - \mathbf{w}_i^*\right| \lesssim \frac{1}{d} \text{ for } i \notin \text{supp}(\mathbf{w}^*).$$

*Meanwhile, the query (gating) iterate $\mathbf{q}^t = \tilde{\mathbf{v}}^t$ obeys $\tilde{\mathbf{v}}^t = \hat{\mathbf{u}} \log t + \boldsymbol{\rho}^t$, where $\hat{\mathbf{u}}$ is the max-margin solution on $\{\tilde{\mathbf{x}}_i - \tilde{\mathbf{z}}_i\}_{i=1}^n$ and $\boldsymbol{\rho}^t$ a bounded residual. Consequently, $p_i^{T^*} \geq 1 - \frac{1}{d^2}$ for all $1 \leq i \leq n$.*

This theorem explains the mechanism by which CCO arises during optimization. Under the dominant-coordinate condition, the dominant causal direction becomes visible to GD: the gate's gradient aligns with the separation direction $(\tilde{\mathbf{x}}^i - \tilde{\mathbf{z}}^i)$ and tracks a max-margin ray with a logarithmically diverging norm, so the attention weight concentrates on the $\mathbf{x}$-branch. As the gate filters out the descendant branch, the squared-parameter head fits the ground-truth weights $\mathbf{w}^*$ up to the error on active coordinates and a $1/d$ tail on inactive ones.

**Role of Positional Encodings.** Distinct fixed encodings $\mathbf{s}_1 \neq \mathbf{s}_2$ attach branch identity and introduce a sample-independent margin in the gate logit, $(\tilde{\mathbf{v}}^t)^\top(\mathbf{s}_1 - \mathbf{s}_2)$. This symmetry breaking enables the two-key attention to identify the dominant feature and drive the attention weights to select the branch associated with it, thereby catalyzing CCO.

**Theorem 2** (Generalization of CCO). *For an independent test triple $(\mathbf{x}, y, \mathbf{z})$, there exists an event $\Omega$ with $\Pr(\Omega) \geq 1 - \frac{8\sqrt{\|\mathbf{s}\|_2^2 + d(B_{\mathbf{x}} + B_{\mathbf{z}})^2}}{\|\mathbf{s}\|_2\sqrt{n}} - \sqrt{\frac{2\ln(2d^2)}{n}}$, such that conditioned on $\Omega$,*

$$p^{T^*} = \sigma\left((\tilde{\mathbf{v}}^{T^*})^\top(\tilde{\mathbf{x}}^{T^*} - \tilde{\mathbf{z}}^{T^*})\right) \geq 1 - \frac{1}{d^2} \quad and \quad \mathbb{E}\left[\left|\mathcal{L} - \frac{\sigma^2}{2}\right| \, \Big| \, \Omega\right] \lesssim \frac{r\sigma^2 \log d}{n}.$$

With high probability (strengthened when $\|\mathbf{s}\|_2^2 \asymp d$), the learned gate continues to prefer the causal branch on test distribution, i.e., $p^{T^*}$ is bounded away from $0$ and close to $1$. Moreover, the test loss approaches the cause only noise floor $\sigma^2/2$ at rate $O(r\,\sigma^2 \log d/n)$, indicating that the predictor essentially relies on $\mathbf{x}$ while filtering out $\mathbf{z}$ on the test distribution.

Theorem 2 controls generalization when train and test share the same data distribution. We next show that the same CCO predictor remains robust under test time shifts that perturb $y \to \mathbf{z}$.

**Corollary 1** (Robust generalization under $y \to \mathbf{z}$ shifts)**.** *At test time, change the $y \to \mathbf{z}$ mechanism so that $\mathbf{z}' = f'(y) + \boldsymbol{\xi}'$ and assume $\sup_j |\mathbf{z}'_j| \le B_{\mathbf{z}'}$. There exists an event $\Omega$ with $\Pr(\Omega) \ge 1 - \frac{8\sqrt{\|\mathbf{s}\|_2^2 + d\,(B_{\mathbf{x}} + B_{\mathbf{z}'})^2}}{\|\mathbf{s}\|_2 \sqrt{n}} - \sqrt{\frac{2\ln(2d^2)}{n}}$, such that conditioned on $\Omega$,*

$$p^{T^*} = \sigma\big((\tilde{\mathbf{v}}^{T^*})^\top (\tilde{\mathbf{x}}^{T^*} - \tilde{\mathbf{z}}'^{,T^*})\big) \ge 1 - \frac{1}{d^2}, \quad \mathbb{E}\left[ \left| \mathcal{L}_{(\mathbf{x},y,\mathbf{z}')}\big(\tilde{\mathbf{w}}^{T^*}, \tilde{\mathbf{v}}^{T^*}\big) - \frac{\sigma^2}{2} \right| \,\Big|\, \Omega \right] \lesssim \frac{r\,\sigma^2 \log d}{n}.$$

## 5 FURTHER DISCUSSION

**Positioning of CCO.** CCO arises under purely correlational training with single environment, no environment labels, and no explicit invariance regularizers. Yet when a dominant causal correlation is present and GD's implicit bias takes hold, the learned predictor moves beyond correlation toward causation: it increasingly relies on causal features while largely discounting spurious correlates. Meanwhile, multi-environment invariance methods also seek causally aligned predictors, but they pursue this goal by explicitly leveraging cross environment heterogeneity.

**When Can Transformers Learn Causation?**
CCO offers a concrete path to cause only behavior under standard Transformer training, but it is not unique, and its assumptions need not always hold. In practice, Transformers/LLMs frequently exploit shortcuts and spurious cues (Bender et al., 2021; Du et al., 2023; Tang et al., 2023; Jin et al., 2024). CCO also has limits: it benefits from a strong causal correlation; when spurious cues are comparably strong or plentiful, single environment ERM may still lean on them. In this regime, multi-environment invariance learning that explicitly leverages heterogeneity remains essential for causal generalization.

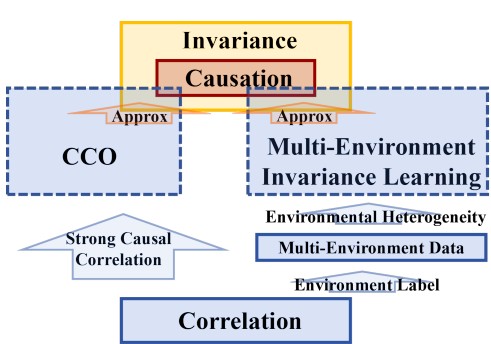

Figure 2: Positioning of CCO.

**Practical Insights.** CCO suggests actionable insightss for training: (i) amplify causal alignment in data to widen the dominant causal gap; (ii) employ mild attention sparsity or large step schedules to accentuate strong features. These steps do not enforce invariance, but they increase the likelihood that standard training will self select a cause only solution when the data permit.

## 6 EXPERIMENTS

### 6.1 SIMULATED EXPERIMENTS

We realize the GD on the two-key attention model in Algorithm 1 and present the simulation result in this section. We consider the case where the data are generated from the same causal chain $\mathbf{x} \to y \to \mathbf{z}$. The structural assignment for each variable is defined as $\mathbf{x} \sim \mathcal{N}(\sigma \mathbf{I}_d, \mu_x)$, $y = \mathbf{x}^\top (\mathbf{w}^*)^{\odot 2} + \epsilon$, $\mathbf{z} = \mathbf{C}y + \boldsymbol{\xi}$, where $\epsilon, \boldsymbol{\xi}$ are independent standard normal distributed and we set $\mathbf{w}^*$ as an all-ones vector. The results are shown in Fig 3. We calculate the weight $p_{x,i}^t$ and display its average across the batch $\bar{p}_x^t = \frac{1}{n} \sum_i p_{x,i}^t$. We then run GD for 5000 iterations with batchsize $n = 64$, and the dimension of data $d \in \{5, 10\}$. We can see that $\bar{p}_x^t$ increases rapidly to 1 in all cases in the first 100 iterations corresponding to the occupation phase, while in the crowding out stage $\bar{p}_x^t$ remains at 1, while $\mathbf{w}$ slowly decreases to the minimum value.

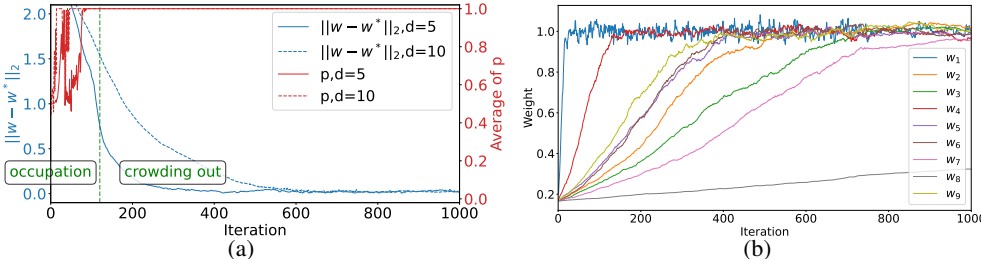

Figure 3: Simulation results for the GD on the two-key attention model. (a): the curve of $\|\mathbf{w} - \mathbf{w}^*\|_2$ and the average of $p$ with $d \in \{5, 10\}$. (b): the first component of $\mathbf{w}$ quickly reaches its optimum during occupation phase, while the other components slowly approach their optima during the crowding-out phase.

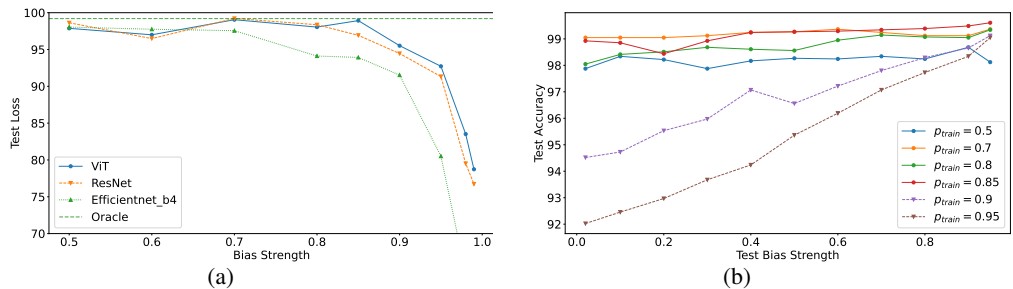

Figure 4: Experiments on waterbirds dataset. (a): The test accuracy with bias strength $p_{\text{test}} = 0.02$ bias strengths on DeiT-Small, ResNet34 and EfficientNet-B4 trained across a full sweep of training bias strengths from 0.5 to 0.99. Oracle is the accuracy on no-biased test data using DeiT-Small trained without bias. (b): The test accuracy with bias strength $p_{\text{test}}$ sweeping from 0.02 to 0.99 on DeiT-Small trained across a full sweep of training bias strengths from 0.5 to 0.95.

## 6.2 EXPERIMENTS ON REAL DATA

**Experiments on Vision Task.** We consider an image object classification task on the birds. The target is to classify water birds ($Y = 1$) and land birds ($Y = 0$ in the CUB dataset (Wah et al., 2011). To eliminate confounding due to foreground–background asymmetry altogether, we introduced a setting where one bird species on the left side serves as the true target label $y$ and another bird species on the right side acts as the spurious bias $z$, both appearing in the foreground. We set the bias strength in the train dataset to 0.9, i.e. $p_{\text{train}} = P(z = y|y) = 0.9$. This ensures that any observed attention shift cannot be attributed to low-level feature quality differences (e.g., texture richness or semantic complexity) between foreground and background.

The results in Fig 1 consistently show that the causal features progressively occupy and crowds out the spurious features (whether background or another bird). We find that the attention map on the left bird raise rapidly in the first 50 iterations, while the attention map on the right side seldom changes, illustrating the occupation phase. By iter 500, attention is concentrated almost entirely on the left side, with the bird on the right side receiving near zero weight, marking crowding-out. These findings confirm that the observed behavior reflects genuine optimization-driven cause preference not artifacts of feature disparity.

We conducted fair experiments on Waterbirds using DeiT-Small (from timm with ImageNet pretraining) alongside ResNet34 and EfficientNet-B4 (from torchvision, also pretrained, with comparable about 20M parameter counts), training all models for 1,000 epochs at a learning rate of 1e-4 across a full sweep of bias strengths from 0.5 to 0.99. As shown in the Fig 4 (a), DeiT-Small maintains significantly higher accuracy at strong bias levels (e.g., 0.9), demonstrating that Transformers can better capture the underlying causal signal—left side bird type—despite overwhelming spurious correlations with right bird, suggesting an advantage over CNNs in leveraging stronger semantic features when spurious cues dominate.

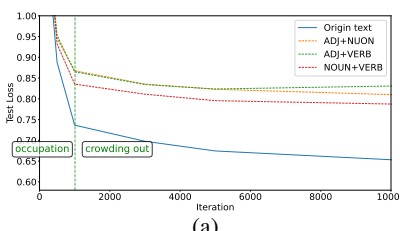 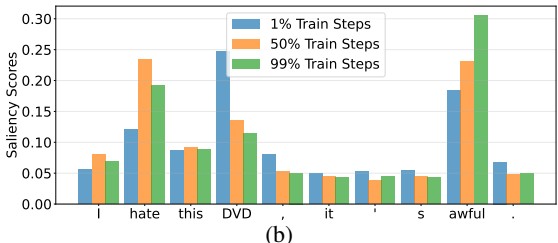

(a)                                         (b)

Figure 5: Experiment results on natural language task. (a): the test loss when mask the noun, adj, verb or their combination in the text. (b): the saliency scores of each token when input "I hate this DVD, it's awful." to the model at $1\%, 50\%, 99\%$ of the training steps.

We also added out-of-distribution (OOD) test experiments in Fig 4 (b). We constructed a water-bird dataset with a base spurious correlation of varying training bias strengths $p_{\text{train}}$, measuring test accuracy on OOD data where the test bias strengths $p_{\text{test}} \in [0,1]$. The curve reveals that when $p_{\text{train}} \geq 0.9$, test accuracy drops as bias increases, indicating that the model fails to learn the invariant causal feature (bird type) and instead relies heavily on the spurious background cue. However, once $p_{\text{train}} \leq 0.85$, test accuracy rises significantly and remains high (above 95%), which is the hallmark of CCO: the model effectively crowd out the spurious features and learn the cause only prediction. Therefore, when the spurious correlation is under the threshold, transformer can obtain a cause only predictor which exhibits robust generalization at test time.

**Experiments on Natural Language Task.** We conduct the sentiment classification task on the Amazon reviews dataset (He & McAuley, 2016) which consists of reviews from amazon. Here $Y \in \{1, 2, 3, 4, 5\}$ represents the reviewer's rating, $X$ denotes the associated adjectives and verbs, and $Z$ indicates the nouns related to the product itself. We finetune the bert-base-uncased model Devlin et al. (2019) for 50k steps, employing the Adam optimizer Kingma & Ba (2014) with a learning rate of 1e-5. When constructing the test data, we mask the noun, adj, verb or their combination in the text. As shown in Fig 5 (a), test loss with masked NOUN+VERB decay rapidly corresponding to the occupation phase. We also observe a final upward trend in the test loss with masked ADJ+VERB, indicating that the attention allocated to NOUNs is being crowded out by cause features. Fig 5 (b) display the saliency scores computed by the gradients of target class score relative to input embeddings, which show which tokens most influence the model's decision. The result indicates that the cause features (hate, awful) crowds out the spurious features during the training process.

## 7 CONCLUSION

In this paper, we identify a new training phenomenon for Transformers training dynamics called CCO, showing that strong causal alignment in the data, coupled with the implicit regularization of GD, can drive the model toward cause only prediction. We demonstrate CCO empirically and develop a theoretical account of its two phase mechanism (occupation and crowding-out). While not the only route to causal generalization, CCO offers a concrete answer to when and through what dynamics standard Transformer training can suppress spurious features and rely almost exclusively on causal ones. The results spark that: amplifying causal alignment in data and designing training procedures that accentuate causal signals can make Transformers more likely to learn causally grounded predictors.

## 8 ACKNOWLEDGMENTS

This work is supported by the NSF China (No.s 92470117 and 62376008).

## 9 ETHICS STATEMENT

Our paper complies with the ICLR Code of Ethics.

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

# A APPENDIX

## A.1 USE OF LLMs

We used LLMs for language polishing.

## A.2 ADDITIONAL EXPERIMENT

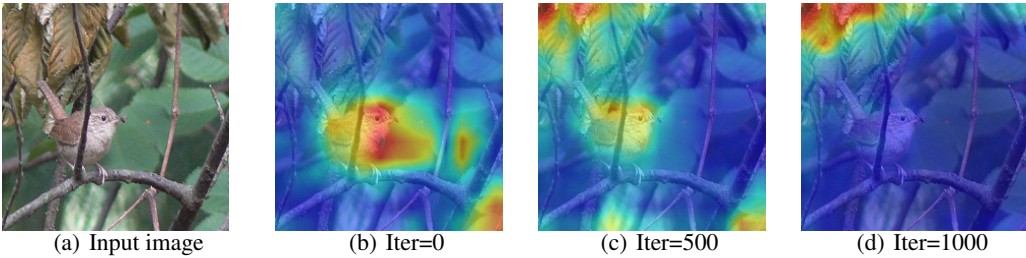

| (a) Input image | (b) Iter=0 | (c) Iter=500 | (d) Iter=1000 |

Figure 6: This figure shows how attention shifts during ViT training on a background-label (target) task. (a) is the input image. Early in training (b, iter 0), attention is diffuse across background (causal feature) and bird (spurious feature). As training proceeds (c, iter 500), attention weight rises on the background illustrating the occupation phase. By (d, iter 1000), attention is concentrated almost entirely on the background, with the bird receiving near zero weight, marking crowding-out.

**Background-label (target) task.** We consider an image object classification task on the background with birds. The target is to classify water environment ($Y = 1$) and land environment ($Y = 0$). We generate datasets by combining the bird images in the CUB dataset (Wah et al., 2011) and the background images in the Places dataset (Zhou et al., 2018) using specific probabilities, which is similar to the waterbird setting in Sagawa et al. (2020) with different target. We set the pixels related to birds as z and place 70% of all water birds against a water background and 70% of all land birds against a land background, generating a dataset with 30k images. We then train the vision Transformer model (Dosovitskiy et al., 2021) using the dataset, fixing the input image size to 224, with patch size set to 16, learning rate set to 1e-4, and batch size set to 16. The results are displayed in Fig 6. Initially, attention grows rapidly in the background with only a slight increase on the bird. Later, during crowding-out, the map is rapidly dominated by background attention, while the bird's attention becomes very faint.

**Sensitive of sign-stability.** In order to empirically verify how sensitive our mechanism is to mild violations of sign-stability, we consider the image object classification task on the background with birds. The target is to classify water bird ($Y = 1$) and land bird ($Y = 0$). We generate datasets by combining the bird images in the CUB dataset (Wah et al., 2011) and the background images in the Places dataset (Zhou et al., 2018) using specific probabilities. But here we flip the label $Y$ with probability $p_{\text{flip}}$:

$$\hat{Y} = \begin{cases} 1 - Y, & \text{with probability } p_{\text{flip}}, \\ Y, & \text{with probability } 1 - p_{\text{flip}}. \end{cases}$$

We also place $p_{\text{train}}$ of all water birds against a water background and $p_{\text{train}}$ of all land birds against a land background, generating a dataset with 30k images. We then train the vision Transformer model using the dataset, fixing the input image size to 224, with patch size set to 16, learning rate set to 1e-4, and batch size set to 16. As shown in Fig 7, we scan the $p_{\text{flip}}$ from 0 to 0.5, and find that when $p_{\text{flip}} = 0.2$, the crowding-out behavior can still be observed in the model with the accuracy reaches over 90%, proving that the mechanism is sign-stable. When $p_{\text{flip}} \leq 0.15$, the test accuracy remains robust across different bias strengths, indicating that CCO is still effective and the model is able to learn invariant cause only prediction.

**CCO boundary conditions.** To empirically characterize the boundary conditions of CCO, we introduce controlled cause-predictive correlation by flipping the true bird class label $Y$ with probability $p_{\text{flip}}$ in a Waterbirds-like setup, where water birds and land birds are composited onto matching backgrounds with strength $p_{\text{train}}$ (measuring spurious-predictive correlation). As shown in Fig 7,

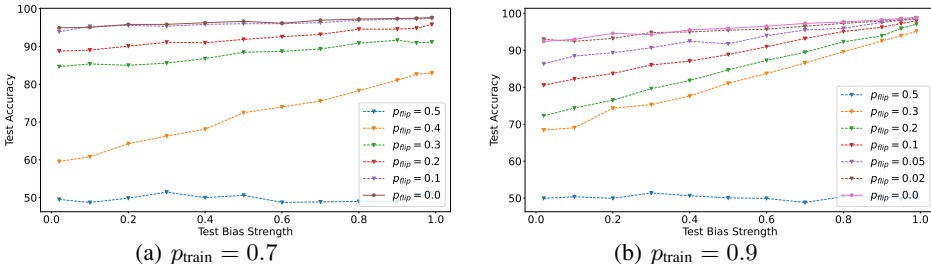

Figure 7: Test Accuracy on Deit-Small trained across a full sweep of label flip probability $p_{\text{flip}}$ from $0$ to $0.5$ with $p_{\text{train}}$ fixed to $0.7$ and $0.9$. We sweep the test bias strength $p_{\text{test}}$ from $0.02$ to $0.99$.

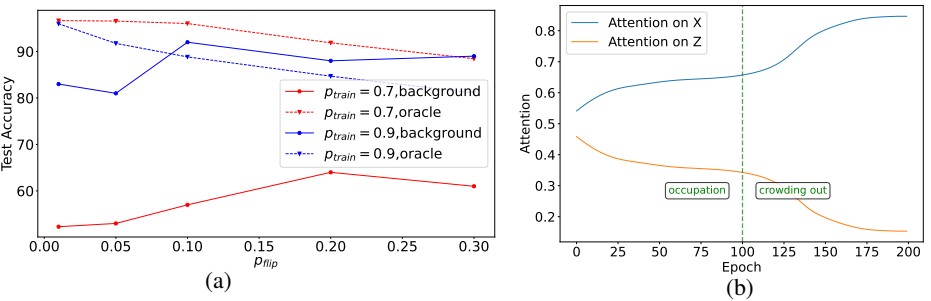

Figure 8: (a): Test accuracy of Deit-Small trained on dataset with various $p_{\text{flip}}$ and $p_{\text{train}}$ on the bird classification task. The background curves show test accuracy on the background dataset without bird. The oracle curves show test accuracy on the dataset with $p_{\text{test}} = 0.5$. (b): The dynamic of attention weight for $\mathbf{x}, \mathbf{z}$ during training in simulation on standard transformer.

when $p_{\text{flip}} \leq 0.3$ in (a) or $p_{\text{flip}} \leq 0.05$ in (b), the Vision Transformer maintains high and stable test accuracy across varying bias strengths, and its attention concentrates on the bird rather than the background—evidence that CCO is active and the model learns an invariant predictor based on the causal features $\mathbf{x}$. Even at $p_{\text{flip}} = 0.2$ in Fig 7(a), the model still achieves over $90\%$ accuracy and exhibits crowding-out behavior, demonstrating robustness to mild violations of sign-stability. However, as $p_{\text{flip}}$ increases further toward $0.4$ or $0.5$, the performance deteriorates sharply, indicating that CCO collapses once the correlation gap falls below a critical threshold. These results establish that CCO operates effectively when the cause-predictive correlation remains sufficiently stronger than the spurious one, defining a practical boundary beyond which the mechanism no longer reliably emerges.

**Generalization effects of spurious-dominant correlation.** We test the accuracy of ViT on the background dataset without bird in a specular setting with dominant correlations between $y$ and $\mathbf{z}$, and weaker ones between $\mathbf{x}$ and $y$. The results are shown in Fig 8 (a). When the correlation between $y$ and $\mathbf{z}$ in the dataset is strong, we observe a specular result: the model achieves high accuracy on the background-only test set, indicating that it primarily relies on features associated with $\mathbf{z}$ for prediction. In contrast, when the correlation between $y$ and $\mathbf{z}$ is relatively weaker compared to the correlation between $\mathbf{x}$ and $y$, the model's accuracy on the background-only test set becomes very low. In this regime, the CCO mechanism emerges: the model's attention focuses predominantly on causal features $\mathbf{x}$, effectively crowding out those spurious features $\mathbf{z}$.

**Image classification on CelebA.** We conduct our experiment on the classification task on CelebA dataset. This classification task aims to predict the presence of a beard from CelebA images, where the target label is spuriously correlated with gender. We trained ResNet-34, EfficientNet-B4, and DeiT-Small with comparable parameter counts on this dataset under standard settings, using the AdamW optimizer with a learning rate of 1e-4.

Table 1: The accuracy of ResNet-34, EfficientNet-B4, and DeiT-Small on train dataset and test dataset including (1) (Test Set 1) masking out facial regions, and (2) (Test Set 2) masking out everything except the facial regions.

| Model | Train Accuracy | Test Set 1 Accuracy ($\downarrow$) | Test Set 2 Accuracy ($\uparrow$) |
|---|---|---|---|
| Deit-small | 0.987 | **0.552** | **0.893** |
| ResNet-34 | 0.992 | 0.577 | 0.861 |
| EfficientNet-B4 | 0.979 | 0.573 | 0.802 |

Table 2: The loss of BERT trained on datasets with various $P(z \mid y)$. The table shows the loss on train dataset and test dataset including (1) (Oracle Test): $z$ and $y$ are independent, $P(z \mid y) = 0.5$, and (2) (Biased Test): $P(z \mid y) = 0.02$.

| $P(z \mid y)$ | Final Train Loss | Oracle Test Loss ($\downarrow$) | Biased Test Loss ($\downarrow$) |
|---|---|---|---|
| 0.5 | 0.64 | 0.61 | 0.65 |
| 0.9 | 0.62 | 0.65 | 0.68 |
| 0.95 | 0.67 | 0.77 | 0.87 |

In the test set, we evaluated two masking conditions based on the bounding box (bbox) annotations provided by the dataset: (1) (Test Set 1) masking out facial regions, and (2) (Test Set 2) masking out everything except the facial regions and the result are shown in Tab 1. On Test Set 2, DeiT-Small outperformed both ResNet-34 and EfficientNet-B4, indicating the CCO mechanism of crowding out spurious features for accurate beard prediction. The performance gap observed on Test Set 2, where only facial regions are visible, underscores that when the dataset contains strong but misleading associations (like gender bias), DeiT-Small leverages its capacity to attend to all parts of the image equally and identify the most predictive elements—the beard itself—thus achieving higher accuracy. This supports the hypothesis that under certain conditions, particularly those involving complex spurious correlations, Transformers exhibit a robustness and adaptability that enables them to focus on invariant cause only prediction, enhancing their generalization capabilities on unseen data.

**Controllable Spuriousness evaluation in NLP task.** We construct NLP evaluation settings where the degree of spurious correlation is known and controllable. When ground-truth labels $y$ are available and the data-generating process allows intervention, we can deliberately manipulate the association between a potentially spurious variable $z$ (e.g., the name of item) and the label $y$. By sampling instances according to a fixed conditional distribution $P(z \mid y) = p$, we can break or calibrate the spurious link between $z$ and $y$.

The Amazon reviews dataset provide the label of scores which is the target $y$ and the name of item, which is a measurement of $z$. Varying $p$ across experimental conditions allows systematic study of how model behavior changes with the strength of the $z$–$y$ association. The table shows the final test loss of BERT under various $p$, where BERT remains lower test loss when $p = 0.9$, demonstrating that transformers can pick up the stronger causal signal in NLP data.

**Simulation on standard transformer.** We conduct simulation experiments on standard multi-token transformer. We take a two-token $\mathbf{X} = [\mathbf{x}, \mathbf{z}]$ as the input, and the causal chain is $\mathbf{x} \to y \to \mathbf{z}$, where $\mathbf{x}, \mathbf{z} \in \mathbb{R}^d$ are vector covariates. We set

$$y = \mathbf{x}^\top \mathbf{w}_x + \epsilon, \qquad \mathbf{z} = \mathbf{w}_z y + \xi$$

Here $\epsilon$ and $\xi$ are both Gaussian random vectors, with variances of $0.1$ and $1$, respectively. We set $\mathbf{w}_x = \mathbf{1}_d$, $\mathbf{w}_z = 0.1 \cdot \mathbf{1}_d$. We then train a 2-layer standard multi-token Transformer with a learning rate of 1e-3 and the dynamic of attention weight for $\mathbf{x}, \mathbf{z}$ during training is shown in Fig 8. The attention weight curve demonstrates that the model initially assigns comparable attention to both the cause $X$ and its effect $Z$, but shows a two-stage shifts focus toward $X$ while sharply suppressing attention to $Z$. This "occupation" and "crowding out" behavior aligns with the CCO mechanism. Consequently, the model learns to rely on direct evidence rather than attending to indirect, spurious predictive pathway.

## A.3 PROOF OF THEOREM 1

The proof of Theorem 1 in three stages. In Stage 1, we show that the squared-parameter FFN rapidly amplifies the weight on the dominant causal coordinate while keeping all other coordinates small. In Stage 2, GD on the gate parameter $\tilde{\mathbf{v}}^t$ towards the max-margin solution on $\{\tilde{\mathbf{x}}_i - \tilde{\mathbf{z}}_i\}_{i=1}^n$ drives $p_i^t \to 1$ and thereby squeezes out the descendant branch. Finally, in Stage 3, after the descendant $z$ is nearly excluded, the squared-parameter FFN recovers the sparse ground truth.

### A.3.1 STAGE 1

**Theorem 3.** *Let $T_1^* = \min\left\{t \in \mathbb{N} : \mathbf{w}_1^t \geq \frac{1}{4}\right\}$. Under Condition 1, suppose the step sizes satisfy $\eta_t \equiv \eta < \frac{1}{2\left(\left|s_1^{\text{eff}}\right| + \frac{m_1}{32}\right)}$, and $\beta_t \equiv 0$, and the initialization scale is $\alpha = \frac{\sqrt{\sigma^2 \log d/n}}{d^3}$, Then with probability at least $1 - \frac{1}{d^2}$, the iterate $\tilde{\mathbf{w}}^{T_1^*} = \begin{bmatrix} \mathbf{0} \\ \mathbf{w}^{T_1^*} \end{bmatrix}$ satisfies $\frac{1}{4} \leq \mathbf{w}_1^{T_1^*} \leq \frac{1}{2}$ and $\left|\mathbf{w}_j^{T_1^*}\right| \leq \frac{\sqrt{\sigma^2 \log d/n}}{d^2}$ for $j > 1$.*

*Proof of Theorem 3.* Throughout Stage 1 we set $\beta_t \equiv 0$, hence $\tilde{\mathbf{v}}^t \equiv \mathbf{0}$ for all $1 \leq t \leq T_1^*$. With the two-key attention, this implies $p_i^t = \sigma(0) = \frac{1}{2}$ for every sample $1 \leq i \leq n$ and iteration $1 \leq t \leq T_1^*$.

By the structure of the squared-parameter head, $\tilde{\mathbf{w}}^t$ keeps the form $\tilde{\mathbf{w}}^t = \begin{bmatrix} \mathbf{0} \\ \mathbf{w}_1^t \end{bmatrix}$.

The update of $\mathbf{w}^t$ satisfies

$$\mathbf{w}^{t+1} = \mathbf{w}^t - \frac{\eta}{n} \sum_{i=1}^n \left(\frac{1}{2}\left(\mathbf{x}^i + \mathbf{z}^i\right)^\top \left(\mathbf{w}^t\right)^{\odot 2} - \left(\mathbf{x}^i\right)^\top \left(\mathbf{w}^*\right)^{\odot 2} - \epsilon_i\right)\left(\mathbf{x}^i + \mathbf{z}^i\right) \odot \mathbf{w}^t. \quad (1)$$

We analyze non-dominant coordinates ($j > 1$) and the dominant coordinate ($j = 1$) in turn.

By the defination of $T_1^*$, we have $\mathbf{w}_1^t < 1/4$, for $t < T_1^*$. We now prove for $j > 1$ and $1 \leq t \leq \left\lceil \frac{\log(d^3/4(\sqrt{\sigma^2 \log d/n}))}{\log(1+\eta\kappa)} \right\rceil \wedge T_1^*$, $\kappa = \frac{15}{16}s_1^{\text{eff}} - \frac{m_1}{32}$, $\left|\mathbf{w}_j^{T_1^*}\right| \leq \frac{\sqrt{\sigma^2 \log d/n}}{d^2}$ holds by induction.

The update for coordinate $j > 1$ is

$$\mathbf{w}_j^{t+1} = \mathbf{w}_j^t - \frac{\eta}{2}\left[\frac{1}{n}\sum_{i=1}^n (\mathbf{x}_1^i + \mathbf{z}_1^i)(\mathbf{x}_j^i + \mathbf{z}_j^i)\right](\mathbf{w}_1^t)^2(\mathbf{w}_j^t) + \eta\left[\frac{1}{n}\sum_{i=1}^n (\mathbf{x}^i)^\top(\mathbf{w}^*)^{\odot 2}(\mathbf{x}_j^i + \mathbf{z}_j^i)\right]\mathbf{w}_j^t$$

$$+ \eta\left[\frac{1}{n}\sum_{i=1}^n \epsilon^i(\mathbf{x}_j^i + \mathbf{z}_j^i)\right]\mathbf{w}_j^t - \frac{\eta}{2n}\sum_{i=1}^n\sum_{k=2}^d (\mathbf{x}_k^i + \mathbf{z}_k^i)^2(\mathbf{w}_k^t)^2(\mathbf{x}_j^i + \mathbf{z}_j^i)\mathbf{w}_j^t.$$

Separating population terms from sampling deviations gives the multiplicative form

$$\mathbf{w}_j^{t+1} = \mathbf{w}_j^t + \eta\left(s_j^{\text{eff}} - \frac{m_{1j}}{2}(\mathbf{w}_1^t)^2\right)(\mathbf{w}_j^t) + \eta\Delta_j^t, \quad (2)$$

where $\Delta_j^t$ is expressed as the following:

$$\Delta_j^t = -\frac{1}{2}\left[\frac{1}{n}\sum_{i=1}^n \left(\mathbf{x}_1^i + \mathbf{z}_1^i\right)\left(\mathbf{x}_j^i + \mathbf{z}_j^i\right) - m_{1j}\right]\left(\mathbf{w}_1^t\right)^2\left(\mathbf{w}_j^t\right)$$

$$+ \left[\frac{1}{n}\sum_{i=1}^n \left(\mathbf{x}^i\right)^\top\left(\mathbf{w}^*\right)^{\odot 2}\left(\mathbf{x}_j^i + \mathbf{z}_j^i\right) - s_j\right]\mathbf{w}_j^t + \left[\frac{1}{n}\sum_{i=1}^n \epsilon^i\left(\mathbf{x}_j^i + \mathbf{z}_j^i\right) - \mu_j\right]\mathbf{w}_j^t$$

$$- \frac{1}{2n}\sum_{i=1}^n\sum_{k=2}^d \left(\mathbf{x}_k^i + \mathbf{z}_k^i\right)^2\left(\mathbf{w}_k^t\right)^2\left(\mathbf{x}_j^i + \mathbf{z}_j^i\right)\mathbf{w}_j^t.$$

By Lemma 6 (concentration) and boundedness, using $\mathbf{w}_1^t < \frac{1}{4}$ and the inductive hypothesis $\left|\mathbf{w}_j^t\right| \leq \frac{\sqrt{\sigma^2 \log d/n}}{d^2}$, we obtain

$$\left|\Delta_j^t\right| \leq (\phi_2 + \phi_\epsilon)\left|\mathbf{w}_j^t\right| + \frac{\phi_1}{32}\left|\mathbf{w}_j^t\right| + \frac{B_{\mathbf{x}+\mathbf{z}}^3}{2d^2}\left|\mathbf{w}_1^t\right| = e_j\left|\mathbf{w}_j^t\right|, \quad (3)$$

where $e_j = \mathcal{O}(d^{-2})$ and, for large $d$, $|e_j| \leq \frac{1}{4} \max_{j>1} \left( \left| s_j^{\text{eff}} \right| + \frac{m_{1j}}{32} \right)$. Combining equation 2 and equation 3,

$$
\begin{aligned}
\frac{|\mathbf{w}_j^{t+1}|}{|\mathbf{w}_j^t|} &\leq 1 + \eta \left( \max_{j>1} \left( \left| s_j^{\text{eff}} \right| + \frac{m_{1j}}{32} \right) + e_j \right) \\
&\leq 1 + \eta \left( \frac{5}{4} \max_{j>1} \left( \left| s_j^{\text{eff}} \right| + \frac{m_{1j}}{32} \right) \right).
\end{aligned}
\tag{4}
$$

By the above inequality, $|\mathbf{w}_j^{t+1}|$ can be bounded by

$$
\begin{aligned}
|\mathbf{w}_j^{t+1}| &\overset{(1)}{\leq} \left( 1 + \eta \left( \frac{5}{4} \max_{j>1} \left( \left| s_j^{\text{eff}} \right| + \frac{m_{1j}}{32} \right) \right) \right) |\mathbf{w}_j^t| \\
&\overset{(2)}{\leq} \left( 1 + \eta \left( \frac{5}{4} \max_{j>1} \left( \left| s_j^{\text{eff}} \right| + \frac{m_{1j}}{32} \right) \right) \right)^{\left\lceil \frac{\log(d^3/4(\sqrt{\sigma^2 \log d/n}))}{\log(1+\eta\kappa)} \right\rceil \wedge T_1^*} \frac{\sqrt{\sigma^2 \log d/n}}{d^3} \\
&\leq \left( 1 + \eta \left( \frac{5}{4} \max_{j>1} \left( \left| s_j^{\text{eff}} \right| + \frac{m_{1j}}{32} \right) \right) \right)^{\left\lceil \frac{\log(d^3/4(\sqrt{\sigma^2 \log d/n}))}{\log(1+\eta\kappa)} \right\rceil} \frac{\sqrt{\sigma^2 \log d/n}}{d^3} \\
&\overset{(3)}{\leq} \frac{\sqrt{\sigma^2 \log d/n}}{d^2}.
\end{aligned}
\tag{5}
$$

where $(1)$ uses equation 4, $(2)$ uses $\alpha = \frac{\sqrt{\sigma^2 \log d/n}}{d^3}$ and $(3)$ uses Condition 1.

Therefore, by induction, for all $j > 1$ and all $1 \leq t \leq \left\lceil \frac{\log(d^2/4)}{\log(1+\eta\kappa)} \right\rceil \wedge T_1^*$, we have $|\mathbf{w}_j^t| \leq \frac{\sqrt{\sigma^2 \log d/n}}{d^2}$.

We then prove for $j = 1$ and $1 \leq t \leq \left\lceil \frac{\log(d^2/4)}{\log(1+\eta\kappa)} \right\rceil \wedge T_1^*$, $\mathbf{w}_1^{t+1} \geq (1+\eta\kappa)\mathbf{w}_1^t$, where $\kappa = \frac{15}{16} s_1^{\text{eff}} - \frac{m_1}{32}$.

The update for $\mathbf{w}_1^{t+1}$ is given by

$$
\begin{aligned}
\mathbf{w}_1^{t+1} = \mathbf{w}_1^t &- \frac{\eta}{2} \left[ \frac{1}{n} \sum_{i=1}^n (\mathbf{x}_1^i + \mathbf{z}_1^i)^2 \right] (\mathbf{w}_1^t)^3 + \eta \left[ \frac{1}{n} \sum_{i=1}^n (\mathbf{x}^i)^\top (\mathbf{w}^*)^{\odot 2} (\mathbf{x}_1^i + \mathbf{z}_1^i) \right] \mathbf{w}_1^t \\
&+ \eta \left[ \frac{1}{n} \sum_{i=1}^n \epsilon^i (\mathbf{x}_1^i + \mathbf{z}_1^i) \right] \mathbf{w}_1^t - \frac{\eta}{2n} \sum_{i=1}^n \sum_{j=2}^d (\mathbf{x}_j^i + \mathbf{z}_j^i)^2 (\mathbf{w}_j^t)^2 (\mathbf{x}_1^i + \mathbf{z}_1^i) \mathbf{w}_1^t.
\end{aligned}
$$

Separate expectations and deviations, we have

$$
\begin{aligned}
\mathbf{w}_1^{t+1} = \mathbf{w}_1^t &+ \eta \left( s_1^{\text{eff}} \mathbf{w}_1^t - \frac{m_1}{2} (\mathbf{w}_1^t)^3 \right) + \eta \Delta_1^t, \\
\Delta_1^t = &-\frac{1}{2} \left[ \frac{1}{n} \sum_{i=1}^n (\mathbf{x}_1^i + \mathbf{z}_1^i)^2 - m_1 \right] (\mathbf{w}_1^t)^3 + \left[ \frac{1}{n} \sum_{i=1}^n (\mathbf{x}^i)^\top (\mathbf{w}^*)^{\odot 2} (\mathbf{x}_1^i + \mathbf{z}_1^i) - s_1 \right] \mathbf{w}_1^t \\
&+ \left[ \frac{1}{n} \sum_{i=1}^n \epsilon^i (\mathbf{x}_1^i + \mathbf{z}_1^i) - \mu_1 \right] \mathbf{w}_1^t - \frac{1}{2n} \sum_{i=1}^n \sum_{j=2}^d (\mathbf{x}_j^i + \mathbf{z}_j^i)^2 (\mathbf{w}_j^t)^2 (\mathbf{x}_1^i + \mathbf{z}_1^i) \mathbf{w}_1^t.
\end{aligned}
$$

By Lemma 6 and boundedness,

$$
|\Delta_1^t| \leq (\phi_2 + \phi_\epsilon) |\mathbf{w}_1^t| + \frac{\phi_1}{2} |\mathbf{w}_1^t|^3 + \frac{B_{\mathbf{x}+\mathbf{z}}^3}{2} \sum_{j=2}^d (\mathbf{w}_j^t)^2 |\mathbf{w}_1^t|.
\tag{6}
$$

Since for $j > 1$, $|\mathbf{w}_j^t| \leq \frac{\sqrt{\sigma^2 \log d/n}}{d^2}$ and $0 < \mathbf{w}_1^t \leq 1/2$, we have $\sum_{j=2}^d (\mathbf{w}_j^t)^2 \leq 1/d$. Hence by equation 6,

$$
\frac{\mathbf{w}_1^{t+1}}{\mathbf{w}_1^t} \geq 1 + \eta \left( s_1^{\text{eff}} - \phi_2 - \phi_\epsilon - \frac{B_{\mathbf{x}+\mathbf{z}}^3}{2d} - \frac{m_1 + \phi_1}{32} \right) \geq 1 + \eta \kappa,
$$

This implies that for $1 \leq t \leq \left\lceil \frac{\log(d^3/4(\sqrt{\sigma^2 \log d/n}))}{\log(1+\eta\kappa)} \right\rceil \wedge T_1^*$, $\mathbf{w}_1^t \geq (1+\eta\kappa)^t \frac{\sqrt{\sigma^2 \log d/n}}{d^3}$. Therefore,

we obtain that $\left\lceil \frac{\log(d^3/4(\sqrt{\sigma^2 \log d/n}))}{\log(1+\eta\kappa)} \right\rceil \wedge T_1^* = T_1^*$. Then we have $|\mathbf{w}_j^{T_1^*}| \leq \frac{\sqrt{\sigma^2 \log d/n}}{d^2}$ for $j > 1$

and $\mathbf{w}_1^{T_1^*-1} \leq \frac{1}{4}$. Then,

$$\frac{\mathbf{w}_1^{t+1}}{\mathbf{w}_1^t} \leq 1 + \eta\left(s_1^{\text{eff}} + \phi_2 + \phi_\epsilon + \frac{B_{\mathbf{x}+\mathbf{z}}^3}{2d} + \frac{m_1 + \phi_1}{32}\right). \tag{7}$$

Since $\eta$ satisfies $\eta\left(s_1^{\text{eff}} + \phi_2 + \phi_\epsilon + \frac{B_{\mathbf{x}+\mathbf{z}}^3}{2d} + \frac{m_1+\phi_1}{32}\right) \leq 1$, we have $\mathbf{w}_1^{T_1^*} \leq \frac{1}{2}$.

Consequently, we obtain $\frac{1}{4} \leq \mathbf{w}_1^{T_1^*} \leq \frac{1}{2}$ and $|\mathbf{w}_j^{T_1^*}| \leq \frac{\sqrt{\sigma^2 \log d/n}}{d^2}$ for $j > 1$.

$\square$

The lemmas required for the Theorem 3 are listed below.

**Lemma 4.** *For any $1 \leq i \leq n$ and $1 \leq j \leq d$, $\left|\mathbf{z}_j^i\right| \leq \|f(0)\|_\infty + L(rB_{\mathbf{x}} + B_\epsilon) + B_{\boldsymbol{\xi}} := B_{\mathbf{z}}$.*

*Proof of Lemma 4.* Because $\mathbf{w}^* \in \{0,1\}^d$ and $|\text{supp}(\mathbf{w}^*)| \leq r$,

$$\left|\left(\mathbf{x}^i\right)^\top (\mathbf{w}^*)^{\odot 2}\right| = \left|\sum_{j \in \text{supp}(\mathbf{w}^*)} \mathbf{x}_j^i\right| \leq \sum_{j \in \text{supp}(\mathbf{w}^*)} |\mathbf{x}_j^i| \leq r B_{\mathbf{x}}. \tag{8}$$

Together with $\left|\epsilon^i\right| \leq B_\epsilon$,

$$|y^i| = \left|\left(\mathbf{x}^i\right)^\top (\mathbf{w}^*)^{\odot 2} + \epsilon^i\right| \leq rB_{\mathbf{x}} + B_\epsilon. \tag{9}$$

For any coordinate $1 \leq j \leq d$,

$$|f_j(y^i)| \leq |f_j(0)| + |f_j(y^i) - f_j(0)| \leq |f_j(0)| + \|f(y^i) - f(0)\|_\infty \leq |f_j(0)| + L|y^i|.$$

Taking $\sup_j$ and using equation 9 yields

$$\|f(y^i)\|_\infty \leq \|f(0)\|_\infty + L(rB_{\mathbf{x}} + B_\epsilon). \tag{10}$$

By the triangle inequality,

$$|\mathbf{z}_j^i| = |f_j(y^i) + \boldsymbol{\xi}_j^i| \leq |f_j(y^i)| + |\boldsymbol{\xi}_j^i| \leq \|f_j(y^i)\|_\infty + B_{\boldsymbol{\xi}}.$$

Combining with equation 10 gives

$$|\mathbf{z}_j^i| \leq \|f(0)\|_\infty + L(rB_{\mathbf{x}} + B_\epsilon) + B_{\boldsymbol{\xi}},$$

and thus $\|\mathbf{z}^i\|_\infty \leq B_{\mathbf{z}}$ almost surely.

$\square$

**Lemma 5** (Bernstein inequality for bounded distributions ( Theorem 2.9.5, Vershynin (2026))). *Let $\{\xi_i\}_{i=1}^n$ be independent, $\mathbb{E}\xi_i = 0$, $|\xi_i| \leq M$ a.s., and $\text{Var}(\xi_i) \leq v$. Then for any $t > 0$,*

$$\Pr\left(\left|\frac{1}{n}\sum_{i=1}^n \xi_i\right| \geq t\right) \leq 2\exp\left(-\frac{nt^2}{2v + \frac{2}{3}Mt}\right),$$

*hence with probability $\geq 1 - \delta$,*

$$\left|\frac{1}{n}\sum_{i=1}^n \xi_i\right| \leq \sqrt{\frac{2v\log(2/\delta)}{n}} + \frac{2M\log(2/\delta)}{3n}. \tag{B1}$$

**Lemma 6.** *Define* $B_{\mathbf{x+z}} := B_{\mathbf{x}} + B_{\mathbf{z}}$. *For any* $\delta \in (0,1)$, *with probability at least* $1 - \delta$, *the following hold simultaneously for all* $1 \leq j \leq d$:

$$\max_{1 \leq k,j \leq d} \left| \frac{1}{n} \sum_{i=1}^{n} \left( (\mathbf{x}_k^i + \mathbf{z}_k^i)(\mathbf{x}_j^i + \mathbf{z}_j^i) - m_{kj} \right) \right| \leq \phi_1 := 2B_{\mathbf{x+z}}^2 \left( \sqrt{\frac{2\log(6d/\delta)}{n}} + \frac{2\log(6d/\delta)}{3n} \right),$$

$$\max_{1 \leq j \leq d} \left| \frac{1}{n} \sum_{i=1}^{n} \left( (\mathbf{x}^i)^\top (\mathbf{w}^*)^{\odot 2} \right) (\mathbf{x}_j^i + \mathbf{z}_j^i) - s_j \right| \leq \phi_2 := r\, B_{\mathbf{x}} B_{\mathbf{x+z}} \left( \sqrt{\frac{2\log(6d/\delta)}{n}} + \frac{2\log(6d/\delta)}{3n} \right),$$

$$\max_{1 \leq j \leq d} \left| \frac{1}{n} \sum_{i=1}^{n} \epsilon^i (\mathbf{x}_j^i + \mathbf{z}_j^i) - \mu_j \right| \leq \phi_\epsilon := B_\epsilon B_{\mathbf{x+z}} \left( \sqrt{\frac{2\log(6d/\delta)}{n}} + \frac{2\log(6d/\delta)}{3n} \right),$$

*where* $B_{\mathbf{x+z}} := B_{\mathbf{x}} + B_{\mathbf{z}}$ *and the factor* $\log(6d/\delta)$ *accounts for a union bound over* $d$ *coordinates and the three families.*

*Proof of Lemma 6.* For fixed $1 \leq k, j \leq d$, set $\Xi_i^{(1)} := (\mathbf{x}_k^i + \mathbf{z}_k^i)(\mathbf{x}_j^i + \mathbf{z}_j^i) - m_{kj}$. Then $|\Xi_i^{(1)}| \leq M_2 := 2B_{\mathbf{x+z}}^2$ and $\mathrm{Var}(\Xi_i^{(1)}) \leq v_2 := M_2^2$. Applying Bernstein and union-bounding over $j$ with probability $\geq 1 - \delta/3$,

$$\max_j \left| \frac{1}{n} \sum_{i=1}^{n} \left( (\mathbf{x}_j^i + \mathbf{z}_j^i)^2 - m_j \right) \right| \leq \phi_1,$$

where

$$\phi_1 := 2B_{\mathbf{x+z}}^2 \left( \sqrt{\frac{2\log(6d/\delta)}{n}} + \frac{2\log(6d/\delta)}{3n} \right).$$

Let $S_i := (\mathbf{x}^i)^\top (\mathbf{w}^*)^{\odot 2}$ and $\Xi_i^{(2)} := S_i(\mathbf{x}_j^i + \mathbf{z}_j^i) - s_j$. Since $(\mathbf{w}^*)^{\odot 2} \in \{0,1\}^d$ with support size $\leq r$, $|S_i| \leq rB_{\mathbf{x}}$, hence $|\Xi_i^{(2)}| \leq M_{\mathrm{M}} := rB_{\mathbf{x}} B_{\mathbf{x+z}}$ and $\mathrm{Var}(\Xi_i^{(\mathrm{M})}) \leq v_{\mathrm{M}} := M_{\mathrm{M}}^2$. Bernstein union bound gives with probability at least $\geq 1 - \delta/3$

$$\max_j \left| \frac{1}{n} \sum_{i=1}^{n} \left( (\mathbf{x}^i)^\top (\mathbf{w}^*)^{\odot 2} \right)(\mathbf{x}_j^i + \mathbf{z}_j^i) - s_j \right| \leq \phi_2,$$

with

$$\phi_2 := r\, B_{\mathbf{x}} B_{\mathbf{x+z}} \left( \sqrt{\frac{2\log(6d/\delta)}{n}} + \frac{2\log(6d/\delta)}{3n} \right).$$

Let $\Xi_i^{(\epsilon)} := \epsilon^i(\mathbf{x}_j^i + \mathbf{z}_j^i) - \mu_j$. Then $|\Xi_i^{(\epsilon)}| \leq M_\epsilon := B_\epsilon B_{\mathbf{x+z}}$ and $\mathrm{Var}(\Xi_i^{(\epsilon)}) \leq v_\epsilon := M_\epsilon^2$. Bernstein plus union bound yields with probability $\geq 1 - \delta/3$

$$\max_j \left| \frac{1}{n} \sum_{i=1}^{n} \epsilon^i(\mathbf{x}_j^i + \mathbf{z}_j^i) - \mu_j \right| \leq \phi_\epsilon,$$

where

$$\phi_\epsilon := B_\epsilon B_{\mathbf{x+z}} \left( \sqrt{\frac{2\log(6d/\delta)}{n}} + \frac{2\log(6d/\delta)}{3n} \right).$$

$\square$

### A.3.2   STAGE 2

**Notation and max-margin solution.**   Set $\mathbf{s} := \mathbf{s}_1 - \mathbf{s}_2$ and define

$$\mathbf{u} := \begin{bmatrix} \mathbf{s} \\ \mathbf{x} - \mathbf{z} \end{bmatrix} \in \mathbb{R}^{M+d}, \qquad \|\mathbf{u}\|_2 \leq \sqrt{\|\mathbf{s}\|_2^2 + d\,(B_{\mathbf{x}} + B_{\mathbf{z}})^2} \text{ a.s.}$$

Let $\{\mathbf{u}^i\}_{i=1}^n$ be the samples form $\mathbf{u}$ and consider the empirical $\ell_2$ max-margin separator

$$\hat{\mathbf{u}} \in \arg \min_{\mathbf{w} \in \mathbb{R}^{M+d}} \frac{1}{2} \|\mathbf{w}\|_2^2 \text{ s.t. } (\mathbf{u}^i)^\top \mathbf{w} \geq 1, \ i = 1, \ldots, n.$$

Its empirical margin is $\gamma_{\text{emp}} := 1/\|\hat{\mathbf{u}}\|_2$. Let $\mathcal{S} \subseteq [n]$ be the support set with $(\mathbf{u}^i)^\top \hat{\mathbf{u}} = 1$ for $i \in \mathcal{S}$. By the KKT conditions there exist multipliers $\alpha_i \geq 0$, nonzero only on $\mathcal{S}$, such that

$$\hat{\mathbf{u}} = \sum_{i \in \mathcal{S}} \alpha_i \mathbf{u}^i.$$

**Lemma 7.** *Define* $\mathbf{u}_{\text{sep}} := \begin{bmatrix} \mathbf{s}/\|\mathbf{s}\|_2^2 \\ \mathbf{0} \end{bmatrix}$. *For every* $i$, $(\mathbf{u}^i)^\top \mathbf{u}_{\text{sep}} = 1$, *hence* $\mathbf{u}_{\text{sep}}$ *satisfies all margin constraints and separates the sample from the origin with margin* $1$. *Since* $\hat{\mathbf{u}}$ *minimizes* $\|\mathbf{u}\|_2$ *over the feasible set*, $\|\hat{\mathbf{u}}\|_2 \leq \|\mathbf{u}_{\text{sep}}\|_2 = 1/\|\mathbf{s}\|_2$, *and thus*

$$\gamma_{\text{emp}} = \frac{1}{\|\hat{\mathbf{u}}\|_2} \geq \|\mathbf{s}\|_2.$$

*Proof of Lemma 7.* Since $\mathbf{u}^i = [\mathbf{s}; \mathbf{x}^i - \mathbf{z}^i]$ and $\mathbf{u}_{\text{sep}} = [\mathbf{s}/\|\mathbf{s}\|_2^2; \mathbf{0}]$, $(\mathbf{u}^i)^\top \mathbf{u}_{\text{sep}} = \mathbf{s}^\top (\mathbf{s}/\|\mathbf{s}\|_2^2) = 1$. Thus $\mathbf{u}_{\text{sep}}$ is feasible and the norm bound follows. $\square$

By the properties established in Lemma 9, each sample gradient can be written as $\nabla \ell_i(\tilde{\mathbf{v}}^t) = \nabla \phi_i(\tilde{\mathbf{v}}^\top \mathbf{u}^i)$. Each $\phi_i$ is monotonically decreasing to zero, $C_{\phi''}$–smooth, and has a $(C_i, \mu_i)$ tight exponential tail. Hence, in Phase II the implicit bias of GD drives the direction toward the $\ell_2$ max-margin solution while the norm diverges, which forces the gate weight $p_i^t$ to converge to one. We formalize this as the following Theorem 8. The proof follows Soudry et al. (2018, Thm. 9), with minor adaptations to handle sample dependent $\phi_i$ and the dominant coordinate condition.

**Theorem 8.** *Let* $T_2^* \asymp d^2/\sqrt{\sigma^2 \log d/n}$. *Under Condition 2, suppose the step sizes satisfy* $\eta_t = 0$, $\beta_t = \beta < \frac{n}{C_{\phi''} \sigma_{\max}(\mathbf{U})^2}$, *where* $C_{\phi''}$ *is a constant defined in Lemma 9, there exists a bounded residual* $\boldsymbol{\rho}^t$ *such that*

$$\tilde{\mathbf{v}}^t = \hat{\mathbf{u}} \log t + \boldsymbol{\rho}^t,$$

*and, in particular,*

$$\lim_{t \to \infty} \frac{\tilde{\mathbf{v}}^t}{\|\tilde{\mathbf{v}}^t\|_2} = \frac{\hat{\mathbf{u}}}{\|\hat{\mathbf{u}}\|_2}, \qquad \min_i (\tilde{\mathbf{v}}^t)^\top \mathbf{u}^i \sim \log t \to +\infty.$$

*Consequently, at* $T_1^* + T_2^*$, $p_i^{T_1^* + T_2^*} = \sigma((\tilde{\mathbf{v}}^{T_1^* + T_2^*})^\top \mathbf{u}^i) \geq 1 - \frac{\sqrt{\sigma^2 \log d/n}}{d^2}$ *for all* $i$.

*Proof of Theorem 8.* We first prove that $\tilde{\mathbf{v}}^t$ can be expressed by $\tilde{\mathbf{v}}^t = \hat{\mathbf{u}} \log t + \boldsymbol{\rho}^t$ with bounded $\boldsymbol{\rho}^t$. Since $\boldsymbol{\rho}^t = \mathbf{r}^t + \tilde{\mathbf{u}}$ and $\mathbf{r}^t = \tilde{\mathbf{v}}^t - \hat{\mathbf{u}} \log t - \tilde{\mathbf{u}}$, then

$$\|\mathbf{r}^{t+1}\|^2 = \|\mathbf{r}^{t+1} - \mathbf{r}^t\|^2 + 2 (\mathbf{r}^{t+1} - \mathbf{r}^t)^\top \mathbf{r}^t + \|\mathbf{r}^t\|^2. \tag{11}$$

By Lemma 9 (A), we have $\phi'((\tilde{\mathbf{v}}^t)^\top \mathbf{u}^i) < 0$. By the definition $\hat{\mathbf{u}}$, we have $\hat{\mathbf{u}}^\top \mathbf{u}^i \geq 1$. This implies $\hat{\mathbf{u}}^\top$

$$\|\mathbf{r}^{t+1} - \mathbf{r}^t\|^2 = \left\| -\beta \nabla \mathcal{L}(\tilde{\mathbf{v}}^t) - \hat{\mathbf{u}} \left( \log \left( 1 + \frac{1}{t} \right) \right) \right\|^2$$

$$= \beta^2 \|\nabla \mathcal{L}(\tilde{\mathbf{v}}^t)\| + \|\hat{\mathbf{u}}\|^2 \left( \log \left( 1 + \frac{1}{t} \right) \right)^2 + 2\beta \hat{\mathbf{u}}^\top \nabla \mathcal{L}(\tilde{\mathbf{v}}^t) \log \left( 1 + \frac{1}{t} \right) \tag{12}$$

$$\leq \beta^2 \|\nabla \mathcal{L}(\tilde{\mathbf{v}}^t)\| + \frac{1}{t^2} \|\hat{\mathbf{u}}\|^2,$$

By Lemma 9 (B) and Lemma 10, we have

$$\sum_{t=0}^{\infty} \|\nabla \mathcal{L}(\tilde{\mathbf{v}}^t)\|_2^2 = C_0 < \infty, \qquad \lim_{t \to \infty} \|\nabla \mathcal{L}(\tilde{\mathbf{v}}^t)\|_2 = 0. \tag{13}$$

Therefore,

$$\left\|\mathbf{r}^{t+1} - \mathbf{r}^t\right\|^2 = o\left(1\right), \quad \sum_{t=T_1^*}^{\infty} \left\|\mathbf{r}^{t+1} - \mathbf{r}^t\right\|^2 = C_0 < \infty. \tag{14}$$

By Lemma 11, for $t > t_1$, $\left(\mathbf{r}^{t+1} - \mathbf{r}^t\right)^\top \mathbf{r}^t \leq C\, t^{-\min\{\theta, 1+0.5\mu_{\min}\}}$.

Thus,

$$\begin{aligned}
\left\|\mathbf{r}^t\right\|^2 - \left\|\mathbf{r}^{t_1}\right\|^2 &= \sum_{k=k_1}^{t-1} \left\|\mathbf{r}^{k+1} - \mathbf{r}^k\right\|^2 + 2\left(\mathbf{r}^{k+1} - \mathbf{r}^k\right)^\top \mathbf{r}^k \\
&\leq C_0 + 2\sum_{k=k_1}^{t-1} C\, k^{-\min\{\theta, 1+0.5\mu_{\min}\}} \\
&< \infty.
\end{aligned} \tag{15}$$

This implies $\|\mathbf{r}^t\|$ is bounded and further $\|\boldsymbol{\rho}^t\|$ is bounded.

$\square$

In Stage 2, $\tilde{\mathbf{w}}^t \equiv \tilde{\mathbf{w}}^{T_1^*}$ is fixed and the gate updates $\tilde{\mathbf{v}}^t$ follow:

$$\begin{aligned}
\tilde{\mathbf{v}}^{t+1} &= \tilde{\mathbf{v}}^t - \frac{\beta}{n} \sum_{i=1}^{n} \nabla \ell_i(\tilde{\mathbf{v}}^t), \quad \tilde{\mathbf{v}}^{T_1^*} = \mathbf{0}, \\
\nabla \ell_i(\tilde{\mathbf{v}}^t) &:= \left(\left(p_i^t \mathbf{x}^i + \left(1 - p_i^t\right) \mathbf{z}^i\right)^\top \left(\mathbf{w}^t\right)^{\odot 2} - \left(\mathbf{x}^i\right)^\top \left(\mathbf{w}^*\right)^{\odot 2} - \epsilon_i\right) \\
&\quad \cdot \left(\mathbf{x}^i - \mathbf{z}^i\right)^\top \left(\mathbf{w}^{T_1^*}\right)^{\odot 2} p_i^t \left(1 - p_i^t\right) \left(\tilde{\mathbf{x}}^i - \tilde{\mathbf{z}}^i\right),
\end{aligned} \tag{16}$$

where $p_i^t = \sigma\left((\tilde{\mathbf{v}}^t)^\top \left(\tilde{\mathbf{x}}^i - \tilde{\mathbf{z}}^i\right)\right)$, and $\quad \sigma(t) = \frac{1}{1+e^{-t}}$.

$\nabla \ell_i(\tilde{\mathbf{v}}^t)$ can be further expressed as

$$\begin{aligned}
\nabla \ell_i(\tilde{\mathbf{v}}^t) &= \left(p_i^t \left(\left(\mathbf{x}_1^i - \mathbf{z}_1^i\right)\left(\mathbf{w}_1^{T_1^*}\right)^2 + \zeta_1^i\right) - \mathbf{x}_1^i + c_1^i + \zeta_2^i\right) \\
&\quad \cdot \left(\left(\mathbf{x}_1^i - \mathbf{z}_1^i\right)\left(\mathbf{w}_1^{T_1^*}\right)^2 + \zeta_1^i\right) p_i^t \left(1 - p_i^t\right) \left(\tilde{\mathbf{x}}^i - \tilde{\mathbf{z}}^i\right) \\
&= \left(p_i^t \left(\left(\mathbf{x}_1^i - \mathbf{z}_1^i\right)^2 \left(\mathbf{w}_1^{T_1^*}\right)^4 + \zeta_3\right) - \left(\mathbf{x}_1^i - c_1^i\right)\left(\mathbf{x}_1^i - \mathbf{z}_1^i\right)\left(\mathbf{w}_1^{T_1^*}\right)^2 + \zeta_4^i\right) \\
&\quad \cdot p_i^t \left(1 - p_i^t\right) \left(\tilde{\mathbf{x}}^i - \tilde{\mathbf{z}}^i\right),
\end{aligned} \tag{17}$$

where $\zeta_1^i, \zeta_2^i, \zeta_3^i, \zeta_4^i$ are small quantities, and $c_1^i$ is a constant.

$$\zeta_1^i = \sum_{j=2}^{d} \left(\mathbf{x}_j^i - \mathbf{z}_j^i\right)\left(\mathbf{w}_j^{T_1^*}\right)^2, \quad \zeta_2^i = \left(\sum_{j=2}^{d} \left(\mathbf{z}_j^i\right)\left(\mathbf{w}_j^{T_1^*}\right)^2\right),$$

$$\zeta_3^i = 2\left(\mathbf{x}_1^i - \mathbf{z}_1^i\right)\left(\mathbf{w}_1^{T_1^*}\right)^2 \zeta_1^i + (\zeta_1^i)^2, \quad \zeta_4^i = \left(\mathbf{x}_1^i - \mathbf{z}_1^i\right)\left(\mathbf{w}_1^{T_1^*}\right)^2 \zeta_2^i - \left(\mathbf{x}_1^i - c_1^i\right)\zeta_1^i - \zeta_1^i \zeta_2^i,$$

$$c_1^i = \left(\mathbf{z}_1^i \left(\mathbf{w}_1^{T_1^*}\right)^2 - \sum_{j=2}^{d} \left(\mathbf{x}_j^i\right)\left(\mathbf{w}_j^*\right)^2 - \epsilon^i\right).$$

Let $\phi_i(\tilde{\mathbf{v}}^\top \mathbf{u}^i) = \ell_i(\tilde{\mathbf{v}}^t)$ such that

$$\begin{aligned}
\phi_i'(u) &= \left(\sigma(u)\left(\left(\mathbf{x}_1^i - \mathbf{z}_1^i\right)^2 \left(\mathbf{w}_1^{T_1^*}\right)^4 + \zeta_3^i\right) - \left(\mathbf{x}_1^i - c_1^i\right)\left(\mathbf{x}_1^i - \mathbf{z}_1^i\right)\left(\mathbf{w}_1^{T_1^*}\right)^2 + \zeta_4^i\right) \\
&\quad \cdot \sigma(u)\left(1 - \sigma(u)\right).
\end{aligned}$$

Define

$$\mathcal{L}(\tilde{\mathbf{v}}) := \frac{1}{n} \sum_{i=1}^{n} \phi_i(\tilde{\mathbf{v}}^\top \mathbf{u}^i), \quad \mathbf{u}^i := \tilde{\mathbf{x}}^i - \tilde{\mathbf{z}}^i. \tag{18}$$

Then $\nabla\mathcal{L}(\tilde{\mathbf{v}}) = \frac{1}{n}\sum_{i=1}^{n} \nabla\ell_i(\tilde{\mathbf{v}}^t)$ which is equal to the gradient of $\overline{\mathcal{L}}(\tilde{\mathbf{v}}) := \mathcal{L}(\tilde{\mathbf{v}}) - \inf_{\tilde{\mathbf{v}}} \mathcal{L}(\tilde{\mathbf{v}}) \geq 0$. Without loss of generality, we assume $\inf_{\tilde{\mathbf{v}}} \mathcal{L}(\tilde{\mathbf{v}}) = 0$.

Since $\mathbf{w}_1^{T_1^*} \in [1/4, 1/2]$ and $|\mathbf{w}_j^{T_1^*}| \leq \frac{\sqrt{\sigma^2 \log d/n}}{d^2}$ for all $j > 1$. For each sample $i$, define and $|\mathbf{x}_j^i| \leq B_{\mathbf{x}}$, $|\mathbf{z}_j^i| \leq B_{\mathbf{z}}$, $|\epsilon^i| \leq B_\epsilon$ a.s. Then

$$|\zeta_1^i| \leq \frac{B_{\mathbf{x}} + B_{\mathbf{z}}}{d}, \qquad |\zeta_2^i| \leq \frac{B_{\mathbf{z}}}{d}, \qquad |\zeta_3^i| \leq \frac{C_3}{d}, \qquad |\zeta_4^i| \leq \frac{C_4}{d},$$

for explicit constants $C_3, C_4$ depending only on $(B_{\mathbf{x}}, B_{\mathbf{z}}, B_\epsilon)$.

Let

$$A_i^{(0)} := (\mathbf{x}_1^i - \mathbf{z}_1^i)^2 (\mathbf{w}_1^{T_1^*})^4, \qquad B_{0,i} := (\mathbf{x}_1^i - c_1^i)(\mathbf{x}_1^i - \mathbf{z}_1^i)(\mathbf{w}_1^{T_1^*})^2,$$

and absorb the small remainders into

$$\zeta_5^i(u) := \sigma(u)\,\zeta_3^i + \zeta_4^i, \qquad |\zeta_5^i(u)| \leq \frac{C_3 + C_4}{d} \quad \text{for all } u \text{ (since } 0 \leq \sigma(u) \leq 1).$$

Then the scalar driving term becomes

$$\phi_i'(u) = \big(\sigma(u)\,A_i^{(0)} - B_{0,i} + \zeta_5^i(u)\big)\,\sigma(u)\big(1 - \sigma(u)\big), \qquad \sigma(u) = \frac{1}{1 + e^{-u}}.$$

**Lemma 9** (Properties of $\phi_i(u)$). *$\phi_i(u)$ has the following properties:*

- *(A) **Monotonicity.** For all $i$ and $u$, $\phi_i'(u) < 0$.*

- *(B) **Second-derivative control.** For all $u$, $|\phi_i''(u)| \leq C_{\phi''}$, hence*

$$\|\nabla^2\mathcal{L}(\tilde{\mathbf{v}})\|_{\mathrm{op}} \leq \frac{C_{\phi''}}{n}\,\sigma_{\max}(\mathbf{U})^2, \qquad \mathbf{U} := [\mathbf{u}^1, \ldots, \mathbf{u}^n],$$

*where*

$$C_{\phi''} := \frac{1}{4}\left(\frac{5(B_{\mathbf{x}} + B_{\mathbf{z}})^2}{64} + \frac{1}{4}\Big(B_{\mathbf{x}} + \frac{1}{4}\Big(B_{\mathbf{x}} + \frac{B_{\mathbf{z}}}{4} + r\,B_{\mathbf{x}} + B_\epsilon\Big)(B_{\mathbf{x}} + B_{\mathbf{z}}) + 1\Big)\right).$$

- *(C) **Exponential tails.** Let*

$$f_i(u) := -\phi_i'(u) = \big(B_{0,i} - \sigma(u)A_i^{(0)} - \zeta_5^i(u)\big)\,\sigma(u)\big(1 - \sigma(u)\big),$$

*there exist $C_i > 0$, $\mu_i > 0$ and $u_{0,i}$ such that, for all $u > u_{0,i}$,*

$$C_i\big(1 - e^{-\mu_i u}\big)e^{-u} \leq f_i(u) \leq C_i\big(1 + e^{-\mu_i u}\big)e^{-u}.$$

*Proof of Lemma 9.* (A) Note that

$$B_{0,i} - A_i^{(0)} = (\mathbf{x}_1^i - c_1^i)(\mathbf{x}_1^i - \mathbf{z}_1^i)(\mathbf{w}_1^{T_1^*})^2 - (\mathbf{x}_1^i - \mathbf{z}_1^i)^2(\mathbf{w}_1^{T_1^*})^4$$

$$= (\mathbf{x}_1^i - \mathbf{z}_1^i)(\mathbf{w}_1^{T_1^*})^2\left(\Big(1 - (\mathbf{w}_1^{T_1^*})^2\Big)\mathbf{x}_1^i + \sum_{j=2}^{d}(\mathbf{x}_j^i)(\mathbf{w}_j^*)^2 + \epsilon^i\right).$$

Since $\mathbf{w}_1^{T_1^*} \in [1/4, 1/2]$, we have $1 - (\mathbf{w}_1^{T_1^*})^2 \geq 1 - (1/2)^2 = 3/4$. By Condition 2 (iii),

$$\Big|\Big(1 - (\mathbf{w}_1^{T_1^*})^2\Big)\mathbf{x}_1^i + \sum_{j=2}^{d}(\mathbf{x}_j^i)(\mathbf{w}_j^*)^2 + \epsilon^i\Big| \geq \frac{3}{4}|\mathbf{x}_1^i| - \Big|\sum_{j=2}^{d}(\mathbf{x}_j^i)(\mathbf{w}_j^*)^2\Big| - |\epsilon^i|$$

$$\geq \frac{3}{4}|\mathbf{x}_1^i| - ((r-1)B_{\mathbf{x}}' + B_\epsilon) \geq \tau_2.$$

Moreover, Condition 2 (iii) also implies

$$\text{sgn}\Big(\Big(1 - (\mathbf{w}_1^{T_1^*})^2\Big)\,\mathbf{x}_1^i + \sum_{j=2}^{d}\big(\mathbf{x}_j^i\big)\big(\mathbf{w}_j^*\big)^2 + \epsilon^i\Big) = \text{sgn}(\mathbf{x}_1^i).$$

By Condition 2 (i) and Condition 2 (ii),

$$\Big|(\mathbf{x}_1^i - \mathbf{z}_1^i)\left(\Big(1 - (\mathbf{w}_1^{T_1^*})^2\Big)\,\mathbf{x}_1^i + \sum_{j=2}^{d}\big(\mathbf{x}_j^i\big)\big(\mathbf{w}_j^*\big)^2 + \epsilon^i\right)\Big| \geq \tau_1\,\tau_2,$$

$$\text{sgn}\Big((\mathbf{x}_1^i - \mathbf{z}_1^i)\left(\Big(1 - (\mathbf{w}_1^{T_1^*})^2\Big)\,\mathbf{x}_1^i + \sum_{j=2}^{d}\big(\mathbf{x}_j^i\big)\big(\mathbf{w}_j^*\big)^2 + \epsilon^i\right)\Big) = +1.$$

Therefore,

$$B_{0,i} - A_i^{(0)} \geq \frac{1}{16}\,\tau_1\,\tau_2.$$

When $d$ is large enough,

$$\sigma(u)\,A_i^{(0)} - B_{0,i} + \zeta_5^i(u) < \frac{\tau_1\,\tau_2}{32} \tag{19}$$

Therefore, for all $u \in \mathbb{R}$,

$$\phi_i'(u) = \big(\sigma(u)A_i^{(0)} - B_{0,i} + \zeta_5^i(u)\big)\cdot\sigma(u)\big(1 - \sigma(u)\big) < 0.$$

(B) Let $s := \sigma(u)$, $g(u) := s(1-s) \in (0, 1/4]$. Define $h(u) := sA_i^{(0)} - B_{0,i} + \zeta_5^i(u)$. Noting $\zeta_5^{i\,\prime}(u) = \sigma'(u)\,\zeta_3^i = g(u)\,\zeta_3^i$, we have

$$\phi_i''(u) = h'(u)\,g(u) + h(u)\,g'(u) = \big(A_i^{(0)} + \zeta_3^i\big)\,g(u)^2 + h(u)\,(1 - 2s)\,g(u).$$

Hence

$$|\phi_i''(u)| \leq \frac{1}{4}\left(\frac{|A_i^{(0)} + \zeta_3^i|}{4} + |A_i^{(0)} + \zeta_3^i| + |B_{0,i}| + |\zeta_5^i(u)|\right) \leq C_{\phi''},$$

where, using $|\mathbf{x}_1^i - \mathbf{z}_1^i| \leq B_\mathbf{x} + B_\mathbf{z}$ and $\mathbf{w}_1^{T_1^*} \in [1/4, 1/2]$,

$$|A_i^{(0)} + \zeta_3^i| \leq \frac{(B_\mathbf{x} + B_\mathbf{z})^2}{16} + \frac{C_3}{d}, \qquad |B_{0,i}| \leq \frac{1}{4}\Big(B_\mathbf{x} + \frac{B_\mathbf{z}}{4} + r\,B_\mathbf{x} + B_\epsilon\Big)(B_\mathbf{x} + B_\mathbf{z}),$$

and thus one can take the explicit bound

$$C_{\phi''} := \frac{1}{4}\left(\frac{5(B_\mathbf{x} + B_\mathbf{z})^2}{64} + \frac{1}{4}\Big(B_\mathbf{x} + \frac{1}{4}\Big(B_\mathbf{x} + \frac{B_\mathbf{z}}{4} + r\,B_\mathbf{x} + B_\epsilon\Big)(B_\mathbf{x} + B_\mathbf{z}) + \frac{C_3 + C_4}{d}\right).$$

Consequently, for the Phase-II scalar objective

$$\mathcal{L}(\tilde{\mathbf{v}}) = \frac{1}{n}\sum_{i=1}^{n}\ell_i(\tilde{\mathbf{v}}) = \frac{1}{n}\sum_{i=1}^{n}\phi_i\big(\tilde{\mathbf{v}}^\top\mathbf{u}^i\big), \qquad \mathbf{u}^i := \tilde{\mathbf{x}}^i - \tilde{\mathbf{z}}^i.$$

Let $\mathbf{U} := \big[\mathbf{u}^1,\ \mathbf{u}^2,\ \ldots,\ \mathbf{u}^n\big] \in \mathbb{R}^{(M+d)\times n}$.

$$\nabla^2\mathcal{L}(\tilde{\mathbf{v}}) = \frac{1}{n}\sum_{i=1}^{n}\phi_i''(\tilde{\mathbf{v}}^\top\mathbf{u}^i)\,\mathbf{u}^i(\mathbf{u}^i)^\top = \frac{1}{n}\,\mathbf{U}\,\mathbf{D}(\tilde{\mathbf{v}})\,\mathbf{U}^\top,$$

$$\mathbf{D}(\tilde{\mathbf{v}}) := \text{diag}\Big(\phi_1''(\tilde{\mathbf{v}}^\top\mathbf{u}^1), \ldots, \phi_n''(\tilde{\mathbf{v}}^\top\mathbf{u}^n)\Big).$$

$$\big\|\nabla^2\mathcal{L}(\tilde{\mathbf{v}})\big\|_{\text{op}} \leq \frac{1}{n}\,\|\mathbf{U}\|_{\text{op}}^2\,\|\mathbf{D}(\tilde{\mathbf{v}})\|_{\text{op}} \leq \frac{C_{\phi''}}{n}\,\sigma_{\max}(\mathbf{U})^2.$$

(C) Let

$$f_i(u) := -\phi_i'(u) = \left(B_{0,i} - \sigma(u)A_i^{(0)} - \zeta_5^i(u)\right)\sigma(u)\left(1 - \sigma(u)\right), \qquad \sigma(u) = \frac{1}{1 + e^{-u}}.$$

Set

$$C_i := B_{0,i} - A_i^{(0)} - \zeta_3^i - \zeta_4^i, \qquad D_i := A_i^{(0)} + \zeta_3^i.$$

Assuming the dominance condition ensures $C_i > 0$ and, for $d$ large, $D_i \geq 0$ (since $A_i^{(0)} > 0$ and $|\zeta_3^i| = O(1/d)$).

Expanding at $e^{-u} \to 0$,

$$f_i(u) = C_i e^{-u} + \underbrace{(D_i - 2C_i)}_{=:a_i} e^{-2u} + \underbrace{(3C_i - 3D_i)}_{=:b_i} e^{-3u} + O(e^{-4u}).$$

Factor $C_i e^{-u}$:

$$f_i(u) = C_i e^{-u}\Big(1 + \underbrace{\tfrac{a_i}{C_i} e^{-u} + \tfrac{b_i}{C_i} e^{-2u} + O(e^{-3u})}_{\text{small for large } u}\Big).$$

Choosing $u > u_{0,i}$ and $\mu_i > 0$ such that $\frac{|a_i|}{C_i}e^{-u} + \frac{|b_i|}{C_i}e^{-2u} \leq e^{-\mu_i u}$ absorbs all higher orders into the single factor $(1 \pm e^{-\mu_i u})$. Then, we have for all $u > u_{0,i}$,

$$C_i\left(1 - e^{-\mu_i u}\right)e^{-u} \leq f_i(u) \leq C_i\left(1 + e^{-\mu_i u}\right)e^{-u}.$$

$\square$

Denote $C_i \exp\left(-\tilde{\mathbf{u}}^\top \mathbf{u}_i\right) = \alpha_i$. Let $\mathbf{r}^t = \tilde{\mathbf{v}}^t - \hat{\mathbf{u}}\log t - \tilde{\mathbf{u}}$, then $\rho^t = \mathbf{r}^t + \tilde{\mathbf{u}}$.

**Lemma 10** ( Soudry et al. (2018, Lemma 10)). *Consider GD updates* $\tilde{\mathbf{v}}^{t+1} = \tilde{\mathbf{v}}^t - \beta\nabla\mathcal{L}(\tilde{\mathbf{v}}^t)$. *If* $\beta < \frac{n}{C_{\phi''}\,\sigma_{\max}(\mathbf{U})^2}$, *then the GD sequence satisfies*

$$\sum_{t=0}^{\infty}\left\|\nabla\mathcal{L}(\tilde{\mathbf{v}}^t)\right\|_2^2 < \infty, \qquad \lim_{t\to\infty}\left\|\nabla\mathcal{L}(\tilde{\mathbf{v}}^t)\right\|_2 = 0.$$

**Lemma 11.** *Define* $\mu_{\min} = \min_{1\leq i\leq n}\mu_i$ *There exists constant $C$ and $t_1$, such that for $t > t_1$,* $\left(\mathbf{r}^{t+1} - \mathbf{r}^t\right)^\top\mathbf{r}^t \leq C\,t^{-\min\{\theta, 1+0.5\mu_{\min}\}}$, $\theta = \arg\min_{i\notin\mathcal{S}}\mathbf{u}^\top\mathbf{u}^i > 1$.

*Proof of Lemma 11.* Since under our setting each sample's scalar loss $\phi_i(u)$ satisfies that $-\phi_i'(u)$ has an exponential tail, Soudry et al. (2018, Lemma 11) yields this lemma.

$$\begin{aligned}
\left(\mathbf{r}^{t+1} - \mathbf{r}^t\right)^\top\mathbf{r}^t &= \left(-\beta\nabla\mathcal{L}(\tilde{\mathbf{v}}^t) - \hat{\mathbf{u}}\left(\log\left(1 + \frac{1}{t}\right)\right)\right)^\top\mathbf{r}^t \\
&= \underbrace{\hat{\mathbf{u}}^\top\mathbf{r}^t\left[\frac{1}{t} - \log\left(1 + \frac{1}{t}\right)\right]}_{(A_1)} - \underbrace{\frac{\beta}{n}\sum_{i\notin\mathcal{S}}\phi_i'\left((\tilde{\mathbf{v}}^t)^\top\mathbf{u}^i\right)(\mathbf{u}^i)^\top\mathbf{r}^t}_{A_2} \\
&\quad - \underbrace{\frac{\beta}{n}\sum_{i\in\mathcal{S}}\left[\frac{1}{t}C_i\exp(-\tilde{\mathbf{u}}^\top\mathbf{u}^i) + \phi_i'\left((\tilde{\mathbf{v}}^t)^\top\mathbf{u}^i\right)\right](\mathbf{u}^i)^\top\mathbf{r}^t}_{A_3}
\end{aligned} \tag{20}$$

Denote by $\mathbf{X}_\mathcal{S}$ the matrix whose columns are the support vectors, and let $\mathbf{P}$ be the orthogonal projection onto the subspace spanned by these support vectors. Then $\mathbf{P}\hat{\mathbf{u}} = \hat{\mathbf{u}}$.

For $A_1$, firstly, the following shows that $\hat{\mathbf{u}}^\top\mathbf{r}^t = o(t)$. Lemma 10 shows that $\lim_{t\to\infty}\left\|\nabla\mathcal{L}(\tilde{\mathbf{v}}^t)\right\|_2 = 0$.

$$\begin{aligned}
\hat{\mathbf{u}}^\top\mathbf{r}^t &= \hat{\mathbf{u}}^\top\left(\tilde{\mathbf{v}}^{T_1^*} - \beta\sum_{s=T_1^*}^{t}\nabla\mathcal{L}(\tilde{\mathbf{v}}^s) - \hat{\mathbf{u}}\log t - \tilde{\mathbf{u}}\right) \\
&\leq \hat{\mathbf{u}}^\top\left(\tilde{\mathbf{v}}^{T_1^*} - \hat{\mathbf{u}}\log t - \tilde{\mathbf{u}}\right) - \beta t\min_{T_1^*\leq s\leq t}\hat{\mathbf{u}}^\top\nabla\mathcal{L}(\tilde{\mathbf{v}}^s) = o(t)
\end{aligned} \tag{21}$$

Then $A_1$ can be bounded from above by:

$$\hat{\mathbf{u}}^\top \mathbf{r}^t \left[ \frac{1}{t} - \log\left(1 + \frac{1}{t}\right) \right] \leq \max\left[ \hat{\mathbf{u}}^\top \mathbf{r}^t, 0 \right] \left[ \frac{1}{t} - \log\left(1 + \frac{1}{t}\right) \right]$$

$$\leq \max\left[ \hat{\mathbf{u}}^\top \mathbf{P} \mathbf{r}^t, 0 \right] t^{-2} \tag{22}$$

$$\leq \begin{cases} \|\| \hat{\mathbf{u}} \|\| \epsilon_1 t^{-2}, & \text{if } \|\mathbf{P}\mathbf{r}^t\| \leq \epsilon_1 \\ o(t^{-1}), & \text{if } \|\mathbf{P}\mathbf{r}^t\| > \epsilon_1 \end{cases}$$

By Lemma 9, we have $\phi_i'(u) < 0$. Then,

$$\hat{\mathbf{u}}^\top \nabla \mathcal{L}(\tilde{\mathbf{v}}) = \frac{1}{n} \sum_{i=1}^n \phi_i(\hat{\mathbf{u}}^\top \tilde{\mathbf{v}}) \hat{\mathbf{u}}^\top \tilde{\mathbf{v}} < 0. \tag{23}$$

This implies that $\mathcal{L}(\tilde{\mathbf{v}})$ does not have finite critical points $\tilde{\mathbf{v}}$. With $\lim_{t\to\infty} \left\| \nabla \mathcal{L}(\tilde{\mathbf{v}}^t) \right\|_2 = 0$, we have $\lim_{t\to+\infty} (\tilde{\mathbf{v}}^t)^\top \mathbf{u}^i = +\infty$. By Lemma 9, there exists $t_1 > 0$ such that for $t > t_1$ and $i \in [n]$,

$$C_i\big(1 - e^{-\mu_i(\tilde{\mathbf{v}}^t)^\top \mathbf{u}^i}\big) e^{-(\tilde{\mathbf{v}}^t)^\top \mathbf{u}^i} \leq -\phi_i'((\tilde{\mathbf{v}}^t)^\top \mathbf{u}^i) \leq C_i\big(1 + e^{-\mu_i(\tilde{\mathbf{v}}^t)^\top \mathbf{u}^i}\big) e^{-(\tilde{\mathbf{v}}^t)^\top \mathbf{u}^i}. \tag{24}$$

For $A_2$, when $t > t_1$, $\left| \phi_i'((\tilde{\mathbf{v}}^t)^\top \mathbf{u}^i) \right| \leq 2 \exp(-(\tilde{\mathbf{v}}^t)^\top \mathbf{u}^i)$.

$$-\frac{\beta}{n} \sum_{i \notin \mathcal{S}} \phi_i'\big( (\tilde{\mathbf{v}}^t)^\top \mathbf{u}^i \big) (\mathbf{u}^i)^\top \mathbf{r}^t \leq -\frac{\beta}{n} \sum_{i \notin \mathcal{S}: (\mathbf{u}^i)^\top \mathbf{r}^t \geq 0} \phi_i'\big( (\tilde{\mathbf{v}}^t)^\top \mathbf{u}^i \big)(\mathbf{u}^i)^\top \mathbf{r}^t$$

$$\leq \frac{\beta}{n} \sum_{i \notin \mathcal{S}: (\mathbf{u}^i)^\top \mathbf{r}^t \geq 0} 2 \exp(-(\mathbf{u}^i)^\top \mathbf{r}^t)(\mathbf{u}^i)^\top \mathbf{r}^t$$

$$= \frac{2\beta}{n} \sum_{i \notin \mathcal{S}: (\mathbf{u}^i)^\top \mathbf{r}^t \geq 0} t^{-(\mathbf{u}^i)^\top \hat{\mathbf{u}}} \exp\big(-\tilde{\mathbf{u}}^\top \mathbf{u}^i - (\mathbf{u}^i)^\top \mathbf{r}^t\big)(\mathbf{u}^i)^\top \mathbf{r}^t$$

$$\leq \frac{2\beta}{n} \sum_{i \notin \mathcal{S}: (\mathbf{u}^i)^\top \mathbf{r}^t \geq 0} t^{-(\mathbf{u}^i)^\top \hat{\mathbf{u}}} \exp\big(-\tilde{\mathbf{u}}^\top \mathbf{u}^i\big)$$

$$\leq 2\beta \exp\left( -\min_{1 \leq i \leq n} \tilde{\mathbf{u}}^\top \mathbf{u}^i \right) t^{-\theta}. \tag{25}$$

For $A_3$, the proof is divided into two cases $(\mathbf{u}^i)^\top \mathbf{r}^t \geq 0$ and $(\mathbf{u}^i)^\top \mathbf{r}^t < 0$.

If $(\mathbf{u}^i)^\top \mathbf{r}^t \geq 0$, for $t > t_1$, by equation 24,

$$-\frac{\beta}{n} \left[ \frac{1}{t} C_i \exp(-\tilde{\mathbf{u}}^\top \mathbf{u}^i) + \phi_i'\big( (\tilde{\mathbf{v}}^t)^\top \mathbf{u}^i \big) \right] (\mathbf{u}^i)^\top \mathbf{r}^t$$

$$\leq \frac{\beta C_i}{n} t^{-1} \exp\big(-\tilde{\mathbf{u}}^\top \mathbf{u}^i\big) \left[ \big(1 + t^{-\mu_i} \exp\big(-\mu_i \tilde{\mathbf{u}}^\top \mathbf{u}^i\big)\big) \exp\big(-(\mathbf{u}^i)^\top \mathbf{r}^t\big) - 1 \right] (\mathbf{u}^i)^\top \mathbf{r}^t. \tag{26}$$

(I) If $0 \leq (\mathbf{u}^i)^\top \mathbf{r}^t \leq t^{-0.5\mu_i}$,

$$-\frac{\beta}{n} \left[ \frac{1}{t} C_i \exp(-\tilde{\mathbf{u}}^\top \mathbf{u}^i) + \phi_i'\big( (\tilde{\mathbf{v}}^t)^\top \mathbf{u}^i \big) \right] (\mathbf{u}^i)^\top \mathbf{r}^t$$

$$\leq \frac{\beta C_i}{n} \exp\left( -(1 + \mu_i) \min_{1 \leq i \leq n} \tilde{\mathbf{u}}^\top \mathbf{u}^i \right) t^{-1 - 0.5\mu_i} \tag{27}$$

$$\leq \frac{\beta C_i}{n} \exp\left( -(1 + \mu_i) \min_{1 \leq i \leq n} \tilde{\mathbf{u}}^\top \mathbf{u}^i \right) t^{-1 - 0.5\mu_{\min}}.$$

(II) If $(\mathbf{u}^i)^\top \mathbf{r}^t > t^{-0.5\mu_i}$, we have

$$
\begin{aligned}
&-\frac{\beta}{n}\left[\frac{1}{t}C_i\exp(-\tilde{\mathbf{u}}^\top\mathbf{u}^i)+\phi_i^{'}\big((\tilde{\mathbf{v}}^t)^\top\mathbf{u}^i\big)\right](\mathbf{u}^i)^\top\mathbf{r}^t\\
\leq&\frac{\beta C_i}{n}t^{-1}\exp\left(-\tilde{\mathbf{u}}^\top\mathbf{u}^i\right)\left[\left(1+t^{-\mu_i}\exp\left(-\mu_i\tilde{\mathbf{u}}^\top\mathbf{u}^i\right)\right)\exp\left(-t^{-0.5\mu_i}\right)-1\right](\mathbf{u}^i)^\top\mathbf{r}^t\\
\leq&\frac{\beta C_i}{n}t^{-1}\exp\left(-\tilde{\mathbf{u}}^\top\mathbf{u}^i\right)\left[\left(1+t^{-\mu_i}\exp\left(-\mu_i\tilde{\mathbf{u}}^\top\mathbf{u}^i\right)\right)\left(1-t^{-0.5\mu_i}+t^{\mu_i}\right)-1\right](\mathbf{u}^i)^\top\mathbf{r}^t\\
\leq&\frac{\beta C_i}{n}t^{-1}\exp\left(-\tilde{\mathbf{u}}^\top\mathbf{u}^i\right)\left[t^{-\mu_i}\exp\left(-\mu_i\tilde{\mathbf{u}}^\top\mathbf{u}^i\right)\left(1-t^{-0.5\mu_i}+t^{\mu_i}\right)-t^{-0.5\mu_i}+t^{\mu_i}\right](\mathbf{u}^i)^\top\mathbf{r}^t\\
\leq&\ 0.\ (t>t_2^i)
\end{aligned}
\tag{28}
$$

(III) If $(\mathbf{u}^i)^\top \mathbf{r}^t > \epsilon_2$, consider $t_3^i > t_2^i$ such that $t_3^i > \exp\left(\min_{1\leq j\leq n}\tilde{\mathbf{u}}^\top\mathbf{u}^j\right)\left(e^{0.5\epsilon_2}-1\right)^{-\frac{1}{\mu_i}}$. Then we have

$$
\begin{aligned}
&-\frac{\beta}{n}\left[\frac{1}{t}C_i\exp(-\tilde{\mathbf{u}}^\top\mathbf{u}^i)+\phi_i^{'}\big((\tilde{\mathbf{v}}^t)^\top\mathbf{u}^i\big)\right](\mathbf{u}^i)^\top\mathbf{r}^t\\
\leq&\frac{\beta C_i}{n}\exp(-\max_{1\leq j\leq n}\tilde{\mathbf{u}}^\top\mathbf{u}^j)\left(1-e^{-0.5\epsilon_2}\right)\epsilon_2 t^{-1}.
\end{aligned}
\tag{29}
$$

If $(\mathbf{u}^i)^\top \mathbf{r}^t < 0$, still consider three case. (I) If $-t^{-0.5\mu_i} \leq (\mathbf{u}^i)^\top \mathbf{r}^t < 0$, since $-\phi_i^{'} > 0$, we have

$$
\begin{aligned}
-\frac{\beta}{n}\left[\frac{1}{t}C_i\exp(-\tilde{\mathbf{u}}^\top\mathbf{u}^i)+\phi_i^{'}\big((\tilde{\mathbf{v}}^t)^\top\mathbf{u}^i\big)\right](\mathbf{u}^i)^\top\mathbf{r}^t &\leq \frac{\beta C_i}{n}\exp(-\tilde{\mathbf{u}}^\top\mathbf{u}^i)\left|(\mathbf{u}^i)^\top\mathbf{r}^t\right|\\
&\leq \frac{\beta C_i}{n}\exp(-\max_{1\leq j\leq n}\tilde{\mathbf{u}}^\top\mathbf{u}^n)t^{-1-0.5\mu_i}\\
&\leq \frac{\beta C_i}{n}\exp(-\max_{1\leq j\leq n}\tilde{\mathbf{u}}^\top\mathbf{u}^n)t^{-1-0.5\mu_{\min}}.
\end{aligned}
\tag{30}
$$

(II) If $(\mathbf{u}^i)^\top \mathbf{r}^t < -t^{-0.5\mu_i}$, for $t > t_1$,

$$
\begin{aligned}
&-\frac{\beta}{n}\left[\frac{1}{t}C_i\exp(-\tilde{\mathbf{u}}^\top\mathbf{u}^i)+\phi_i^{'}\big((\tilde{\mathbf{v}}^t)^\top\mathbf{u}^i\big)\right](\mathbf{u}^i)^\top\mathbf{r}^t\\
\leq&\frac{\beta C_i}{n}t^{-1}e^{-\tilde{\mathbf{u}}^\top\mathbf{u}^i}\left[1-e^{-(\mathbf{u}^i)^\top\mathbf{r}^t}\left(1-\left(t^{-1}e^{-\tilde{\mathbf{u}}^\top\mathbf{u}^i}e^{-(\mathbf{u}^i)^\top\mathbf{r}^t}\right)^{\mu_i}\right)\right]\left|(\mathbf{u}^i)^\top\mathbf{r}^t\right|
\end{aligned}
\tag{31}
$$

We then show that there exists $t_4^i > t_3^i$ such that for $t > t_4^i$ the right hand of the above inequality is negative. Since $\left(t^{-1}e^{-\tilde{\mathbf{u}}^\top\mathbf{u}^i}e^{-(\mathbf{u}^i)^\top\mathbf{r}^t}\right)^{\mu_i} = \exp\left(-\mu_i(\tilde{\mathbf{v}}^t)^\top\mathbf{u}^i\right) \to 0$, there exists $t_5^i > t_3^i$ such that for $t > t_5^i$, $\left(t^{-1}e^{-\tilde{\mathbf{u}}^\top\mathbf{u}^i}e^{-(\mathbf{u}^i)^\top\mathbf{r}^t}\right)^{\mu_i} < \frac{1}{2}$. If $e^{-(\mathbf{u}^i)^\top\mathbf{r}^t} \geq 3$, then

$$
e^{-(\mathbf{u}^i)^\top\mathbf{r}^t}\left(1-\left(t^{-1}e^{-\tilde{\mathbf{u}}^\top\mathbf{u}^i}e^{-(\mathbf{u}^i)^\top\mathbf{r}^t}\right)^{\mu_i}\right) \geq 1.5 > 1.
\tag{32}
$$

If $e^{-(\mathbf{u}^i)^\top\mathbf{r}^t} < 3$, then

$$
\begin{aligned}
&e^{-(\mathbf{u}^i)^\top\mathbf{r}^t}\left(1-\left(t^{-1}e^{-\tilde{\mathbf{u}}^\top\mathbf{u}^i}e^{-(\mathbf{u}^i)^\top\mathbf{r}^t}\right)^{\mu_i}\right)\\
>&e^{-(\mathbf{u}^i)^\top\mathbf{r}^t}\left(1-\left(3t^{-1}e^{-\tilde{\mathbf{u}}^\top\mathbf{u}^i}\right)^{\mu_i}\right)\\
\geq&\left(1+t^{-0.5\mu_i}\right)\left(1-t^{-\mu_i}\left(3e^{-\tilde{\mathbf{u}}^\top\mathbf{u}^i}\right)^{\mu_i}\right)\\
\geq&1+t^{-0.5\mu_i}-t^{-\mu_i}\left(3e^{-\tilde{\mathbf{u}}^\top\mathbf{u}^i}\right)^{\mu_i}-t^{-1.5\mu_i}\left(3e^{-\tilde{\mathbf{u}}^\top\mathbf{u}^i}\right)^{\mu_i}.
\end{aligned}
\tag{33}
$$

By taking $t_4^i > t_3^i$ such that for $t > t_4^i$, equation 32 and equation 33 larger than 1, we can obtain that

$$
-\frac{\beta}{n}\left[\frac{1}{t}C_i\exp(-\tilde{\mathbf{u}}^\top\mathbf{u}^i)+\phi_i^{'}\big((\tilde{\mathbf{v}}^t)^\top\mathbf{u}^i\big)\right](\mathbf{u}^i)^\top\mathbf{r}^t < 0.
\tag{34}
$$

(III) If $(\mathbf{u}^i)^\top \mathbf{r}^t < -t^{-0.5\mu_i}$, there exists $t_5^i > t_4^i$ such that for $t > t_5^i$,

$$e^{-(\mathbf{u}^i)^\top \mathbf{r}^t} \left(1 - \left(t^{-1} e^{-\tilde{\mathbf{u}}^\top \mathbf{u}^i} e^{-(\mathbf{u}^i)^\top \mathbf{r}^t}\right)^{\mu_i}\right) \geq 1.5. \tag{35}$$

Then there exists constant $c_1$ such that

$$-\frac{\beta}{n} \left[\frac{1}{t} C_i \exp(-\tilde{\mathbf{u}}^\top \mathbf{u}^i) + \phi_i'\left((\tilde{\mathbf{v}}^t)^\top \mathbf{u}^i\right)\right] (\mathbf{u}^i)^\top \mathbf{r}^t < -c_1 t^{-1}. \tag{36}$$

FInally, consider $t > t_2 := \max_{1 \leq i \leq n} t_5^i$: If $\|\mathbf{P}\mathbf{r}^t\| > \epsilon_1$, then

$$\max_{i \in \mathcal{S}} \left|(\mathbf{u}^i)^\top \mathbf{r}^t\right|^2 \geq \frac{1}{|\mathcal{S}|} \sum_{i \in \mathcal{S}} \left|(\mathbf{u}^i)^\top \mathbf{P}\mathbf{r}^t\right|^2 = \frac{1}{|\mathcal{S}|} \left\|\mathbf{X}_\mathcal{S}^\top \mathbf{P}\mathbf{r}^t\right\|^2 \geq \frac{1}{|\mathcal{S}|} \sigma_{\min}^2\left(\mathbf{X}_\mathcal{S}\right) \epsilon_1^2. \tag{37}$$

Let $\epsilon_2 = \sqrt{|\mathcal{S}|^{-1} \sigma_{\min}^2(\mathbf{X}_\mathcal{S})} \epsilon_1$, then there exists $i \in \mathcal{S}$ such that $\left|(\mathbf{u}^i)^\top \mathbf{r}^t\right| \geq \epsilon_2$. By equation 29, equation 36 and equation 22, we can obtain that there exists $c_2 > 0$, such that for $t > t_2$,

$$\left(\mathbf{r}^{t+1} - \mathbf{r}^t\right)^\top \mathbf{r}^t \leq -c_2 t^{-1} + o(t^{-1}). \tag{38}$$

Then there exists $t_3 > t_2$ such that for $t > t_3$,

$$\left(\mathbf{r}^{t+1} - \mathbf{r}^t\right)^\top \mathbf{r}^t \leq 0 \leq C\, t^{-\min\{\theta, 1+0.5\mu_{\min}\}}. \tag{39}$$

If $\|\mathbf{P}\mathbf{r}^t\| > \epsilon_1$, then by equation 27, equation 28, equation 30, and equation 31, there exists $c_3 > 0$ such that for $t > t_2$, we have

$$\left(\mathbf{r}^{t+1} - \mathbf{r}^t\right)^\top \mathbf{r}^t \leq c_3\, t^{-\min\{\theta, 1+0.5\mu_{\min}\}}. \tag{40}$$

Taking $C = c_3$ and $t_1 = t_4$, we obtain that for $t > t_1$, $\left(\mathbf{r}^{t+1} - \mathbf{r}^t\right)^\top \mathbf{r}^t \leq C\, t^{-\min\{\theta, 1+0.5\mu_{\min}\}}$.

$\square$

### A.3.3 STAGE 3

After Stage 2, we have with probablity at least $1 - \mathcal{O}(\frac{1}{d^2})$, $\tilde{\mathbf{w}}^{T_2^*} = \begin{bmatrix} \mathbf{0} \\ \mathbf{w}^{T_2^*} \end{bmatrix}$ satisfies $\frac{1}{4} \leq \mathbf{w}_1^{T_2^*} \leq \frac{1}{2}$, for $j \in S \setminus \{1\}$, $\frac{\sqrt{\sigma^2 \log d/n}}{d^3} \leq \mathbf{w}_j^{T_2^*} \leq \frac{\sqrt{\sigma^2 \log d/n}}{d^2}$, and for $j \in S^c$, $\mathbf{w}_j^{T_2^*} \leq \frac{\sqrt{\sigma^2 \log d/n}}{d^2}$. Moreover, the gate iterate $\tilde{\mathbf{v}}^t$ satisfies $p_i^{T_1^* + T_2^*} = \sigma((\tilde{\mathbf{v}}^{T_1^* + T_2^*})^\top \mathbf{u}^i) \geq 1 - \frac{\sqrt{\sigma^2 \log d/n}}{d^2}$ for all $i$. We now show that, in Stage 3 with $T_3^* \asymp \frac{1}{\eta} \log\left(\frac{n}{\sigma^2 \log(dr)}\right)$, and $T^* = T_1^* + T_2^* + T_3^*$, GD yields $\tilde{\mathbf{w}}^{T^*} = \begin{bmatrix} \mathbf{0} \\ \mathbf{w}^{T^*} \end{bmatrix}$, $\left\|\mathbf{w}_S^{T^*} - 1\right\|_\infty \lesssim \sqrt{\frac{\sigma^2 \log d}{n}}$, $\left\|\mathbf{w}_{S^c}^{T^*}\right\|_\infty \lesssim \frac{\sqrt{\sigma^2 \log d/n}}{d}$. Since Stage 3 freezes the gate, $\tilde{\mathbf{v}}^{T^*}$ remains at its Stage 2 form: $\tilde{\mathbf{v}}^{T^*} = \hat{\mathbf{u}} \log T_2^* + \boldsymbol{\rho}^{T_2^*}$, where $\hat{\mathbf{u}}$ is the max-margin solution on $\{\tilde{\mathbf{x}}_i - \tilde{\mathbf{z}}_i\}_{i=1}^n$, $\boldsymbol{\rho}^{T_2^*}$ a bounded residual, and $T_2^* \asymp d^2 / \sqrt{\sigma^2 \log d/n}$. This completes the proof of Theorem 1. The Stage 3 bounds follow by combining Lemma 12 and Lemma 13.

As a preparatory step, we provide a recursion expression of $\mathbf{w}^t$ in Stage 3, which will be used repeatedly in the proof. The update of $\mathbf{w}^t$ in Stage 3 takes the form:

$$\mathbf{w}^{t+1} = \mathbf{w}^t \left[\mathbf{1}_d - \frac{\eta}{n} \sum_{i=1}^n \left((\mathbf{x}^i)^\top (\mathbf{w}^t)^{\odot 2} - (1 - p_i^{T_2^*})(\mathbf{x}^i - \mathbf{z}^i)^\top (\mathbf{w}^t)^{\odot 2} \right.\right.$$
$$\left.\left. - (\mathbf{x}^i)^\top (\mathbf{w}^*)^{\odot 2} - \epsilon_i\right) \cdot \left(\mathbf{x}^i - (1 - p_i^{T_2^*})(\mathbf{x}^i - \mathbf{z}^i)\right)\right]. \tag{41}$$

Collecting the small factors involving $\left(1 - p_i^{T_2^*}\right)$ yields:

$$
\begin{aligned}
\mathbf{w}^{t+1} = \mathbf{w}^t \odot \Bigg[ \mathbf{1}_d - \eta \bigg( &\frac{1}{n} \sum_{i=1}^{n} \mathbf{x}^i (\mathbf{x}^i)^\top \left( (\mathbf{w}^t)^{\odot 2} - (\mathbf{w}^*)^{\odot 2} \right) - \frac{1}{n} \sum_{i=1}^{n} \mathbf{x}^i \epsilon_i \\
&- \frac{1}{n} \sum_{i=1}^{n} \left(1 - p_i^{T_2^*}\right) \mathbf{x}^i (\mathbf{x}^i - \mathbf{z}^i)^\top (\mathbf{w}^t)^{\odot 2} \\
&- \frac{1}{n} \sum_{i=1}^{n} \left(1 - p_i^{T_2^*}\right) (\mathbf{x}^i - \mathbf{z}^i)(\mathbf{x}^i)^\top \left( (\mathbf{w}^t)^{\odot 2} - (\mathbf{w}^*)^{\odot 2} \right) \\
&+ \frac{1}{n} \sum_{i=1}^{n} \left(1 - p_i^{T_2^*}\right) (\mathbf{x}^i - \mathbf{z}^i) \epsilon_i \\
&- \frac{1}{n} \sum_{i=1}^{n} \left(1 - p_i^{T_2^*}\right)^2 (\mathbf{x}^i - \mathbf{z}^i)(\mathbf{x}^i - \mathbf{z}^i)^\top (\mathbf{w}^t)^{\odot 2} \bigg) \Bigg].
\end{aligned} \tag{42}
$$

We then decompose $\mathbf{w}^t$ over the support of $\mathbf{w}^*$. Let $\mathrm{supp}\,(\mathbf{w}^*) = S$, with $|S| = r$. Define $\mathbf{w}_S^t = \mathbf{w}^t \odot \mathbf{1}_S$, $\mathbf{w}_{S^c}^t = \mathbf{w}^t \odot \mathbf{1}_{S^c}$. Then we define,

$$
\begin{aligned}
\mathbf{r}^t = &\left( \frac{1}{n} \sum_{i=1}^{n} \mathbf{x}^i \left(\mathbf{x}^i\right)^\top - \mathbf{H} \right) \left( (\mathbf{w}_S^t)^{\odot 2} - (\mathbf{w}^*)^{\odot 2} \right), \\
\mathbf{e}^t = &\bigg( \frac{1}{n} \sum_{i=1}^{n} \mathbf{x}^i (\mathbf{x}^i)^\top (\mathbf{w}_{S^c}^t)^{\odot 2} - \frac{1}{n} \sum_{i=1}^{n} \mathbf{x}^i \epsilon_i - \frac{1}{n} \sum_{i=1}^{n} \left(1 - p_i^{T_2^*}\right) \mathbf{x}^i (\mathbf{x}^i - \mathbf{z}^i)^\top (\mathbf{w}^t)^{\odot 2} \\
&- \frac{1}{n} \sum_{i=1}^{n} \left(1 - p_i^{T_2^*}\right) (\mathbf{x}^i - \mathbf{z}^i)(\mathbf{x}^i)^\top \left( (\mathbf{w}^t)^{\odot 2} - (\mathbf{w}^*)^{\odot 2} \right) + \frac{1}{n} \sum_{i=1}^{n} \left(1 - p_i^{T_2^*}\right) (\mathbf{x}^i - \mathbf{z}^i) \epsilon_i \\
&- \frac{1}{n} \sum_{i=1}^{n} \left(1 - p_i^{T_2^*}\right)^2 (\mathbf{x}^i - \mathbf{z}^i)(\mathbf{x}^i - \mathbf{z}^i)^\top (\mathbf{w}^t)^{\odot 2} \bigg).
\end{aligned} \tag{43}
$$

Then the dynamic of $\mathbf{w}^t$ can be expressed as

$$
\mathbf{w}^{t+1} = \mathbf{w}^t \odot \left[ \mathbf{1}_d - \eta \mathbf{H} \left( (\mathbf{w}_S^t)^{\odot 2} - (\mathbf{w}^*)^{\odot 2} \right) - \mathbf{r}^t + \mathbf{e}^t \right]. \tag{44}
$$

With $\mathbf{H} = \mathrm{diag}(a, 1, \ldots, 1)$, let $c = a$ for $j = 1$ and $c = 1$ for $j > 1$. Since $(\mathbf{w}_j^*)^2 = \mathbf{w}_j^* \in \{0, 1\}$,

$$
\begin{aligned}
\text{If } j \in S: \quad & \mathbf{w}_j^{t+1} = \mathbf{w}_j^{t+1} \left( 1 - \eta \left( c \left(\mathbf{w}_j^{t+1}\right)^2 - c - \mathbf{r}_t - \mathbf{e}_t \right) \right), \\
\text{If } j \notin S: \quad & \mathbf{w}_j^{t+1} = \mathbf{w}_j^t \left( 1 + \eta \left( \mathbf{r}_t + \mathbf{e}_t \right) \right).
\end{aligned}
$$

**Lemma 12.** *Define $B_1 := C_1\, r \sqrt{\frac{\log(dr)}{n}}$ and $B_2 := C_2 \sqrt{\frac{\sigma^2 \log(dr)}{n}}$, for absolute constants $C_1 > 0$. Assume the step size $0 < \eta \leq \frac{1}{8\,a\,(1 + B_1)}$. Set the phase lengths*

$$
T_4^* := \left\lceil \frac{5}{4\eta} \log \left( \frac{\max\{a(\mathbf{w}_1^{T_2^*})^2,\, 1\}}{5a\, B_1} \right) \right\rceil + \left\lceil \frac{5}{4\eta} \right\rceil, \quad T_5^* := \left\lceil \frac{5}{4\eta} \log \left( \frac{B_1}{B_2} \right) \right\rceil, \quad T_3^* := T_4^* + T_5^*.
$$

*Then, with probability at least $1 - \frac{1}{d^2}$, the following statements hold. For all $t \leq T_2^* + T_3^*$,*

$$
\forall j \in S : 0 \leq \mathbf{w}_j^t \leq 1 + B_1, \qquad \forall j \in S^c : \mathbf{w}_j^t \leq \frac{1}{d} \sqrt{\frac{\sigma^2 \log d}{n}}.
$$

*Proof of Lemma 12.* Define the inductive property $\mathsf{P}(t)$ for $t \geq T_2^*$:

$\mathsf{P}(t): \forall s \in [T_2^*, t],\, \forall j \in S : 0 \leq \mathbf{w}_j^s \leq 1 + B_1,\, \forall s \in [T_2^*, t],\, \forall j \in S^c : \mathbf{w}_j^s \leq d^{-1} \sqrt{\sigma^2 \log d / n}.$

Base case holds by initialization. Assuming $\mathsf{P}(t)$, with concerntration Lemma 14 15 16 17 18  19 and  20 we bound
$$\|\mathbf{r}^t\|_\infty \le 2L_r r \le \tfrac{1}{2}B_1, \qquad \|\mathbf{e}^t\|_\infty \le \tfrac{1}{2}B_1,$$
hence $\|\mathbf{r}^t-\mathbf{e}^t\|_\infty \le B_1$. By Lemma 21 with $B = B_1$ and the stepsize, we have $0 \le \mathbf{w}_j^{t+1} \le 1+B_1$ for $j \in S$. For $j \in S^c$,
$$\mathbf{w}_j^{t+1} \le (1+\eta B_1)\,\mathbf{w}_j^t \;\Rightarrow\; \mathbf{w}_j^{T_2^*+\tau} \le e^{\eta B_1 \tau}\mathbf{w}_j^{T_2^*}.$$

Since $T_3^* := T_4^* + T_5^*$ satisfies
$$T_3^* \;\le\; \Big\lceil \frac{1}{\eta B_1} \log\Big( \frac{d\,\sqrt{\sigma^2 \log d/n}}{\mathbf{w}_{\max,S^c}^{T_2^*}} \Big)\Big\rceil,$$

then $\mathbf{w}_j^{T_2^*+\tau} \le d^{-1}\sqrt{\sigma^2 \log d/n}$ for all $\tau \le T_3^*$. Thus $\mathsf{P}(t+1)$ holds. By induction, the claim holds. $\qquad\square$

**Lemma 13.** *For all $t \ge T_2^* + T_4^*$ and all $j \in S$,*
$$\big|\mathbf{w}_j^t - 1\big| \;\le\; 5\,B_1.$$
*If $n \gtrsim \sigma^2 r^2 \log^3(dr)$, then for all $t \ge T_2^* + T_4^* + T_5^* = T_2^* + T_3^*$ and all $j \in S$,*
$$\big|\mathbf{w}_j^t - 1\big| \;\le\; 5\,B_2.$$

*Proof of Lemma 13.* Fix $j \in S$ and $x_t := c(\mathbf{w}_j^t)^2$, $c \in \{a,1\}$, $\gamma = 1/4$, $\gamma_w = \gamma/2 = 1/8$. When $\mathbf{w}_j^t \le 1 - \gamma_w$,
$$\mathbf{w}_j^{t+1} = \mathbf{w}_j^t \Big( 1 + \eta\big(c(1-(\mathbf{w}_j^t)^2) - b_t\big)\Big),$$
with $|b_t| \le B_1$. Since $1-(\mathbf{w}_j^t)^2 \ge 15/64$, assuming $B_1 \le 11/64$ yields $c(1-(\mathbf{w}_j^t)^2)-b_t \ge 11/64$, hence $\mathbf{w}_j^{t+1} \ge (1+\eta \cdot 11/64)\mathbf{w}_j^t$. From $\mathbf{w}_j^{T_2^*} \in [1/4, 1/2]$, in at most
$$T(B_1) \le \Big\lceil \frac{64}{11\,\eta} \log \frac{1-\gamma_w}{\mathbf{w}_j^{T_2^*}} \Big\rceil \le \frac{33}{\eta}$$

steps we have $\mathbf{w}_j^t \ge 1 - \gamma_w$, i.e., $|x_t - c| \le \gamma$. Then by Lemma 22 (with $B = B_1$),
$$|x_{t+1} - c| \le (1-\kappa\eta)|x_t - c| + \beta c\eta B_1, \quad \kappa = \tfrac{2}{3},\ \beta = 2.1,$$
and whenever $|x_t - c| \ge \Lambda c B_1$ with $\Lambda \ge 2\beta/\kappa$ ($\Lambda = 7$),
$$|x_{t+1} - c| \le \big(1 - \tfrac{\kappa}{2}\eta\big)|x_t - c|.$$
Hence after
$$T_4^* \;\le\; \Big\lceil \frac{5}{4\eta} \log \frac{|x_{T_2^*} - c|}{5cB_1} \Big\rceil + \Big\lceil \frac{5}{4\eta} \Big\rceil$$
iterations, $|x_t - c| \le 5cB_1$, which implies $|\mathbf{w}_j^t - 1| \le 5B_1$.

This implies $\|(\mathbf{w}_S^t)^{\odot 2} - (\mathbf{w}^*)^{\odot 2}\|_\infty \le C_w B_1$. Therefore,
$$\|\mathbf{r}^t\|_\infty \le L_r\, r\, C_w B_1 \le C_r \frac{\sigma^2 r^2 \log^2(dr)}{n} = B_2, \qquad \|\mathbf{e}^t\|_\infty \le C_e \sqrt{\frac{\sigma^2 \log d}{n}} \le B_2.$$
By Lemma 21 with $B = B_2$, for $t \ge T_2^* + T_4^* + T_5^*$,
$$T_5^* \;=\; \Big\lceil \frac{5}{4\eta} \log \frac{B_1}{B_2} \Big\rceil \;\Rightarrow\; |\mathbf{w}_j^t - 1| \;\le\; 5B_2.$$

$\qquad\square$

**Lemma 14.** *Given vector $\mathbf{u}$ supported on $S$, for any $\delta \in (0,1)$, with probability at least $1 - \delta$,*
$$\Big\|\Big(\frac{1}{n}\sum_{i=1}^n \mathbf{x}^i(\mathbf{x}^i)^\top - \mathbf{H}\Big)\mathbf{u}\Big\|_\infty \;\le\; B_{\mathbf{x}}^2 \sqrt{\frac{2}{n} \log\Big(\frac{2dr}{\delta}\Big)} \cdot r\,\|\mathbf{u}\|_\infty.$$

*Proof of Lemma 14.* Fix $j \in [d]$. Then

$$\left[\left(\tfrac{1}{n}\sum_i \mathbf{x}^i(\mathbf{x}^i)^\top - \mathbf{H}\right)\mathbf{u}\right]_j = \sum_{k \in S}\left(\tfrac{1}{n}\sum_i \mathbf{x}_j^i\mathbf{x}_k^i - \mathbb{E}[\mathbf{x}_j\mathbf{x}_k]\right)\mathbf{u}_k.$$

For each $k$, $\mathbf{x}_j^i\mathbf{x}_k^i \in [-B_\mathbf{x}^2, B_\mathbf{x}^2]$, so Hoeffding implies

$$\Pr\left(\left|\tfrac{1}{n}\sum_i \mathbf{x}_j^i\mathbf{x}_k^i - \mathbb{E}[\mathbf{x}_j\mathbf{x}_k]\right| \geq t\right) \leq 2\exp\left(-\frac{2nt^2}{(2B_\mathbf{x}^2)^2}\right).$$

A union bound over $(j,k) \in [d] \times S$ gives the uniform deviation. Hence,

$$\left|\left[\left(\tfrac{1}{n}\sum_i \mathbf{x}^i(\mathbf{x}^i)^\top - \mathbf{H}\right)\mathbf{u}\right]_j\right| \leq B_\mathbf{x}^2\sqrt{\tfrac{2}{n}\log(\tfrac{2dr}{\delta})}\sum_{k \in S}|\mathbf{u}_k| \leq B_\mathbf{x}^2\sqrt{\tfrac{2}{n}\log(\tfrac{2dr}{\delta})}\cdot r\|\mathbf{u}\|_\infty,$$

and taking the maximum over $j$ yields the claim. $\square$

**Theorem 15.** *Given $\mathbf{u} \in \mathbb{R}^d$, for any $\delta \in (0,1)$, with probability at least $1-\delta$,*

$$\left\|\tfrac{1}{n}\sum_{i=1}^n \mathbf{x}^i(\mathbf{x}^i)^\top\mathbf{u}\right\|_\infty \leq \left[\max\{a,1\} + B_\mathbf{x}^2\,d\,\sqrt{\tfrac{2\log(2d^2/\delta)}{n}}\right]\|\mathbf{u}\|_\infty.$$

*Proof of Lemma 15.* Decompose

$$\frac{1}{n}\sum_{i=1}^n \mathbf{x}^i(\mathbf{x}^i)^\top\mathbf{u} = \mathbf{H}\mathbf{u} + \left(\tfrac{1}{n}\sum_{i=1}^n \mathbf{x}^i(\mathbf{x}^i)^\top - \mathbf{H}\right)\mathbf{u}.$$

Since $\mathbf{H} = \mathrm{diag}(a,1,\ldots,1)$,

$$\|\mathbf{H}\mathbf{u}\|_\infty \leq \max\{a,1\}\|\mathbf{u}\|_\infty.$$

For the deviation, define

$$Y_{i,j} := \mathbf{x}_j^i(\mathbf{x}^i)^\top\mathbf{u} - \mathbb{E}[\mathbf{x}_j\,\mathbf{x}^\top\mathbf{u}], \qquad \left[\left(\tfrac{1}{n}\sum_{i=1}^n \mathbf{x}^i(\mathbf{x}^i)^\top - \mathbf{H}\right)\mathbf{u}\right]_j = \frac{1}{n}\sum_{i=1}^n Y_{i,j}.$$

The $Y_{i,j}$ are i.i.d., mean zero, and almost surely

$$|Y_{i,j}| \leq |\mathbf{x}_j^i|\sum_{k=1}^d |\mathbf{x}_k^i|\,|\mathbf{u}_k| \leq B_\mathbf{x}\cdot(d\,B_\mathbf{x})\|\mathbf{u}\|_\infty = B_\mathbf{x}^2\,d\,\|\mathbf{u}\|_\infty.$$

By Hoeffding's inequality,

$$\Pr\left(\left|\tfrac{1}{n}\sum_{i=1}^n Y_{i,j}\right| \geq t\right) \leq 2\exp\left(-\frac{n\,t^2}{2B_\mathbf{x}^4 d^2\,\|\mathbf{u}\|_\infty^2}\right).$$

A union bound over $j = 1,\ldots,d$ gives, with probability at least $1-\delta$,

$$\max_{1 \leq j \leq d}\left|\tfrac{1}{n}\sum_{i=1}^n Y_{i,j}\right| \leq B_\mathbf{x}^2\,d\,\|\mathbf{u}\|_\infty\sqrt{\frac{2\log(2d^2/\delta)}{n}}.$$

Combining with the population bound yields the result. $\square$

**Lemma 16.** *For any $\delta \in (0,1)$, with probability at least $1-\delta$,*

$$\left\|\tfrac{1}{n}\sum_{i=1}^n \mathbf{x}^i\epsilon_i\right\|_\infty \leq B_\mathbf{x}B_\epsilon\sqrt{\tfrac{2}{n}\log\left(\tfrac{2d}{\delta}\right)}.$$

*Proof of Lemma 16.* Apply Hoeffding coordinatewise and union bound over $d$ coordinates. $\square$

**Lemma 17.** *Define $B_y := r B_{\mathbf{x}} + B_\epsilon$ and $B_{\mathbf{xz}} := B_{\mathbf{x}} + \|f(0)\|_\infty + L B_y + B_{\boldsymbol{\xi}}$. Then, given $\mathbf{u} \in \mathbb{R}^d$, for any $\delta \in (0,1)$, with probability at least $1 - \delta$,*

$$\left\| \tfrac{1}{n} \sum_{i=1}^n \mathbf{x}^i (\mathbf{x}^i - \mathbf{z}^i)^\top \mathbf{u}_S \right\|_\infty + \left\| \tfrac{1}{n} \sum_{i=1}^n \mathbf{x}^i (\mathbf{x}^i - \mathbf{z}^i)^\top \mathbf{u}_{S^c} \right\|_\infty$$

$$\leq B_{\mathbf{x}} B_{\mathbf{xz}} \left( 1 + \sqrt{\tfrac{2 \log(4d/\delta)}{n}} \right) \left( r \|\mathbf{u}_S\|_\infty + (d-r) \|\mathbf{u}_{S^c}\|_\infty \right).$$

*Consequently, with $p_{\min} = \min_i p_i^{T_2^*}$,*

$$\left\| \tfrac{1}{n} \sum_{i=1}^n \left( 1 - p_i^{T_2^*} \right) \mathbf{x}^i (\mathbf{x}^i - \mathbf{z}^i)^\top (\mathbf{w}^t)^{\odot 2} \right\|_\infty$$

$$\leq (1 - p_{\min}) B_{\mathbf{x}} B_{\mathbf{xz}} \left( 1 + \sqrt{\tfrac{2 \log(4d/\delta)}{n}} \right) \left( r \|(\mathbf{w}_S^t)^{\odot 2}\|_\infty + (d-r) \|(\mathbf{w}_{S^c}^t)^{\odot 2}\|_\infty \right).$$

*Proof of Lemma 17.* By the model assumptions, $\|\mathbf{x}\|_\infty \leq B_{\mathbf{x}}$ almost surely. With $y = \mathbf{x}^\top (\mathbf{w}^*)^{\odot 2} + \epsilon$ and $\|\mathbf{x}\|_\infty \leq B_{\mathbf{x}}$, we have $|y| \leq r B_{\mathbf{x}} + B_\epsilon =: B_y$. Since $\mathbf{z} = f(y) + \boldsymbol{\xi}$ with $f$ being $L$-Lipschitz and $\|\boldsymbol{\xi}\|_\infty \leq B_{\boldsymbol{\xi}}$, it follows that

$$\|\mathbf{z}\|_\infty \leq \|f(0)\|_\infty + L|y| + \|\boldsymbol{\xi}\|_\infty \leq \|f(0)\|_\infty + L B_y + B_{\boldsymbol{\xi}}.$$

Hence $\|\mathbf{x} - \mathbf{z}\|_\infty \leq B_{\mathbf{x}} + \|f(0)\|_\infty + L B_y + B_{\boldsymbol{\xi}} =: B_{\mathbf{xz}}$.

Fix any $j \in [d]$. For the $j$-th coordinate of $\frac{1}{n} \sum_{i=1}^n \mathbf{x}^i (\mathbf{x}^i - \mathbf{z}^i)^\top \mathbf{u}_S$, we have

$$\left| \frac{1}{n} \sum_{i=1}^n \mathbf{x}_j^i \left( (\mathbf{x}^i - \mathbf{z}^i)^\top \mathbf{u}_S \right) \right| \leq \frac{1}{n} \sum_{i=1}^n |\mathbf{x}_j^i| \, \|\mathbf{x}^i - \mathbf{z}^i\|_\infty \|\mathbf{u}_S\|_1 \leq B_{\mathbf{x}} B_{\mathbf{xz}} \|\mathbf{u}_S\|_1.$$

Taking expectation shows the same bound for the population mean. For the centered fluctuations, define

$$Y_i^{(j,S)} := \mathbf{x}_j^i \left( (\mathbf{x}^i - \mathbf{z}^i)^\top \mathbf{u}_S \right) - \mathbb{E}\left[ \mathbf{x}_j^i \left( (\mathbf{x}^i - \mathbf{z}^i)^\top \mathbf{u}_S \right) \right].$$

Then $|Y_i^{(j,S)}| \leq 2 B_{\mathbf{x}} B_{\mathbf{xz}} \|\mathbf{u}_S\|_1$ almost surely, and by Hoeffding's inequality,

$$\Pr\left( \left| \frac{1}{n} \sum_{i=1}^n Y_i^{(j,S)} \right| \geq B_{\mathbf{x}} B_{\mathbf{xz}} \|\mathbf{u}_S\|_1 \sqrt{\frac{2 \log(2d/\delta)}{n}} \right) \leq \frac{\delta}{2d}.$$

A union bound over $j \in [d]$ yields, with probability at least $1 - \delta/2$,

$$\left\| \tfrac{1}{n} \sum_{i=1}^n \mathbf{x}^i (\mathbf{x}^i - \mathbf{z}^i)^\top \mathbf{u}_S \right\|_\infty \leq B_{\mathbf{x}} B_{\mathbf{xz}} \left( 1 + \sqrt{\tfrac{2 \log(2d/\delta)}{n}} \right) \|\mathbf{u}_S\|_1.$$

Using $\|\mathbf{u}_S\|_1 \leq r \|\mathbf{u}_S\|_\infty$ completes the bound for the $\mathbf{u}_S$ term. An identical argument with $\mathbf{u}_{S^c}$ (and $\|\mathbf{u}_{S^c}\|_1 \leq (d-r) \|\mathbf{u}_{S^c}\|_\infty$) gives, with probability at least $1 - \delta/2$,

$$\left\| \tfrac{1}{n} \sum_{i=1}^n \mathbf{x}^i (\mathbf{x}^i - \mathbf{z}^i)^\top \mathbf{u}_{S^c} \right\|_\infty \leq B_{\mathbf{x}} B_{\mathbf{xz}} \left( 1 + \sqrt{\tfrac{2 \log(2d/\delta)}{n}} \right) \|\mathbf{u}_{S^c}\|_1.$$

A union bound over the two events replaces $\log(2d/\delta)$ by $\log(4d/\delta)$ and yields the first display.

For the consequence, note $0 \leq 1 - p_i^{T_2^*} \leq (1 - p_{\min})$. Factoring this out and applying the first bound with $\mathbf{u} = (\mathbf{w}^t)^{\odot 2}$ (so $\mathbf{u}_S = (\mathbf{w}_S^t)^{\odot 2}$ and $\mathbf{u}_{S^c} = (\mathbf{w}_{S^c}^t)^{\odot 2}$) proves the second display. $\qquad\square$

**Lemma 18.** *Under the same assumptions as Lemma 17, for any $\delta \in (0,1)$, with probability at least $1 - \delta$,*

$$\left\| \tfrac{1}{n} \sum_{i=1}^n (\mathbf{x}^i - \mathbf{z}^i)(\mathbf{x}^i)^\top \mathbf{u}_S \right\|_\infty + \left\| \tfrac{1}{n} \sum_{i=1}^n (\mathbf{x}^i - \mathbf{z}^i)(\mathbf{x}^i)^\top \mathbf{u}_{S^c} \right\|_\infty$$

$$\leq B_{\mathbf{x}} B_{\mathbf{xz}} \left( 1 + \sqrt{\tfrac{2 \log(\frac{4d}{\delta})}{n}} \right) \left( r \|\mathbf{u}_S\|_\infty + (d-r) \|\mathbf{u}_{S^c}\|_\infty \right).$$

*Consequently, with $p_{\min} = \min_i p_i^{T_2^*}$,*

$$\left\| \frac{1}{n} \sum_{i=1}^{n} \left(1 - p_i^{T_2^*}\right)(\mathbf{x}^i - \mathbf{z}^i)(\mathbf{x}^i)^\top \left((\mathbf{w}^t)^{\odot 2} - (\mathbf{w}^*)^{\odot 2}\right) \right\|_\infty$$

$$\leq (1 - p_{\min}) B_{\mathbf{x}} B_{\mathbf{xz}} \left(1 + \sqrt{\frac{2 \log(\frac{4d}{\delta})}{n}}\right) \left(r \|(\mathbf{w}_S^t)^{\odot 2} - (\mathbf{w}_S^*)^{\odot 2}\|_\infty + (d - r)\|(\mathbf{w}_{S^c}^t)^{\odot 2}\|_\infty\right).$$

*Proof of Lemma 18.* Same proof as Lemma 17, exchanging the two factors. $\qquad\square$

**Lemma 19.** *Let $S_n := \frac{1}{n} \sum_{i=1}^{n} (\mathbf{x}^i - \mathbf{z}^i)\,\epsilon_i \in \mathbb{R}^d$. Then for any $\delta \in (0, 1)$, with probability at least $1 - \delta$,*

$$\left\| \frac{1}{n} \sum_{i=1}^{n} \left(1 - p_i^{T_2^*}\right)(\mathbf{x}^i - \mathbf{z}^i)\,\epsilon_i \right\|_\infty$$

$$\leq (1 - p_{\min}) \left( B_{\mathbf{xz}}\,\sigma + B_{\mathbf{xz}} \sqrt{\frac{2\sigma^2 \log(\frac{2d}{\delta})}{n}} + \frac{2}{3} B_{\mathbf{xz}} B_\epsilon \frac{\log(\frac{2d}{\delta})}{n} \right).$$

*Proof of Lemma 19.* Fix $j \in [d]$ and define $X_i^{(j)} := (\mathbf{x}_j^i - \mathbf{z}_j^i)\,\epsilon_i$. Since $\|\mathbf{x}^i - \mathbf{z}^i\|_\infty \leq B_{\mathbf{xz}}$ a.s., we have $|X_i^{(j)}| \leq B_{\mathbf{xz}} B_\epsilon$ and $\mathrm{Var}(X_i^{(j)}) \leq B_{\mathbf{xz}}^2 \sigma^2$. Let $\mu_j := \mathbb{E}[X_i^{(j)}] = \mathbb{E}[(\mathbf{x}_j - \mathbf{z}_j)\epsilon]$. By Cauchy–Schwarz and the a.s. bound, $|\mu_j| \leq B_{\mathbf{xz}}\,\sigma$. Let $Y_i^{(j)} := X_i^{(j)} - \mu_j$, then $|Y_i^{(j)}| \leq |X_i^{(j)}| + |\mu_j| \leq B_{\mathbf{xz}}(B_\epsilon + \sigma)$ and $\mathrm{Var}(Y_i^{(j)}) = \mathrm{Var}(X_i^{(j)}) \leq B_{\mathbf{xz}}^2\sigma^2$. By Bernstein's inequality, for any $t > 0$,

$$\Pr\left( \left| \frac{1}{n} \sum_{i=1}^{n} Y_i^{(j)} \right| \geq t \right) \leq 2 \exp\left( -\frac{nt^2}{2B_{\mathbf{xz}}^2 \sigma^2 + \frac{2}{3} B_{\mathbf{xz}}(B_\epsilon + \sigma)\,t} \right).$$

Choose $t = B_{\mathbf{xz}}\sigma \sqrt{\frac{2\log(2d/\delta)}{n}} + \frac{2}{3} B_{\mathbf{xz}}(B_\epsilon + \sigma)\frac{\log(2d/\delta)}{n}$ and take a union bound over $j = 1, \ldots, d$ to obtain, with probability at least $1 - \delta$,

$$\left\| \frac{1}{n} \sum_{i=1}^{n} X_i^{(j)} \right\|_\infty \leq \|\boldsymbol{\mu}\|_\infty + B_{\mathbf{xz}}\sigma \sqrt{\frac{2\log(2d/\delta)}{n}} + \frac{2}{3} B_{\mathbf{xz}}(B_\epsilon + \sigma)\frac{\log(2d/\delta)}{n}.$$

Since $\|\boldsymbol{\mu}\|_\infty \leq B_{\mathbf{xz}}\sigma$ and $B_\epsilon + \sigma \leq 2B_\epsilon$, we have

$$\left\| \frac{1}{n} \sum_{i=1}^{n} X_i^{(j)} \right\|_\infty \leq B_{\mathbf{xz}}\sigma + B_{\mathbf{xz}}\sigma \sqrt{\frac{2\log(2d/\delta)}{n}} + \frac{2}{3} B_{\mathbf{xz}} B_\epsilon \frac{\log(2d/\delta)}{n}.$$

Finally, $0 \leq 1 - p_i^{T_2^*} \leq (1 - p_{\min})$ implies

$$\left\| \frac{1}{n} \sum_{i=1}^{n} \left(1 - p_i^{T_2^*}\right)(\mathbf{x}^i - \mathbf{z}^i)\,\epsilon_i \right\|_\infty$$

$$\leq (1 - p_{\min}) \left[ B_{\mathbf{xz}}\sigma + B_{\mathbf{xz}}\sigma \sqrt{\frac{2\log(2d/\delta)}{n}} + \frac{2}{3} B_{\mathbf{xz}} B_\epsilon \frac{\log(2d/\delta)}{n} \right],$$

which completes the proof. $\qquad\square$

**Lemma 20.** *Let $p_{\min} = \min_i p_i^{T_2^*}$. For any $\delta \in (0, 1)$, with probability at least $1 - \delta$,*

$$\left\| \frac{1}{n} \sum_{i=1}^{n} \left(1 - p_i^{T_2^*}\right)^2 (\mathbf{x}^i - \mathbf{z}^i)(\mathbf{x}^i - \mathbf{z}^i)^\top (\mathbf{w}^t)^{\odot 2} \right\|_\infty$$

$$\leq (1 - p_{\min})^2 B_{\mathbf{xz}}^2 \left(1 + \sqrt{\frac{2\log(4d/\delta)}{n}}\right) \left(r\|(\mathbf{w}_S^t)^{\odot 2}\|_\infty + (d - r)\|(\mathbf{w}_{S^c}^t)^{\odot 2}\|_\infty\right).$$

*Proof of Lemma 20.* Apply the same argument as in Lemma 17. $\qquad\square$

Consider the scalar update

$$w^+ = G_{c,b}(w) := w\big(1 - \eta(cw^2 - c - b)\big), \qquad c \geq 1, \ |b| \leq B, \ \eta > 0,$$

and let $x := cw^2$, $T_b(x) := x(1 - \eta(x - (c+b)))^2$ (so $x^+ = T_b(x)$).

**Lemma 21** (Invariance and monotonicity). *If $0 < \eta \leq \frac{1}{8\,c\,(1+B)^2}$, then for every $|b| \leq B$ the map $G_{c,b}$ is nondecreasing on $[0, 1+B]$ and $G_{c,b}\big([0, 1+B]\big) \subseteq [0, 1+B]$.*

*Proof of Lemma 21.* We have $G'_{c,b}(w) = 1 - \eta(3cw^2 - c - b)$. On $[0, 1+B]$, $\max(3cw^2 - c - b) \leq 3c(1+B)^2$, hence $G'_{c,b} \geq 0$ if $\eta \leq 1/[3c(1+B)^2]$, which is implied by $\eta \leq 1/[8c(1+B)^2]$. For invariance, note that $G_{c,b}(0) = 0$, and by monotonicity in both $w$ and $b$,

$$G_{c,b}(w) \leq G_{c,B}(1+B) = (1+B)\Big(1 - \eta\big(c((1+B)^2 - 1) - B\big)\Big) \leq 1 + B,$$

since $c((1+B)^2 - 1) - B \geq B > 0$ and $\eta\big(c((1+B)^2 - 1) - B\big) \leq 1$. Nonnegativity of the bracket is ensured because

$$\min_{w \in [0,1+B], \ |b| \leq B} \Big(1 - \eta(cw^2 - c - b)\Big) = 1 - \eta\big(c(1+B)^2 - c + B\big) \geq 0$$

under $\eta \leq 1/[8c(1+B)^2]$. $\qquad\square$

**Lemma 22** (Local contraction with bounded noise). *Fix $\gamma \in (0, 1/4]$, assume $|b| \leq B$, $B \leq 1/8$ and $\eta \leq 1/8$. If $|x - c| \leq \gamma$, then*

$$|x^+ - c| \ \leq \ (1 - \kappa\eta)\,|x - c| + \beta\,c\,\eta\,B, \qquad \kappa = \tfrac{2}{3}, \ \beta = 2.1.$$

*Moreover, if $|x - c| \geq \Lambda cB$ with $\Lambda \geq 2\beta/\kappa$ (e.g. $\Lambda = 7$), then*

$$|x^+ - c| \ \leq \ \big(1 - \tfrac{\kappa}{2}\eta\big)\,|x - c| \ \leq \ \big(1 - \tfrac{1}{3}\eta\big)\,|x - c|.$$

*Consequently,*

$$t \ \geq \ \frac{2}{\kappa\eta} \log \frac{|x_0 - c|}{\Lambda cB} \quad \Rightarrow \quad |w_t - 1| \leq \Lambda B.$$

*Proof of Lemma 22.* Let $y := x - (c+b)$. Then $T'_b(x) = (1 - \eta y)(1 - \eta y - 2\eta x)$. On $|x - c| \leq \gamma$, $|b| \leq B$, we have $|y| \leq \gamma + B \leq 3/8$ and $2x - |y| \geq 2(c - \gamma) - (\gamma + B) \geq 9/8$. Hence

$$|T'_b(x)| \leq (1 + \eta|y|)\big(1 - \eta(2x - |y|)\big) \leq 1 - \big((2x - |y|) - |y|\big)\eta + |y|(2x - |y|)\eta^2.$$

Using $(2x - |y|) - |y| \geq 2c - 4\gamma - 2B \geq 3/4$, $|y|(2x - |y|) \leq (3/8)(9/8) = 27/64$, and $\eta \leq 1/8$, we get $|T'_b(x)| \leq 1 - \tfrac{2}{3}\eta$. By the mean-value theorem, $|T_b(x) - T_b(c)| \leq (1 - \kappa\eta)|x - c|$ with $\kappa = 2/3$. Also $|T_b(c) - c| = |c(1 + \eta b)^2 - c| \leq 2c\eta B + c\eta^2 B^2 \leq 2.1\,c\,\eta B$ for $\eta \leq 1/8$. Combining gives the first claim. If $|x - c| \geq \Lambda cB$, then

$$|x^+ - c| \leq \Big(1 - \kappa\eta + \frac{\beta}{\Lambda}\eta\Big)|x - c| \leq \big(1 - \tfrac{\kappa}{2}\eta\big)|x - c|$$

whenever $\Lambda \geq 2\beta/\kappa$. This yields the second claim and the exponential-time bound. $\qquad\square$

### A.4 PROOF OF THEOREM 2

Recall $\hat{\mathbf{u}}$ be the max-margin solution on $\{\mathbf{u}^i\}_{i=1}^n$ and $\gamma_{\mathrm{emp}} := 1/\|\hat{\mathbf{u}}\|_2$.

**Lemma 23.** *Fix $\varepsilon \in (0, 1)$ and $\delta \in (0, 1)$. With probability at least $1 - \delta$ over the training sample,*

$$\Pr_{\mathbf{u}}\big((\hat{\mathbf{u}})^\top \mathbf{u} > \varepsilon\big) \ \geq \ 1 - \frac{2\sqrt{\|\mathbf{s}\|_2^2 + d(B_{\mathbf{x}} + B_{\mathbf{z}})^2}}{\|\mathbf{s}\|_2(1 - \varepsilon)\sqrt{n}} - \sqrt{\frac{2\ln(2/\delta)}{n}}.$$

*Proof.* Let $\mathsf{U} = \{\mathbf{u}^1, \ldots, \mathbf{u}^n\}$ be the training sample and $\mathsf{U}' = \{\mathbf{u}'^1, \ldots, \mathbf{u}'^n\}$ be another i.i.d. samples.

Let $\bar{\varepsilon} = \varepsilon/\|\hat{\mathbf{u}}\|_2 = \varepsilon\gamma_{\mathrm{emp}}$ and $\hat{\boldsymbol{v}} = \hat{\mathbf{u}}/\|\hat{\mathbf{u}}\|_2$. Define the ramp function at threshold $\bar{\varepsilon}$ and width $s > 0$:

$$\psi_{(\bar{\varepsilon}, s)}(t) := \begin{cases} 1, & t \leq \bar{\varepsilon}, \\ 1 - \frac{t - \bar{\varepsilon}}{s}, & \bar{\varepsilon} < t < \bar{\varepsilon} + s, \\ 0, & t \geq \bar{\varepsilon} + s. \end{cases}$$

Then $\psi \in [0, 1]$ and is $1/s$-Lipschitz. Set $s = \gamma_{\mathrm{emp}} - \bar{\varepsilon} = \gamma_{\mathrm{emp}}(1 - \varepsilon) > 0$. Since $\min_i \hat{\boldsymbol{v}}^\top \mathbf{u}^i = \gamma_{\mathrm{emp}} \geq \bar{\varepsilon} + s$, the empirical ramp loss is zero:

$$\frac{1}{n} \sum_{i=1}^n \psi_{(\bar{\varepsilon}, s)}(\hat{\boldsymbol{v}}^\top \mathbf{u}^i) = 0, \tag{45}$$

and $\mathbf{1}\{\hat{\boldsymbol{v}}^\top \mathbf{u} \leq \bar{\varepsilon}\} \leq \psi_{(\bar{\varepsilon}, s)}(\hat{\boldsymbol{v}}^\top \mathbf{u})$. Now apply the high-probability Rademacher uniform deviation bound (Shalev-Shwartz & Ben-David (2014), Theorem 26.5) to the loss $\ell = \psi_{(\bar{\varepsilon}, s)}$ (bounded by 1), over the linear class $\mathcal{F} = \{\mathbf{u} \mapsto \boldsymbol{v}^\top \mathbf{u} : \|\boldsymbol{v}\|_2 \leq 1\}$: with probability at least $1 - \delta$,

$$\mathbb{E}\big[\ell(\hat{\boldsymbol{v}}^\top \mathbf{u})\big] \leq \frac{1}{n} \sum_{i=1}^n \ell(\hat{\boldsymbol{v}}^\top \mathbf{u}^i) + 2\,\mathbb{E}_{\mathsf{U}'}\big[R(\ell \circ \mathcal{F} \circ \mathsf{U}')\big] + \sqrt{\tfrac{2\ln(2/\delta)}{n}}.$$

Using the contraction lemma (Shalev-Shwartz & Ben-David (2014), Lemma 26.9), $R(\ell \circ \mathcal{F} \circ \mathsf{U}') \leq \frac{1}{s} R(\mathcal{F} \circ \mathsf{U}')$. By the linear class Rademacher bound (Shalev-Shwartz & Ben-David (2014), Lemma 26.10), $R(\mathcal{F} \circ \mathsf{U}') \leq R/\sqrt{n}$ since $\|\mathbf{u}\| \leq R$ a.s. Because the empirical ramp loss is zero, we get

$$\mathbb{E}\big[\psi_{(\bar{\varepsilon}, s)}(\hat{\boldsymbol{v}}^\top \mathbf{u})\big] \leq \frac{2R}{s\sqrt{n}} + \sqrt{\frac{2\ln(2/\delta)}{n}}.$$

Finally, $\Pr(\hat{\boldsymbol{v}}^\top \mathbf{u} \leq \bar{\varepsilon}) \leq \mathbb{E}[\psi_{(\bar{\varepsilon}, s)}(\hat{\boldsymbol{v}}^\top \mathbf{u})]$ yields

$$\Pr\big((\hat{\mathbf{u}})^\top \mathbf{u} \leq \varepsilon\big) \leq \frac{2R}{\gamma_{\mathrm{emp}}(1 - \varepsilon)\sqrt{n}} + \sqrt{\frac{2\ln(2/\delta)}{n}},$$

which implies the complementary lower bound for $\Pr((\hat{\mathbf{u}})^\top \mathbf{u} > \varepsilon)$. Combining with Lemma 7 gives the explicit version. $\qquad\square$

Define

$$\mathbf{s} := \mathbf{s}_1 - \mathbf{s}_2, \qquad \mathbf{u} := \begin{bmatrix} \mathbf{s} \\ \mathbf{x} - \mathbf{z} \end{bmatrix}, \qquad \mathrm{Rad} := \sqrt{\|\mathbf{s}\|_2^2 + d(B_{\mathbf{x}} + B_{\mathbf{z}})^2}.$$

Since with probability at least $1 - \mathcal{O}(1/d^2)$ the output at $T^* = T_1^* + T_2^* + T_3^*$ satisfies

$$\tilde{\mathbf{v}}^{T^*} = \hat{\mathbf{u}} \log T_2^* + \boldsymbol{\rho}^{T_2^*}, \qquad \|\boldsymbol{\rho}^{T_2^*}\|_2 = O(1), \qquad T_2^* \asymp \frac{d^2}{\sqrt{\sigma^2 \log d/n}}.$$

Moreover,

$$\tilde{\mathbf{w}}^{T^*} = \begin{bmatrix} \mathbf{0} \\ \mathbf{w}^{T^*} \end{bmatrix}, \qquad \|\mathbf{w}_S^{T^*} - \mathbf{1}_S\|_\infty \lesssim \sqrt{\frac{\sigma^2 \log d}{n}}, \qquad \|\mathbf{w}_{S^c}^{T^*}\|_\infty \lesssim \frac{\sqrt{\sigma^2 \log d/n}}{d}.$$

Hence

$$\big\|(\mathbf{w}^{T^*})^{\odot 2} - (\mathbf{w}^*)^{\odot 2}\big\|_2^2 \lesssim r \frac{\sigma^2 \log d}{n}, \qquad \|(\mathbf{w}^{T^*})^{\odot 2}\|_2^2 \leq r(1 + o(1)).$$

Let $p = \sigma\big((\tilde{\mathbf{v}}^{T^*})^\top \mathbf{u}\big)$ and, for $\tau \in (0, 1/2)$,

$$\Omega_\tau := \{p \geq 1 - \tau\}, \qquad \kappa_\tau := \log \frac{1 - \tau}{\tau}.$$

Choose $\varepsilon_\tau := (\kappa_\tau + C_\rho \operatorname{Rad})/\log T_2^*$ with $|(\boldsymbol{\rho}^{T_2^*})^\top \mathbf{u}| \le C_\rho \operatorname{Rad}$ a.s.; then $\{\widehat{\mathbf{u}}^\top \mathbf{u} \ge \varepsilon_\tau\} \subseteq \Omega_\tau$. By Lemma 23, with probability at least $1 - \delta$ over the training sample,

$$\Pr(\Omega_\tau) \ge 1 - \frac{2\operatorname{Rad}}{\|\mathbf{s}\|_2(1-\varepsilon_\tau)\sqrt{n}} - \sqrt{\frac{2\ln(2/\delta)}{n}}. \tag{46}$$

In particular, taking $\tau := c_\tau \sigma^2 \log d/(n\,d)$ and $\log T_2^* \ge 4C_\rho \operatorname{Rad} + 2\log\left(\frac{nd}{\sigma^2 \log d}\right)$ gives $1 - \varepsilon_\tau \ge 1/4 - o(1)$ and thus $\Pr(\Omega_\tau)$ close to one; moreover $p \ge 1 - \tau \ge 1 - \frac{1}{d^2}$.

**Lemma 24.** *Let* $\Gamma_\tau := \lambda_{\max}\big(\mathbb{E}[(\mathbf{z} - \mathbf{x})(\mathbf{z} - \mathbf{x})^\top \mid \Omega_\tau]\big) \le C_\Gamma d$, $C_\Gamma = (B_\mathbf{x} + B_\mathbf{z})^2$. *Then*

$$\mathbb{E}\left[\left|\mathcal{L}\big(\tilde{\mathbf{w}}^{T^*}, \tilde{\mathbf{v}}^{T^*}\big) - \frac{\sigma^2}{2}\right| \,\Big|\, \Omega_\tau\right]$$
$$\le \lambda_{\max}(\mathbf{H})\,\|(\mathbf{w}^{T^*})^{\odot 2} - (\mathbf{w}^*)^{\odot 2}\|_2^2 \;+\; \tau^2\,\Gamma_\tau\,\|(\mathbf{w}^{T^*})^{\odot 2}\|_2^2 \;+\; B_\epsilon\,\tau\,\sqrt{\Gamma_\tau}\,\|(\mathbf{w}^{T^*})^{\odot 2}\|_2.$$

*Proof.* Let $\mu_x(\mathbf{x}) = \mathbb{E}[y \mid \mathbf{x}] = \mathbf{x}^\top (\mathbf{w}^*)^{\odot 2}$ and $\hat{y} = \big(p\,\tilde{\mathbf{x}} + (1-p)\,\tilde{\mathbf{z}}\big)^\top (\tilde{\mathbf{w}}^{T^*})^{\odot 2}$. Then we have

$$\hat{y} - \mu_x = \mathbf{x}^\top\big((\mathbf{w}^{T^*})^{\odot 2} - (\mathbf{w}^*)^{\odot 2}\big) + (1-p)\,g, \qquad g = (\mathbf{z} - \mathbf{x})^\top (\mathbf{w}^{T^*})^{\odot 2}. \tag{47}$$

Since $\epsilon \perp \mathbf{x}$, $\mathbb{E}[\epsilon] = 0$, we have

$$\mathcal{L} - \sigma^2/2 = \tfrac{1}{2}\,\mathbb{E}[(\hat{y} - \mu_x)^2] - \mathbb{E}[(\hat{y} - \mu_x)\epsilon]. \tag{48}$$

Conditioning on $\Omega_\tau$ (so $0 \le 1 - p \le \tau$), we get

$$\mathbb{E}\big[(\hat{y} - \mu_x)^2 \mid \Omega_\tau\big] \le 2\big((\mathbf{w}^{T^*})^{\odot 2} - (\mathbf{w}^*)^{\odot 2}\big)^\top \mathbf{H}\big((\mathbf{w}^{T^*})^{\odot 2} - (\mathbf{w}^*)^{\odot 2}\big) + 2\,\tau^2\,\mathbb{E}[g^2 \mid \Omega_\tau]$$
$$\le 2\,\lambda_{\max}(\mathbf{H})\|\big((\mathbf{w}^{T^*})^{\odot 2} - (\mathbf{w}^*)^{\odot 2}\big)\|_2^2 + 2\,\tau^2\,\Gamma_\tau\,\|(\mathbf{w}^{T^*})^{\odot 2}\|_2^2.$$

For the cross term,

$$|\mathbb{E}[(\hat{y} - \mu_x)\epsilon \mid \Omega_\tau]| = |\mathbb{E}[(1-p)g\,\epsilon \mid \Omega_\tau]| \le B_\epsilon\,\tau\,\mathbb{E}[|g| \mid \Omega_\tau] \le B_\epsilon\,\tau\,\sqrt{\Gamma_\tau}\,\|(\mathbf{w}^{T^*})^{\odot 2}\|_2.$$

Combining yields the claim. $\qquad\square$

*Proof of Theorem 2.* Let $\Omega := \Omega_\tau$. The probability lower bound follows from equation 46 with $\varepsilon_\tau$ and the stated scale of $T_2^*$. Since $p^{T^*} \ge 1 - \tau \ge 1 - \frac{1}{d^2}$. Apply Lemma 24 and substitute $\|\big((\mathbf{w}^{T^*})^{\odot 2} - (\mathbf{w}^*)^{\odot 2}\big)\|_2^2 \lesssim r\,\frac{\sigma^2 \log d}{n}$, $\|(\mathbf{w}^{T^*})^{\odot 2}\|_2^2 \le r$, $\Gamma_\tau \le C_\Gamma d$, and $\tau \le \frac{\sigma^2 \log d}{nd}$ to obtain the stated rate. $\qquad\square$

*Proof of Corollary 1.* Define $\mathbf{u}' = \begin{bmatrix} \mathbf{s} \\ \mathbf{x} - \mathbf{z}' \end{bmatrix}$ and $\operatorname{Rad}' := \sqrt{\|\mathbf{s}\|_2^2 + d\,(B_\mathbf{x} + B_{\mathbf{z}'})^2}$. Replicating the high-probability calibration of the gate (as in the proof of Theorem 2) with $\mathbf{u}$ replaced by $\mathbf{u}'$ and $\operatorname{Rad}$ by $\operatorname{Rad}'$ yields the event $\Omega$ with

$$\Pr(\Omega) \ge 1 - \frac{8\operatorname{Rad}'}{\|\mathbf{s}\|_2\sqrt{n}} - \sqrt{\frac{2\ln(2d^2)}{n}},$$

on which $p^{T^*} = \sigma((\tilde{\mathbf{v}}^{T^*})^\top \mathbf{u}') \ge 1 - 1/d^2$.

For the risk, write $\hat{y} = \big(p^{T^*}\tilde{\mathbf{x}} + (1 - p^{T^*})\tilde{\mathbf{z}}'\big)^\top (\tilde{\mathbf{w}}^{T^*})^{\odot 2}$ and $\mu_x(\mathbf{x}) = \mathbf{x}^\top(\mathbf{w}^*)^{\odot 2}$. Decompose (as in Lemma 24)

$$\hat{y} - \mu_x = \mathbf{x}^\top\big((\mathbf{w}^{T^*})^{\odot 2} - (\mathbf{w}^*)^{\odot 2}\big) + (1 - p^{T^*})\,g', \quad g' = (\mathbf{z}' - \mathbf{x})^\top (\mathbf{w}^{T^*})^{\odot 2}.$$

Conditioned on $\Omega$, we have $0 \le 1 - p^{T^*} \le 1/d^2$. Hence,

$$\mathbb{E}\big[(\hat{y} - \mu_x)^2 \mid \Omega\big] \le 2\,\lambda_{\max}(\mathbf{H})\,\big\|(\mathbf{w}^{T^*})^{\odot 2} - (\mathbf{w}^*)^{\odot 2}\big\|_2^2 + 2\,\tau^2\,\Gamma_\tau'\,\big\|(\mathbf{w}^{T^*})^{\odot 2}\big\|_2^2,$$

where $\tau = 1/d^2$ and $\Gamma_\tau' := \lambda_{\max}(\mathbb{E}[(\mathbf{z}' - \mathbf{x})(\mathbf{z}' - \mathbf{x})^\top \mid \Omega]) \le C_\Gamma' d$ with $C_\Gamma' = (B_\mathbf{x} + B_{\mathbf{z}'})^2$ by the envelope bounds. The cross term satisfies $|\mathbb{E}[(\hat{y} - \mu_x)\epsilon \mid \Omega]| \le B_\epsilon\,\tau\,\sqrt{\Gamma_\tau'}\,\|(\mathbf{w}^{T^*})^{\odot 2}\|_2$.

Finally, substitute the parameter accuracies from Theorem 1: $\|(\mathbf{w}^{T^*})^{\odot 2} - (\mathbf{w}^*)^{\odot 2}\|_2^2 \lesssim r\,\sigma^2 \log d/n$ and $\|(\mathbf{w}^{T^*})^{\odot 2}\|_2^2 \le r(1 + o(1))$, to conclude

$$\mathbb{E}\left[\left|\mathcal{L}_{(\mathbf{x},y,\mathbf{z}')}\big(\tilde{\mathbf{w}}^{T^*}, \tilde{\mathbf{v}}^{T^*}\big) - \frac{\sigma^2}{2}\right| \,\Big|\, \Omega\right] \lesssim \frac{r\,\sigma^2 \log d}{n}.$$

$\qquad\square$

A.5 EXAMPLES

**Example 25** (Under Dominant-Coordinate Condition, population linear regression retains a constant fraction of $\mathbf{z}$). *Let*

$$y = \mathbf{x}_1 + \epsilon, \qquad \mathbf{z}_1 = c\,y + \xi, \quad \mathbf{z}_j \equiv 0 \ (j \geq 2),$$

*with mutually independent coordinates and*

$$\mathrm{Var}(\mathbf{x}_1) = a > 0, \quad \mathrm{Var}(\mathbf{x}_j) = 1 \ (j \geq 2), \quad \mathrm{Var}(\epsilon) = \sigma^2, \quad \mathrm{Var}(\xi) = \sigma_\xi^2,$$

*and $\epsilon \perp (\mathbf{x}, \xi)$, $\xi \perp \mathbf{x}$. Consider the population linear regression problem*

$$(\beta_\mathbf{x}^\star, \beta_\mathbf{z}^\star) \in \arg\min_{\beta_\mathbf{x}, \beta_\mathbf{z}} \ \mathbb{E}\big[(y - \beta_\mathbf{x}^\top \mathbf{x} - \beta_\mathbf{z}^\top \mathbf{z})^2\big].$$

*If $\sigma_\xi^2 > 0$, the unique solution is*

$$\beta_{\mathbf{z},1}^\star = \frac{c\,\sigma^2}{\sigma_\xi^2 + c^2\sigma^2}, \quad \beta_{\mathbf{z},j}^\star = 0 \ (j \geq 2), \qquad \beta_{\mathbf{x},1}^\star = \frac{\sigma_\xi^2}{\sigma_\xi^2 + c^2\sigma^2}, \quad \beta_{\mathbf{x},j}^\star = 0 \ (j \geq 2).$$

*Equivalently,*

$$\hat{y} = \mathbf{x}_1 + \frac{c^2\sigma^2}{\sigma_\xi^2 + c^2\sigma^2}\,\epsilon + \frac{c\,\sigma^2}{\sigma_\xi^2 + c^2\sigma^2}\,\xi,$$

*so linear regression uses $\mathbf{z}_1$ to fit the residual noise $\epsilon$ with coefficient $c^2\sigma^2/(\sigma_\xi^2 + c^2\sigma^2)$, thereby retaining a* constant fraction *of the $\mathbf{z}$ component.*

*Proof of Example 25.* From $y = \mathbf{x}_1 + \epsilon$ and independence,

$$\mathrm{Var}(y) = a + \sigma^2, \quad \mathrm{Cov}(\mathbf{x}_1, y) = a, \quad \mathrm{Cov}(\mathbf{x}_j, y) = 0 \ (j \geq 2).$$

Moreover,

$$\mathbf{z}_1 = c\,y + \xi = c\,\mathbf{x}_1 + c\,\epsilon + \xi, \qquad \mathbf{z}_j \equiv 0 \ (j \geq 2),$$

so

$$\mathrm{Cov}(\mathbf{x}_1, \mathbf{z}_1) = c\,a, \quad \mathrm{Cov}(\mathbf{x}_j, \mathbf{z}_1) = 0 \ (j \geq 2), \quad \mathrm{Var}(\mathbf{z}_1) = c^2(a + \sigma^2) + \sigma_\xi^2.$$

The population projection is

$$\Pi_{y\mathbf{x}} = \Sigma_{\mathbf{xx}}^{-1}\Sigma_{\mathbf{x}y}.$$

Since $y = \mathbf{x}_1 + \epsilon$ with $\epsilon \perp \mathbf{x}$, we have

$$\Pi_{y\mathbf{x}} = (1, 0, \ldots, 0)^\top, \qquad y' := y - \Pi_{y\mathbf{x}}^\top \mathbf{x} = \epsilon.$$

Likewise, only $c\,\mathbf{x}_1$ in $\mathbf{z}_1 = c\,\mathbf{x}_1 + c\,\epsilon + \xi$ projects on $\mathbf{x}$, hence

$$\Pi_{\mathbf{z}_1\mathbf{x}} = \frac{\mathrm{Cov}(\mathbf{x}_1, \mathbf{z}_1)}{\mathrm{Var}(\mathbf{x}_1)} = \frac{c\,a}{a} = c, \qquad \mathbf{z}_1' := \mathbf{z}_1 - \Pi_{\mathbf{z}_1\mathbf{x}}\,\mathbf{x}_1 = c\,\epsilon + \xi,$$

and $\mathbf{z}_j' \equiv 0$ for $j \geq 2$.

Frisch-Waugh-Lovell Theorem (Frisch & Waugh, 1933) yields the $\mathbf{z}$-coefficients of the joint regression by regressing $y'$ on $\mathbf{z}'$. Thus,

$$\beta_{\mathbf{z},1}^\star = \frac{\mathrm{Cov}(\mathbf{z}_1', y')}{\mathrm{Var}(\mathbf{z}_1')} = \frac{\mathrm{Cov}(c\,\epsilon + \xi,\ \epsilon)}{\mathrm{Var}(c\,\epsilon + \xi)} = \frac{c\,\sigma^2}{c^2\sigma^2 + \sigma_\xi^2},$$

and $\mathbf{z}_j' \equiv 0$ implies $\beta_{\mathbf{z},j}^\star = 0$ for $j \geq 2$.

The $\mathbf{x}$-only regression gives

$$\beta_{\mathbf{x},1}^{(\text{only-}\mathbf{x})} = 1, \qquad \beta_{\mathbf{x},j}^{(\text{only-}\mathbf{x})} = 0 \ (j \geq 2).$$

The joint coefficient equals the above minus what is explained via $\mathbf{z}_1$:

$$\beta_{\mathbf{x},1}^\star = \beta_{\mathbf{x},1}^{(\text{only-}\mathbf{x})} - \frac{\mathrm{Cov}(\mathbf{x}_1, \mathbf{z}_1)}{\mathrm{Var}(\mathbf{x}_1)}\,\beta_{\mathbf{z},1}^\star = 1 - c\,\beta_{\mathbf{z},1}^\star = \frac{\sigma_\xi^2}{\sigma_\xi^2 + c^2\sigma^2},$$

and by symmetry $\beta_{\mathbf{x},j}^\star = 0$ for $j \geq 2$.

$\square$

**Example 26.** *Let* $x_1 \sim \text{Unif}\big([\underline{X}, \overline{X}] \cup [-\overline{X}, -\underline{X}]\big)$ *(symmetric,* $\underline{X} > 0$*),* $x_2 \sim \text{Unif}[-B, B]$*,* $\epsilon \sim \text{Unif}[-E, E]$*,* $\xi_1 \sim \text{Unif}[-\Xi_1, \Xi_1]$*,* $\xi_2 \sim \text{Unif}[-\Xi_2, \Xi_2]$*, all independent, and* $x_j \equiv 0$ *for* $j \geq 3$*. Let*

$$y = x_1 + x_2 + \epsilon, \qquad z_1 = (1 - \kappa)\, y + \xi_1, \qquad z_2 = \alpha_2\, y + \xi_2,$$

*with* $0 < \kappa < 1$*,* $\alpha_2 > 0$*. Set* $B_x := \max\{\overline{X}, B\}$*,* $B_x' := B$*, and,* $B_\epsilon := E$*. Then:*

(1) Descendant stronger than a non-dominant causal coordinate. *Since* $\text{Var}(x_2) = B^2/3$ *and* $\text{Var}(y) =: V_y = \text{Var}(x_1) + \text{Var}(x_2) + \text{Var}(\epsilon)$,

$$\text{Cov}(x_2, y) = \text{Var}(x_2) = B^2/3, \qquad \text{Cov}(z_2, y) = \alpha_2\, V_y.$$

*Hence for any* $\alpha_2 > (B^2/3)/V_y$ *we have* $\text{Cov}(z_2, y) > \text{Cov}(x_2, y)$ *even though* $x_2$ *is causal.*

(2) Population dominance of coordinate 1 (Condition 1). *Since*

$$s_j = \mathbb{E}\big[y(x_j + z_j)\big], \qquad \mu_j = \mathbb{E}\big[\epsilon(x_j + z_j)\big], \qquad s_j^{\text{eff}} = s_j + \mu_j.$$

*Then*

$$s_1^{\text{eff}} = \text{Var}(x_1) + (1 - \kappa)\text{Var}(y) + (1 - \kappa)\text{Var}(\epsilon), \quad s_2^{\text{eff}} = \text{Var}(x_2) + \alpha_2\text{Var}(y) + \alpha_2\text{Var}(\epsilon),$$

*and* $s_j^{\text{eff}} = 0$ *for* $j \geq 3$*. Because* $m_1, m_{1j}$ *are bounded under our assumptions and* $s_2^{\text{eff}}$ *is fixed by* $(B, \alpha_2, E, \kappa)$*, choosing* $\text{Var}(x_1)$ *large enough (e.g., increasing* $\overline{X}$*) guarantees*

$$s_1^{\text{eff}} > \frac{2m_1}{15} + \max_{j>1}\Big(4|s_j^{\text{eff}}| + \frac{m_{1j}}{8}\Big),$$

*i.e., Condition 1.*

(3) Per-sample margins (Condition 2). *Since* $x_1 - z_1 = \kappa x_1 - (1 - \kappa)(x_2 + \epsilon) - \xi_1$,

$$|x_1 - z_1| \geq \kappa|x_1| - (1 - \kappa)(|x_2| + |\epsilon|) - |\xi_1| \geq \kappa\underline{X} - (1 - \kappa)(B + E) - \Xi_1.$$

*Thus if* $\kappa\,\underline{X} \geq (1 - \kappa)(B + E) + \Xi_1 + \tau_1$*, then* $|x_1 - z_1| \geq \tau_1$ *a.s. Moreover, if* $\kappa\,\underline{X} > (1 - \kappa)(B + E) + \Xi_1$*, then the sign of* $x_1 - z_1$ *equals the sign of* $x_1$ *a.s. Finally, if* $\frac{3}{4}\,\underline{X} \geq (r - 1)\,B_x' + B_\epsilon + \tau_2$*, then* $\frac{3}{4}|x_1| \geq (r - 1)\,B_x' + B_\epsilon + \tau_2$ *a.s. Hence Condition 2 holds.*

