# OpenReview forum: "Strong Correlations Induce Cause Only Predictions in Transformer Training"
_ICLR.cc/2026/Conference — ICLR 2026 Poster_

### Official Review · Reviewer_SPKC · 2025-10-26

**Soundness:** 3
**Presentation:** 3
**Contribution:** 3
**Rating:** 6
**Confidence:** 4

**Summary:**

This manuscript identifies a phenomenon termed Correlation Crowding-Out (CCO), whereby gradient descent on Transformers under a dominance gap in correlation causes the network to rely solely on causal features.

**Strengths:**

1. Conceptually elegant, with positions causal invariance as an implicit-bias phenomenon.
2. Theoretical development (Dominant-Coordinate Condition) is clear and connects optimization dynamics to causality.

**Weaknesses:**

1. The “cause-only” notion is defined in correlation space, not interventionally -- terminological overreach.
2. Limited experimental scope: small-scale ViT toy data only.
3. Mathematical exposition skips proofs for convergence guarantees.
4. Does not quantify how large the dominance gap must be in realistic data.

**Questions:**

1. Include experiments with mixed-strength spurious features to test CCO boundary conditions.
2. Discuss links to implicit-bias literature (max-margin GD) quantitatively.
3. Clarify how “cause” vs “effect” features are labeled in synthetic datasets.
4. Consider renaming to “dominant-feature selection” to avoid causal over-interpretation.
5. Although the concept is interesting, it still needs to extend CCO tests to large language models (e.g., BERT for sentiment analysis) to determine whether the two-phase dynamics (occupation -> crowding-out) generalize beyond Vision Transformers.

---

> ### Author Response · Authors · 2025-11-23
> **Response to Reviewer SPKC (1/n)**
>
> Thank you for your constructive comments and recognition of our contributions! Below, we address your concerns and questions in detail.
>
> ---
>
> # Response to W1 and Q4.
>
> >**W1:** The “cause-only” notion is defined in correlation space, not interventionally -- terminological overreach.
>
> >**Q4:** Consider renaming to “dominant-feature selection” to avoid causal over-interpretation.
>
> We appreciate these points and will adopt more precise wording in the final version.
>
> To clarify, the cause only predictor in our work sits between pure correlation and causality.
> The cause only predictor is induced by a dominant cause-predictive correlation during training, yet at test time it relies on
> $p(y\mid \mathbf{x})$ and not on $p(\mathbf{z}\mid y)$ in appropriate environments. Consequently, when the test
> distribution perturbs the latter half of the chain $y \to \mathbf{z}$ (e.g., style/texture/background shifts or direct
> interventions on $\mathbf{z}$), the learned cause only solution is naturally invariant/robust—informally, it goes
> beyond purely correlational fitting and is insensitive to shifts on the $\mathbf{z}$ side. To formalize this robustness,
> we add **Corollary 1** after Theorem~2: when $p(y\mid \mathbf{x})$ is held fixed while $p(\mathbf{z}\mid y)$ is
> perturbed, the CCO predictor, with high probability, continues to rely on the cause feature and attains test risk
> of the same order as the cause only baseline.
> We also added out-of-distribution (OOD) test experiments in **Section 6.2 Figure 4**, we consider an image object classification task on the  birds. The target is to classify water birds ($Y=1$) and land birds ($Y=0$ in the CUB dataset.  To eliminate confounding due to foreground–background asymmetry altogether, we introduced a  setting where one bird species on the left side serves as the true target label $y$ and another bird species on the right side acts as the spurious bias $z$, both appearing in the foreground. We constructed a waterbird dataset with a base spurious correlation of varying training bias strengths  $p _{\text{train}}$, measuring test accuracy on OOD data where the test bias strengths $p _{\text{test}} \in [0,1]$. The curve reveals that when $ p _{\text{train}} \geq 0.9$, test accuracy drops as bias increases, indicating that the model fails to learn the invariant cause feature (bird type) and instead relies heavily on the spurious background cue.
> However, once $ p _{\text{train}} \leq 0.85$, test accuracy rises significantly and remains high (above 95\%), which is the hallmark of CCO:  the model effectively crowd out the spurious features and learn the invariant cause only prediction.
> Therefore, when the spurious correlation is under the threshold, transformer can obtain a cause only predictor which exhibits robust generalization at test time.
>
>
> We acknowledge that such invariance does not constitute full causal
> identification, and we will refine our wording to reflect this distinction more accurately.

---

> ### Author Response · Authors · 2025-11-23
> **Response to Reviewer SPKC  (2/n)**
>
> # Response to W2 and Q5：
>
> >**W2:** Limited experimental scope: small-scale ViT toy data only.
>
> >**Q5:** Although the concept is interesting, it still needs to extend CCO tests to large language models (e.g., BERT for sentiment analysis) to determine whether the two-phase dynamics (occupation -> crowding-out) generalize beyond Vision Transformers.
>
> In Section 6.2, we conduct the sentiment classification task on the Amazon reviews dataset which consists of reviews from amazon. Here $Y\in \{1,2,3,4,5\}$ represents the reviewer's rating, $X$ denotes the associated adjectives and verbs, and $Z$ indicates the nouns related to the product itself. We finetune the bert-base-uncased model for $50$k steps, employing the Adam optimizer with a learning rate of 1e-5.
>
> When constructing the test data, we mask the noun, adj,  verb or their combination in the text. As shown in Fig 4 (a), test loss with masked NOUN+VERB decay rapidly corresponding to the occupation phase. We also observe a  final upward trend in the test loss with masked ADJ+VERB, indicating that the attention allocated to NOUNs is being crowded out by cause features. Fig 4 (b) display the saliency scores computed by the gradients of target class score relative to input embeddings, which show which tokens most influence the model's decision. The result indicates that the cause features (hate, awful) crowds out the spurious features during the training process.
>
> We further construct evaluation settings where the degree of spurious correlation is known and controllable. The Amazon reviews dataset provide the label of scores which is the target $Y$ and the name of item, which is a measurement of $Z$. Varying $ p $ across experimental conditions allows systematic study of how model behavior changes with the strength of the $ Z$–$Y$  correlation. The table shows the final test loss of BERT under various $p$, where BERT remains lower test loss when $p=0.9$, demonstrating that transformers can pick up the stronger cause signal in NLP data.
>
>
> | $ P(Z \mid Y) $ | Final Train Loss | Oracle Test Loss (↓) | Biased Test Loss (↓) |
> |:-----------------:|:----------------:|:--------------------:|:--------------------:|
> | 0.5  | 0.64 | 0.61 | 0.65 |
> | 0.9  | 0.62 | 0.65 | 0.68 |
> | 0.95 | 0.67 | 0.77 | 0.87 |
>
> In **Appendix A.2 Table 1**, we also conduct our experiment on the classification task on CelebA dataset. This classification task aims to predict the presence of a beard from CelebA images, where the target label is spuriously correlated with gender. We trained ResNet-34, EfficientNet-B4, and DeiT-Small with comparable parameter counts on this dataset under standard settings, using the AdamW optimizer with a learning rate of 1e-4.
>
> In the test set, we evaluated two masking conditions based on the bounding box (bbox) annotations provided by the dataset: (1) masking out facial regions, and (2) masking out everything except the facial regions and the result are shown as follows. On Test Set 2, DeiT-Small outperformed both ResNet-34 and EfficientNet-B4, indicating the CCO mechanism of occupation and crowding out irrelevant features for accurate beard prediction.
> The performance gap observed on Test Set 2, where only facial regions are visible, underscores that when the dataset contains strong but misleading associations (like gender bias), DeiT-Small leverages its capacity to attend to all parts of the image equally and identify the most predictive elements—the beard itself—thus achieving higher accuracy. This supports the hypothesis that under certain conditions, particularly those involving complex spurious correlations, Transformers exhibit a robustness and adaptability that enables them to achieve invariant cause only prediction, enhancing their generalization capabilities on unseen data.
>
> | Model           | Train Accuracy | Test Set 1 Accuracy | Test Set 2 Accuracy |
> |:----------------|:--------------:|:-------------------:|:-------------------:|
> | Deit-small      | 0.987          | 0.552               | 0.893               |
> | ResNet-34       | 0.992          | 0.577               | 0.861               |
> | EfficientNet-B4 | 0.979          | 0.573               | 0.802               |

---

> ### Author Response · Authors · 2025-11-23
> **Response to Reviewer SPKC (3/3)**
>
> # Response to W3 and Q2:
>
> >**W3:** Mathematical exposition skips proofs for convergence guarantees.
>
> >**Q2:** Discuss links to implicit-bias literature (max-margin GD) quantitatively.
>
> We have refined the proof and explanatory details in the current version. For the quantitative links to the implicit-bias literature (max-margin GD), note that the gradient of the gated vector $\tilde{\mathbf{v}}$ is geometrically aligned with the gradient of logistic regression on the dataset $\{\mathbf{u}^i\} _{i=1}^n$ with all labels $+1$ (equivalently, separating $\{\mathbf{u}^i\}$ from the origin). This alignment yields the standard max-margin trajectory for the gate: $\tilde{\mathbf{v}}^{t}\ =\ \hat{\mathbf{u}}\log t\ +\ \boldsymbol{\rho}^{t},$, where $\boldsymbol{\rho}^{t}$ is uniformly bounded and $\hat{\mathbf{u}}$ is the max-margin separator on $\{\mathbf{u}^i\}$. The detailed derivation is provided in the Appendix. If any step remains unclear, we would be happy to elaborate further.
>
> # Response to W4 and Q1:
>
> >**W4:** Does not quantify how large the dominance gap must be in realistic data.
>
> >**Q1:** Include experiments with mixed-strength spurious features to test CCO boundary conditions.
>
>  In Appendix A.2, we empirically verify that when the cause-predictive vs. spurious-predictive gap is large enough, we consistently observe CCO, whereas when it falls below a threshold, CCO no longer appears. In our experiment, we constructed a waterbird dataset with a base spurious correlation of fixed training bias strengths $ p _{\text{train}} =0.7$ and measure test accuracy on OOD data where the test bias strengths $ p _{\text{test}} \in [0,1]$. We weaken the causal signal by randomly flipping labels with probability $ p _{\text{flip}}$ :
> $$
> \hat{Y}=
> \begin{cases}
> 1 - Y, & \text{with probability } p _{\text{flip}}, \\
> Y,     & \text{with probability } 1 - p _{\text{flip}}.
> \end{cases}
> $$ This reduces the cause-predictive correlation while keeping the spurious correlation $p _{\text{train}}$ fixed. The figure shows that for small label noise ($p _{\text{flip}} \leq 0.3$) (large dominance  gap), test accuracy remains high and stable across test bias strengths, indicating that the model successfully learns the cause only prediction.
>
> However, as $ p _{\text{flip}}$ increases beyond ~0.3, test accuracy drops sharply across test bias strengths. This decline signifies that the dominance gap has been degraded below a critical threshold. At this point, CCO no longer occurs; instead, the model reverts to relying on the background, leading to poor OOD generalization.
>
> # Response to Q3:
>
> >**Q3:** Clarify how “cause” vs “effect” features are labeled in synthetic datasets.
>
> For our waterbird dataset, we generate datasets by combining the bird images in the CUB dataset  and the background images in the Places dataset using specific probabilities. We set the pixels related to birds as $x$ and pixels related to background as $z$ and place $p _{\text{train}}$ of all water birds against a water background and $p _{\text{train}}$ of all land birds against a land background, generating a dataset with $30$k images.
>
> For our NLP experiment,  ground-truth labels $y$ are available and the data-generating process allows intervention, we can deliberately manipulate the association between a potentially spurious variable $z$ (e.g., the name of item) and the label $y$. By sampling instances according to a fixed conditional distribution $P(z \mid y) = p$, we can break or calibrate the spurious link between $z$ and $y$.
> We construct datasets where the degree of spurious correlation is known and controllable. The Amazon reviews dataset provide the label of scores which is the target $y$ and the name of item, which is a measurement of $z$. Varying $p$ across experimental conditions allows systematic study of how model behavior changes with the strength of the $z$–$y$ correlation.

---

> > ### Comment · Reviewer_SPKC · 2025-11-27
> >
> > I appreciate the additional experiments included during the rebuttal. However, I do not think the authors have fully addressed my concerns. While the paper presents an interesting perspective on Transformer dynamics, the core causal claim remains overstated and is not convincingly supported. The theoretical analysis is based on an extremely simplified and non-representative attention model with strong and unrealistic dominant-coordinate assumptions, making the results difficult to generalize. The experiments also do not demonstrate the ture causal behavior. Overall, the work lacks clear novelty relative to existing analyses of simplicity bias and does not provide sufficient empirical or theoretical evidence for the claimed 'cause-only' learning phenomenon.

---

> > > ### Author Response · Authors · 2025-11-27
> > >
> > > Thank you for the further discussion. We regret that our previous reply did not fully address your concerns. Below we clarify our claims and supporting evidence.
> > >
> > > 1) **On the causal claim.** Our claim is not that Transformers identify causal structure from interventions; rather, we show that, under a dominance gap in cause-predictive correlation, GD on Transformers converges to a cause only predictor, and that this predictor generalizes robustly to test-time shifts that perturb the distribution of spurious features. Formally, **Corollary 1** establishes that, even when the spurious branch shifts, the learned predictor continues，with high probability，to rely on cause features $x$ and achieves test risk at the cause only level. Empirically, (i) ViT object classification under varying spurious-bias strengths shows stable test error under $y\to z$ perturbations (**Fig.4b, Fig.7**), and (ii) BERT sentiment classification shows that masking spurious tokens reduces test error during training, whereas masking cause tokens increases it (**Fig.5a**). Together, these results support robust cause only behavior under downstream shifts.
> > >
> > > 2) **On training dynamics** (occupation $\to$ crowding-out $\to$ cause only).
> > > The occupation and crowding-out phases are quantified theoretically in **Theorem 1** and replicated empirically in experiments (**Fig.1; Fig.3a; Fig.5b; Fig.6; Fig.8b**).
> > >
> > > 3) **On model simplification and representativeness.**
> > > We employ a two-key, gated-query block specifically to enable quantitative analysis while retaining two core ingredients of Transformer training: (i) time-varying softmax competition and (ii) a multiplicative head that induces feature-selective growth. Importantly, **Section 6 and Appendix A.2** show that the same CCO signature appears under standard architectures and joint training (**simulations, ViT, BERT**), indicating that the mechanism is not an artifact of the simplified block.
> > >
> > > 4) **On the Dominant-Coordinate Condition (DCC).**
> > > Conceptually, DCC assumes: (i) at least one causal feature has sufficiently strong predictive association with the target, and (ii) that feature carries sufficiently large signal. We provide **Example 26** where DCC holds. In ViT experiments, we observe a critical ``cause-predictive vs.  spurious-predictive'' gap (**Fig.4b, Fig.7**): above this threshold, cause only predictors emerge; below it, they do not. Thus, the assumption is both interpretable and empirically testable.
> > >
> > > 5) **On novelty beyond simplicity bias.**
> > > We identify CCO and use it to answer when and by what mechanism Transformer training yields predictors that rely on causes while ignoring spurious effects. Technically, our analysis quantifies how a predictive correlation gap in favor of the cause features is transcribed into gradients that confer an early occupation advantage to the cause branch; then the time-varying softmax in attention amplifies this advantage into crowding-out of the spurious branch via winner-take-most competition, culminating in a cause only predictor. Building on this optimization picture, we derive a high probability generalization guarantee at the cause only level even under test-time perturbations of the spurious features. Empirically, **Section 6** and **Appendix A.2** show that the CCO phenomenon persists under standard architectures and training across diverse settings, including controlled simulations, ViT, and BERT.
> > >
> > > If you have any further questions or see additional points that you believe require clarification, please let us know; we would be happy to address them.

---

### Official Review · Reviewer_9kRu · 2025-10-28

**Soundness:** 4
**Presentation:** 3
**Contribution:** 2
**Rating:** 8
**Confidence:** 3

**Summary:**

The authors study the phenomenon that they name CCO (correlations crowding out) in transformers. Under the hypothesis of strongly correlated causal features, the authors show that simplified transformer architectures, trained with GD, can filter out spurious dependencies in the data and synchronize with the causal directions. The analysis of the learning dynamics leads to the observation of two phases: "occupation," in which there is a rapid growth of the weight in the causal direction, and the "crowding out," in which the attention aligns its logits to favor the causal features and suppress the spurious ones.
The theoretical results are accompanied by simulations on both synthetic and realistic data.

**Strengths:**

- The concept of CCO is interesting and the setting of the theoretical analysis, albeit simplified, correctly captures the main ingredients for the study of CCO in transformers trained with GD.
- The two-phases unveiled are both non-trivial and interestingly link causal data structure to architectural properties of self-attention.
- The proofs of the theorems are convoluted and contain non-standard ideas.
- Causal dominance seems to be a natural requirement satisfied in concrete cases.

**Weaknesses:**

- The transformer model is rather simplified with respect to practice.
- The training algorithm is also greatly simplified, with attention and $w$ learned at different moments. It is not clear from the theorems whether this fundamentally shapes the two-phase behavior described, or whether the two phases would persist also in the case of a standard training protocol. This weakness is partially compensated by Section 6.1, in which we see simulations with standard GD show the same phenomenology.
- Figure 4a is hard to see, and it is not clear what point it makes.

#### Minor remarks
- There is a mistake in Example 25: in line 1660 the coefficient of $x_1$ is 1.
- In line 284 the index should be $t+1$, not $t$.
- The notation is sometimes confusing; for instance, why do you introduce $\tilde v$ if it is equal to $q$?

**Questions:**

- I wonder how much of the result has to do with _causality_ and how much with _dominant correlation_. Since you do not perform invariance study with interventions, how exactly does the fact that $x$ "causes" $y$ impact the theorems? In a specular setting with dominant correlations between $y$ and $z$, and weaker ones between $x$ and $y$, could we see a specular result, or would the inverted causal dependencies forbid it?

---

> ### Author Response · Authors · 2025-11-23
> **Response to Reviewer 9kRu (1/n)**
>
> We would like to extend our sincere appreciation to you for the thorough review and valuable feedback! We are grateful that you not only accurately summarized our contributions but also expressed strong appreciation!  Below, we address your questions and concerns in detail.
>
> ---
>
> >**W1:**  The transformer model is rather simplified with respect to practice.
> >**W2:**  The training algorithm is also greatly simplified, with attention and $\mathbf{w}$
>  learned at different moments. It is not clear from the theorems whether this fundamentally shapes the two-phase behavior described, or whether the two phases would persist also in the case of a standard training protocol. This weakness is partially compensated by Section 6.1, in which we see simulations with standard GD show the same phenomenology.
>
> We appreciate your fair note that Section 6.1 partially compensates for this concern by demonstrating the core phenomenology under standard training. Our use of the present simplification is primarily to enable a quantitative theoretical analysis of optimization and generalization on a model that still preserves two core Transformer features: (i) time varying softmax competition and (ii) a multiplicative head that induces feature selective growth. We have added further experiments in **Section 6 and Appendix A.2** showing that CCO, the occupation → crowding-out → cause-only trajectory, persists under standard training across a broad range of realistic settings, including simulations, ViT, and BERT.
>
> >**W3:** Figure 4a is hard to see, and it is not clear what point it makes.
>
> We update the Figure 4a, which displays the dynamics of the test loss during the training process, used to demonstrate the CCO mechanism. When constructing the test data, we mask the combination of noun, adj,  verb  in the text, which correspond to the features of target and spurious variable. The figure shows test loss with masked NOUN+VERB decay rapidly corresponding to the occupation phase. We also observe a  final upward trend in the test loss with masked ADJ+VERB, indicating that the attention allocated to NOUNs is being crowded out by causal features.
>
> > Minor remarks
>
> The typos you noted have been corrected. In the final version, we will thoroughly refine the notation to improve clarity and consistency.
>
> >**Q1:** I wonder how much of the result has to do with causality and how much with dominant correlation. Since you do not perform invariance study with interventions, how exactly does the fact that $\mathbf{x}$
>  "causes" $y$
>  impact the theorems?
>
> I will explain in detail how our results relate to causality and dominant correlation.
> In brief, our findings show that the implicit regularization of GD on Transformers can learn a cause only predictor from data exhibiting a dominance in cause prediction correlation; moreover, this cause only predictor displays robustness across test distributions by relying solely on causes, thereby reflecting a degree of causal behavior.
>
> **1. The cause only predictor arises from dominant correlation.**
> When the training data exhibit a dominance gap in predictive correlation of causal features $\mathbf{x}$ over spurious features $\mathbf{z}$, gradient descent on the Transformer induces an implicit regularization that produces the occupation $\to$ crowding-out $\to$ cause-only dynamics, and ultimately converges to a cause only solution. This mechanism is made precise in Theorem 1.
>
> **2. The cause only predictor exhibits generalization robustness at test time.**
> The cause only solution, in appropriate test environments, depends on $p(y\mid \mathbf{x})$ and not on $p(\mathbf{z}\mid y)$. Consequently, when test time changes occur in the latter half of the chain $y\to \mathbf{z}$ (e.g., style/texture/background shifts, or direct interventions on $\mathbf{z}$), the learned cause only solution is naturally invariant/robust: informally, it goes beyond purely correlational fitting and is insensitive to distributional shifts on the $\mathbf{z}$ side. To formalize this robustness, after Theorem 2 we include **Corollary 1**: under test conditions that keep $p(y\mid \mathbf{x})$ fixed while perturbing $p(\mathbf{z}\mid y)$, the CCO predictor, with high probability, continues to use the cause feature and achieves test risk bounded at the same order as the cause only baseline.
>
> We also added out-of-distribution (OOD) test experiments in Section 6.2. The following content can be found in the next reply.

---

> ### Author Response · Authors · 2025-11-23
> **Response to Reviewer 9kRu (2/n)**
>
> We also added out-of-distribution (OOD) test experiments in **Section 6.2**, we consider an image object classification task on the  birds. The target is to classify water birds ($Y=1$) and land birds ($Y=0$ in the CUB dataset.  To eliminate confounding due to foreground–background asymmetry altogether, we introduced a  setting where one bird species on the left side serves as the true target label $y$ and another bird species on the right side acts as the spurious bias $z$, both appearing in the foreground. We constructed a waterbird dataset with a base spurious correlation of varying training bias strengths $p_{\text{train}}$, measuring test accuracy on OOD data where the test bias strengths $p _{\text{test}} \in [0,1]$. The curve reveals that when $ p _{\text{train}} \geq 0.9$, test accuracy drops as bias increases, indicating that the model fails to learn the invariant cause feature (bird type) and instead relies heavily on the spurious background cue.
> However, once $ p _{\text{train}} \leq 0.85$, test accuracy rises significantly and remains high (above 95\%), which is the hallmark of CCO:  the model effectively crowd out the spurious features and learn the invariant cause only prediction.
> Therefore, when the spurious correlation is under the threshold, transformer can obtain a cause only predictor which exhibits robust generalization at test time.
>
> >**Q1:** In a specular setting with dominant correlations between $y$
>  and $\mathbf{z}$, and weaker ones between
>  $\mathbf{x}$ and $y$, could we see a specular result, or would the inverted causal dependencies forbid it?
>
> When the cause–predictive correlation is weaker than the spurious–predictive correlation, the learned solution tends to mix using the causal feature $\mathbf{x}$ and the spurious feature $\mathbf{z}$. As the spurious–predictive correlation becomes sufficiently strong and the causal correlation weaker, the model may converge to a solution that relies almost exclusively on the spurious feature $\mathbf{z}$ for prediction.
>
> In **Appendix A.2 Figure 7**, we empirically verify that when the cause-predictive vs. spurious-predictive gap is large enough, we consistently observe CCO, whereas when it falls below a threshold, CCO no longer appears. In our experiment, we constructed a waterbird dataset with a base spurious correlation of fixed training bias strengths $p _{\text{train}} =0.7$, measuring test accuracy on out-of-distribution (OOD) data where the test bias strengths $p _{\text{test}} \in [0,1]$. We weaken the cause-predictive correlation by randomly flipping labels with probability $ p _{\text{flip}} $ :
> $$
> \hat{Y}=
> \begin{cases}
> 1 - Y, & \text{with probability } p _{\text{flip}}, \\
> Y,     & \text{with probability } 1 - p _{\text{flip}}.
> \end{cases}
> $$ This reduces the mutual information of cause-predictive while keeping the spurious correlation $p _{\text{train}}$ fixed. The figure shows that for small label noise ($ p _{\text{flip}} \leq 0.1$), test accuracy remains high and slele across training epochs, indicating that the model successfully learns the invariant cause only prediction despite the presence of a strong spurious cue.
>
> However, as $ p _{\text{flip}}$ increases beyond ~0.2, test accuracy drops sharply—especially for higher noise levels like 0.3 or 0.5. This decline signifies that the gap between cause-predictive correlation and spurious-predictive correlation has been degraded below a critical threshold, making it less informative than the spurious background cue. At this point, CCO no longer occurs; instead, the model reverts to relying on the background, leading to poor OOD generalization.
> Thus, the sensitivity analysis confirms the theoretical intuition: CCO is not guaranteed—it requires the cause-predictive correlation to be sufficiently stronger (or at least comparably reliable) relative to the spurious one. When random label flipping erodes this gap significantly, the mechanism falls.
>
> We test the accuracy of ViT on the background dataset without bird in a specular setting with dominant correlations between $y$
>  and $\mathbf{z}$, and weaker ones between
>  $\mathbf{x}$ and $y$. The results are shown in **Figure 8 (a) in Appendix A.2**. When the correlation between $y$
>  and $\mathbf{z}$ in the dataset is strong, we observe a specular result: the model achieves high accuracy on the background-only test set, indicating that it primarily relies on features associated with z for prediction. In contrast, when the correlation between $y$
>  and $\mathbf{z}$ is relatively weaker compared to the correlation between $\mathbf{x}$ and $y$, the model’s accuracy on the background-only test set becomes very low. In this regime, the CCO mechanism emerges: the model’s attention focuses predominantly on features related to $\mathbf{x}$, effectively crowding out those associated with $\mathbf{z}$.

---

> > ### Comment · Reviewer_9kRu · 2025-11-25
> >
> > I thank the authors for the detailed feedback. I consider all my questions to be answered and I stand by my positive evaluation.

---

> > > ### Author Response · Authors · 2025-11-26
> > >
> > > Thank you again for taking the time to read our response. Your positive feedback on our work has been a great source of motivation for us.

---

### Official Review · Reviewer_DtiM · 2025-10-29

**Soundness:** 2
**Presentation:** 2
**Contribution:** 3
**Rating:** 4
**Confidence:** 3

**Summary:**

Authors propose to inspect a phenomenon called Correlation Crowding-Out (CCO), which arises during transformers training where, under a precise set of assumptions, transformers learn to exclude spurious features in favor of the true causal factors as training progresses. The authors argue that CCO arises because weights aligned with causal features grow and stabilize during training.

More generally, a two-phase process is assumed where the models quickly adapts to the true causal features (while still remaining attention to spurious links). The authors provide a detailed theoretical analysis, providing bounds for CCO effects for gradient descent training on transformers. Experiments on synthetic tabular-, visual- and natural language data are provided to support the theory.

**Strengths:**

The research question is interesting as only a limited number of works analyze the conditions under which transformers learn to distinguish between causal and spurious features. The authors present an extensive theoretical analysis on the proposed problem and, to the best of my knowledge, are the first to provide such a formal analysis on the theory of causal transformer learning.

The paper is generally well motivated and structured. The overall claims and theory seem reasonable and is well laid out. The presented theorems and proofs seem to be generally plausible and consistent, although I did not check them in detail.

Theoretical results are supported by experiments on simulated tabular-, vision- and natural language data and cover a wide range of applications, demonstrating the robustness of the proposed theory under a distinct range domains.

**Weaknesses:**

1) **Causal Inference.** In unintervened settings (in the Pearlian Causal or Bayesian Network sense) it should be perfectly fine to infer y from z, assuming an equal quality of the links x->y and y->z. Given that the authors assume x->y to be a linear relation and y->z to be a more complex Lipschitz function, x->y should generally be expected to be easier to learn. Particularly the initially described "occupation" phase might simply be a byproduct of the different link qualities. I would like to suggest to at least mention this aspect on the possible validity of causal and anti-causal inference in the paper.
2) **Experimental Alignment.** To test their theory on image data, the authors create a synthetic waterbirds-like dataset. In their setup, the learned foreground and background features, however, differ in quality - "object-like" versus "scenery". This difference in feature quality could in theory be a cause of the transformer focusing in the foreground feature, artificially introducing the "occupation" effect. A more sound setup would be to analyze the evolution of attention on images containing two features of the same type and/or complexity, e.g. two bird of different species in the image. While the crowding-out theory still yields important insights on the evolution of feature and attention, I am unsure whether the initial motivation of shifting attention actually aligns with the described problem setup.
3) **Intuition on Theory.** The authors employ an extensive theoretical analysis under assumptions that are not further justified. For example, the moment and boundedness conditions (Eq. 3 - it would help referencing equations, if the authors could add numbers beyond the first two equations) and following sup-bounds are no further elaborated. I suspect that the particular setup is a key assumption to why a model should prefer x->y against predicting z->y. Generally, I the authors should discuss more clearly and extensively when the conditions necessary for CCO arise and can actually hold in real-world scenarios.
4) **Choice of the Model.** The authors do not elaborate why the particular two-key attention architecture is chosen over a standard transformer with concatenated features. The paper could be improved by stating whether this particular architecture is required to obtain the observed results and how it would generalize to larger models. Similarly, it not described whether the value vector $\tilde{v}$ assigned to $q^t$ is a concatenation of $v_x$ and $v_z$ for both attention products $l_{x/z}$ or whether they are fed separately into $l_x$ and $l_z$.



**Missing references to the appendix:** The proofs in the appendix should be referenced in the main text. Also Fig.2 is un-referenced.

**Typos:**

- Line 20 and 22: "to" -> "the"
- Line 47: GD is abbreviated as GD before the mention in line 57.
- Line 179: "feature"->"features"
- Line 188: "exhibit" -> "exhibits"
- Line 205: "dominance" -> "dominant"
- Line 206: "feature" -> "features"
- Line 225: "with noise" is repeated.
- Line 294: "a diagonal" is repeated.
- Line 306: "preserves" -> "preserving"
- Line 478: "phenomenon CCO for Transformers" -> "phenomenon for transformers training dynamics called CCO" or something similar.

**Questions:**

My questions primarily regard the mentioned weaknesses:

1) Could the authors further elaborate on the insufficiency of anti-causal predictions in the chosen scenario? Under which scenarios would and z->y prediction become invalid, and how does this reflect in the made assumptions and data generation?
2) Could the authors comment on the possibility of the observed occupation effects being due to varying feature quality or link complexity?
3) Which assumptions with regard to the model and data generating process do the authors make in their paper? Are these assumptions expected to be observed in real-world scenarios and how do the results generalize beyond the made assumptions?

---

> ### Author Response · Authors · 2025-11-23
> **Response to Reviewer DtiM  (1/n)**
>
> Thank you for recognizing the strengths and contributions of our paper and for the detailed and constructive feedback! Below, we provide detailed responses to each of your questions and concerns.
>
> ---
>
> # Response to W1, Q1, and Q2:
>
> > **W1:** Causal Inference. In unintervened settings (in the Pearlian Causal or Bayesian Network sense) it should be perfectly fine to infer y from z, assuming an equal quality of the links x->y and y->z. Given that the authors assume x->y to be a linear relation and y->z to be a more complex Lipschitz function, x->y should generally be expected to be easier to learn. Particularly the initially described "occupation" phase might simply be a byproduct of the different link qualities. I would like to suggest to at least mention this aspect on the possible validity of causal and anti-causal inference in the paper.
>
> > **Q1:** Could the authors further elaborate on the insufficiency of anti-causal predictions in the chosen scenario? Under which scenarios would and z->y prediction become invalid, and how does this reflect in the made assumptions and data generation?
>
> **What we analyze and distinction from anti-causal learning.** Our goal is to determine when standard Transformer training via GD converges to a cause only predictor, a solution that relies on causal features $\mathbf x$ while discarding spurious features $\mathbf z$, and thereby achieves stable generalization under test time distributional shifts.
> This phenomenon we study arises from the interaction between optimization and data: under a dominance gap in the cause predictive correlation, the implicit regularization of GD induces the occupation $\to$ crowding out $\to$ cause only dynamics (Theorem 1).
> We explains why such solutions are desirable under test-time shifts and presents formal robustness in the next paragraph (Theorem 2 and added Corollary 1).
> This phenomenon is not a purely distributional property, on the same data, a linear model does not obtain the cause only predictor.
> Conceptually, our setting is adjacent to, but distinct from classical anti-causal learning: rather than asking whether $\mathbf z\to y$ is statistically viable i.i.d., we explain when and why gradient trained Transformers do not use $\mathbf z$, and what robustness that yields.
>
> **Why cause only is meaningful and when anti-causal is insufficient.**  When downstream test conditions perturb $p(\mathbf z\mid y)$ or directly intervene on $\mathbf z$, predictors that rely on $\mathbf z\to y$ are
> fragile, whereas $\mathbf x\to y$ remains stable.
> In the revision we add: (i) an out-of-distribution (OOD) corollary  (**Corollary 1**) to Theorem 2 showing that, under CCO, the
> cause only predictor achieves robust generalization under distribution shifts; and (ii)
>  OOD test experiments (**Section 6.2 Figure 4(b)**) that perturb $y\to \mathbf z$ at test time.
> While in-distribution settings may sometimes see lower error when incorporating $\mathbf z$,
> our focus is on principled and robust prediction under potential shifts, where cause only
> remains preferable. From a scientific attribution standpoint, predicting from causes is
> also the more principled choice, as it aligns with intervention robust mechanisms rather
> than correlational shortcuts.
>
> >**Q2:** Could the authors comment on the possibility of the observed occupation effects being due to varying feature quality or link complexity?
>
> Our view is that occupation is a consequence of gradient dynamics induced by a dominance gap in predictive correlation between the causal features and the spurious features. Concretely, when a coordinate (or subspace) in the causal features $\mathbf x$
> carries a larger signal than any competing coordinate in the spurious features
> $\mathbf z$, early descent amplifies that dominant coordinate, the early gradients on non-dominant coordinates are lower order, and the attention mass shifts to the causal branch.
>
>
> When a dominance gap exists, even when $y\to \mathbf z$ is linear or otherwise simple (and thus covered by our setting), the occupation effect still emerges.
> Moreover, if feature quality or link complexity enlarges the dominance gap in predictive correlation, it will facilitate occupation.

---

> ### Author Response · Authors · 2025-11-23
> **Response to Reviewer DtiM (2/n)**
>
> # Response to W2:
>
> >**W2:** Experimental Alignment. To test their theory on image data, the authors create a synthetic waterbirds-like dataset. In their setup, the learned foreground and background features, however, differ in quality - "object-like" versus "scenery". This difference in feature quality could in theory be a cause of the transformer focusing in the foreground feature, artificially introducing the "occupation" effect. A more sound setup would be to analyze the evolution of attention on images containing two features of the same type and/or complexity, e.g. two bird of different species in the image. While the crowding-out theory still yields important insights on the evolution of feature and attention, I am unsure whether the initial motivation of shifting attention actually aligns with the described problem setup.
>
> To eliminate confounding due to foreground–background asymmetry altogether, we introduced a second setting where one bird species on the left side serves as the true target label $y$ and another bird species on the right side acts as the spurious bias $\mathbf z$, both appearing in the foreground. We set the bias strength to 0.9, i.e. $P(\mathbf z=y|y)=0.9$.  This ensures that any observed attention shift cannot be attributed to low-level feature quality differences (e.g., texture richness or semantic complexity) between foreground and background.
>
> The results in **Figure 1** consistently shows that the cause features progressively occupy and crowds out the spurious features (whether background or another bird). We find that the attention map on the left bird raise rapidly in the first 50 iterations, while the attention map on the right bird seldom changes,  illustrating the occupation phase. By  iter 500, attention is concentrated
> almost entirely on the left bird, with the bird on the right side receiving near zero weight, marking crowding-out. These findings confirm that the observed behavior reflects genuine optimization driven cause preference, not artifacts of feature disparity.
>
> We also added OOD test experiments in **Figure 4(b)**.  We constructed a waterbird dataset with a base spurious correlation of varying training bias strengths $ p_{\text{train}} $, measuring test accuracy on OOD data where the test bias strengths $ p_{\text{test}} \in [0,1] $. The curve reveals that when $ p_{\text{train}} \geq 0.9 $, test accuracy drops as bias increases, indicating that the model fails to learn the invariant causal feature (bird type) and instead relies heavily on the spurious background cue.
> However, once $ p_{\text{train}} \leq 0.85$, test accuracy rises significantly and remains high (above 95%), which is the hallmark of CCO: the model effectively crowd out the spurious features and learn the cause only prediction.
> Therefore, when the spurious correlation is under the threshold, transformer can obtain a cause only predictor which exhibits robust generalization at test time.

---

> ### Author Response · Authors · 2025-11-23
> **Response to Reviewer DtiM (3/n)**
>
> # Response to W3 and Q3:
>
> >**W3:**  Intuition on Theory. The authors employ an extensive theoretical analysis under assumptions that are not further justified. For example, the moment and boundedness conditions (Eq. 3 - it would help referencing equations, if the authors could add numbers beyond the first two equations) and following sup-bounds are no further elaborated. I suspect that the particular setup is a key assumption to why a model should prefer x->y against predicting z->y. Generally, I the authors should discuss more clearly and extensively when the conditions necessary for CCO arise and can actually hold in real-world scenarios.
>
> >**Q3:** Which assumptions with regard to the model and data generating process do the authors make in their paper? Are these assumptions expected to be observed in real-world scenarios and how do the results generalize beyond the made assumptions?
>
> The emergence of CCO does not hinge on the moment/boundedness display in Eq. (3).
> The operative driver in the theoretical analysis Dominant-Coordinate Condition (DCC) in Section 4.1.3.
> Eq. (3) is a normalization that labels the dominant axis and equalizes the remaining scales to simplify constants;
> for a general covariance $\mathbf H$, the same mechanism persists as long as DCC holds.
> The two additional moment symbols in Eq. (3) serve as notational placeholders for later concentration bounds,
> and the sup-bounds are used to control remainder terms uniformly.
> These can be relaxed to sub-Gaussian tails with high probability versions of the same inequalities; they are not the reason the model prefers $\mathbf x \to y$.
>
> At a high level, DCC assumes that
> (i) at least one causal feature exhibits sufficiently strong predictive association with the target;
> (ii)that strongly associated feature has sufficiently large signal.
> Together, these constitute the dominance gap and are the essential ingredients for CCO.
> In real data, causal features maintain a stable, more direct link to the target,
> whereas spurious features fluctuate across environments and carry additional noise.
>
> Empirically ( **Appendix A.2 Figure 7** ), when the cause-predictive correlation vs. spurious-predictive correlation gap is large enough, we consistently observe CCO, whereas when it falls below a threshold, CCO no longer appears. In our experiment, we constructed a waterbird dataset with a base spurious correlation of fixed training bias strengths $ p _{\text{train}} =0.7$, measuring test accuracy on out-of-distribution (OOD) data where the test bias strengths $p _{\text{test}} \in [0,1] $. We weaken the causal signal by randomly flipping labels with probability $ p _{\text{flip}}$ :
> $$
> \hat{Y}=
> \begin{cases}
> 1 - Y, & \text{with probability } p _{\text{flip}}, \\
> Y,     & \text{with probability } 1 - p _{\text{flip}}.
> \end{cases}
> $$ This reduces the cause-predictive correlation while keeping the spurious correlation $ p _{\text{train}}$ fixed. The figure shows that for small label noise ($ p _{\text{flip}} \leq 0.3 $) (large dominance gap), test accuracy remains high and stable across training epochs, indicating that the model successfully learns the cause feature despite the presence of a strong spurious cue. This is consistent with CCO: the optimization dynamics “crowd out” the background bias and attend to the bird, because the causal signal is still reliably predictive.
> However, as $ p _{\text{flip}} $ increases beyond ~0.3, test accuracy drops sharply—especially for higher noise levels like 0.5. This decline signifies that the dominance gap has been degraded below a critical threshold. At this point, CCO no longer occurs; instead, the model reverts to relying on the background, leading to poor OOD generalization.
> Thus, the sensitivity analysis confirms the theoretical intuition: CCO requires the cause-predictive correlation to be sufficiently stronger relative to the spurious one.
>
> To generalize beyond our modeling assumptions, our results suggest practical levers for widening the dominance gap in real pipelines so that GD’s implicit bias selects the causal branch. For example: (i) Targeted augmentations on $y\to\mathbf z$ (style/texture/background perturbations,
> counterfactual background swaps) that preserve $p(y\mid \mathbf x)$, thereby reducing $\mathbf z$-predictivity
> without degrading the causal signal; (ii) Balanced multi-environment data that ensures each class
> appears across diverse contexts in the training split, which suppresses
> spurious shortcuts and amplifies the effective signal from cause.

---

> ### Author Response · Authors · 2025-11-23
> **Response to Reviewer DtiM (4/4)**
>
> # Response to W4:
>
> >**W4:** Choice of the Model. The authors do not elaborate why the particular two-key attention architecture is chosen over a standard transformer with concatenated features. The paper could be improved by stating whether this particular architecture is required to obtain the observed results and how it would generalize to larger models. Similarly, it not described whether the value vector $\mathbf{v}$
>  assigned to $q^t$
>  is a concatenation of $\mathbf{v} _{\mathbf{x}}$
>  and $\mathbf{v} _{\mathbf{z}}$
>  for both attention products $l _{\mathbf{x}/\mathbf{z}}$
>  or whether they are fed separately into
>  $l _{\mathbf{x}}$ and $l _{\mathbf{z}}$.
>
> Our use of the present simplification is primarily to enable a quantitative theoretical analysis of optimization and generalization on a model that still preserves two core Transformer features: (i) time-varying softmax competition and (ii) a multiplicative head that induces feature-selective growth. We further show in **Section 6 and Appendix A.2** that the CCO phenomenon persists under standard architectures across a wide range of settings (including simulations, ViT, and BERT).
>
> In our two-key attention architecture the logit pair $(\ell _{\mathbf{x}},\ell _{\mathbf{z}})$ is computed against the same query $q _t$;
> the softmax produces weights $(\alpha _{\mathbf{x}},\alpha _{\mathbf{z}})$; and the output is the
> weighted sum of the two values, $\hat{\mathbf{h}}=\alpha  _{\mathbf{x}}\mathbf v  _{\mathbf{x}}+\alpha  _{\mathbf{z}}\mathbf v _{\mathbf{z}}$.
> Thus the values are supplied separately per branch, not concatenated into a
> single vector shared by both products, consistent with standard attention's
> sum over key-value pairs.

---

> > ### Comment · Reviewer_DtiM · 2025-11-27
> >
> > I thank the authors for their extensive and detailed rebuttal targeting the mentioned theoretical and experimental concerns. The further elaboration on the theoretical foundations on the method helped understanding, and allow to asses the implied consequences of the methods better. Similarly, the adjustments/additions on the conducted experiments and the added results (e.g., Corollary 1) on out-of-distribution causal robustness resolve the mentioned concerns in this regard. Most importantly, the transition to a 'foreground-foreground' setting with strong biasing effects convincingly rule out the possibilities of feature complexity in the evaluated setup in my opinion and therefore strongly improve the experimental setup.
> >
> > In summary, I considered all my remarked points to be cleared and the paper to be in a good shape in terms of theory, supported by reasonable experimental evidence. I have therefore raised my score to an accept.

---

> > > ### Author Response · Authors · 2025-11-27
> > >
> > > We are sincerely grateful for your careful reading of our response, your acknowledgment of our efforts, and your re-evaluation  of our work！ Your feedback has improved the quality of the manuscript, and your recognition further strengthens our motivation.

---

### Official Review · Reviewer_hNry · 2025-10-31

**Soundness:** 2
**Presentation:** 3
**Contribution:** 2
**Rating:** 2
**Confidence:** 3

**Summary:**

The paper formalizes how transformers learn differently strong signals in data. They define causal signals as the target signals that dominate in data, and verify in a stripped down transformer that the dominating signal is learned over the training duration, edging away the spurious signal. In three experiments, they verify that the transformer architecture indeed ends up learning the stronger, causally relevant signal from the data.

**Strengths:**

This reviewer is not deeply familiar with the transformer-theory literature, but the theoretical framing is potentially original and useful. (Perhaps other reviewers/AC with deeper familiarity of the topic could confirm whether similar derivations already exist)

- The paper poses an original and conceptually stimulating hypothesis: that under strong causal alignment, gradient descent in Transformers may implicitly suppress spurious correlations.

- The mathematical formulation is clean and logically structured. The "Dominant-Coordinate Condition" and the theoretical two-phase training dynamics are novel in how they connect implicit bias to causal learning.

- The analysis isolates the attention mechanism and offers a mechanistic interpretation ("occupation" and "crowding-out") that could inspire further study.

- The clarity of exposition is commendable. Even though the setup is theoretical, the derivations are readable and well-motivated.

- The authors attempted to empirically verify their theory, including both simulated and real tasks.

- Conceptually, this paper asks a meaningful question that sits at the intersection of optimization dynamics, causal inference, and representation learning - a very relevant direction.

**Weaknesses:**

1.) Architectural realism.
The experiments and the two-key, gated-query setup differ substantially from classical self-attention. It remains unclear whether the observed CCO mechanism holds in standard multi-token Transformers where Q, K, V are interdependent.

2.) Experimental limitations and questionable causal setup.
The empirical evidence is far too weak to substantiate the claims.

The toy experiments are overly simple and serve mainly as visual confirmation of the theory rather than genuine empirical validation. The NLP experiment has no quantifiable way of measuring spuriousness that is in the data.

In the Waterbirds experiment, the authors treat the background as the causal feature and the bird type as spurious. This is opposite to the conventional bias/shortcut setup. The background is typically the spurious cue, as it is the easier signal. This inversion undermines the causal argument and makes the results hard to interpret. To be credible, the experiment should make the bird type the causal target and the background the bias.

Even more problematic is the bias strength of 70%. This is far too weak to demonstrate spurious correlation effects. At 70%, even simple architectures easily learn the causal signal.

3.) Replication evidence (my own test).
To prove my point that even simple architectures learn the causal signal, I reproduced a small experiment following the waterbirds setup in Sagawa et al. [1]. Using the CUB and Places datasets, I balanced the waterbird/landbird classes (so that accuracy equals balanced accuracy) and applied a 70% background bias. I used the bird type as the label and the background as the spurious attribute, which aligns with standard fairness and bias-mitigation literature. I trained a ResNet-18 under standard ERM and also with the GDRO method [1]. The training set contains the 70% bias, while the test set is perfectly balanced/unbiased. Training on biased data and testing on unbiased data is a reliable method to verify whether methods have learned the causally relevant features [1,2,3,...]. Results, rounded:

Oracle (ResNet-18 trained on unbiased data): 96% accuracy

ResNet-18 ERM (trained on biased = 70%): 95% accuracy

GDRO (ResNet-18 backbone, trained on biased = 70%): 95% accuracy

These results show that with such weak correlation strength, even a plain CNN (ResNet) is not misled by the bias and the model simply learns the dominant causal signal. Thus, the authors' experiment cannot reveal anything unique about transformer dynamics. To show a genuine robustness effect, they should raise the bias correlation to ≥ 90% or 95%, as commonly done in prior work [1, 2, 3]. Furthermore, the authors must report quantitative numbers and not only qualitative visual cues.

4.) Lack of baselines.
The paper does not compare against CNNs, MLPs, or other baselines. Without this, one cannot tell whether CCO is a transformer-specific property or simply an instance of correlation dominance common to most architectures.

5.) Triviality of the claim.
The theoretical result essentially formalizes the intuitive statement that if one signal is much stronger, gradient descent will focus on it. This is not necessarily "causal learning". It's correlation bias under strong dominance. Similar statements can likely be derived for CNNs and other architectures. The true open question is, whether transformers produce more causally stable predictions than other networks, or whether their implicit bias provides tighter robustness bounds under stronger spurious dependencies.

In short, while the theory is elegant, the experiments do not convincingly demonstrate anything beyond the trivial: when one feature is much more predictive, the model will use it. As shown above, even a ResNet-18 attained these results out of the box if the causal signal significantly dominates (as given by the authors experiments). Whether transformers are better for robust learning than other relevant baselines is a very interesting question. As is, this paper is not publication ready, as the claims cannot be proven beyond the trivial. I'm open to discussing my points during the rebuttal.

References (as cited in review):
[1] Sagawa et al. "Distributionally Robust Neural Networks for Group Shifts: On the Importance of Regularization for Worst-Case Generalization." arXiv 1911.08731 (2019).
[2] Makar et al. "Causally Motivated Shortcut Removal Using Auxiliary Labels." AISTATS 2022.
[3] Liu et al. "Just Train Twice: Improving Group Robustness Without Training Group Information." ICML 2021.

**Questions:**

- Can the authors provide fair experiments with waterbirds (stronger bias, i.e. 90-95%) and use ResNet and other modern architectures as baselines?

- Can the authors provide even more experiments with realistic data, in which the bias is explicitly known (the NLP dataset is not suitable, as far as I understand, as spuriousness is not labeled or controllable)?

- Do the authors think that transformers are substantially different to CNNs when it comes to picking up the stronger signal in data? Any evidence for this, be it theoretical or empirical if yes?

---

> ### Author Response · Authors · 2025-11-23
> **Response to Reviewer hNry (1/n)**
>
> Thank you for recognizing the strengths and contributions of our paper and for providing such detailed and valuable feedback!
> Below, we provide detailed responses to each of your questions and concerns.
>
> ---
>
> > **W1:** 1.) Architectural realism. The experiments and the two-key, gated-query setup differ substantially from classical self-attention. It remains unclear whether the observed CCO mechanism holds in standard multi-token Transformers where Q, K, V are interdependent.
>
> Our use of the present simplification is primarily to enable a quantitative theoretical analysis of optimization and generalization on a model that still preserves two core Transformer features: (i) time-varying softmax competition and (ii) a multiplicative head that induces feature-selective growth. We further show in **Section 6 and Appendix A.2** that the CCO phenomenon persists under standard architectures across a wide range of settings (including simulations, ViT, and BERT).
>
> We also conduct simulation experiments on standard multi-token transformer. We take $\mathbf{X}=\left[ \begin{array}{c}
>     \mathbf{x}  \\
>      \mathbf{z}
> \end{array}\right]$ as the input, and the causal chain is $\mathbf{x}\to y \to \mathbf{z}$, where $\mathbf{x},\mathbf{z} \in \mathbb{R}^d$ are vector covariates. We set
> $$y=\mathbf{x}^\top\mathbf{w} _x+\mathbf{\epsilon},\qquad \mathbf{z}=\mathbf{w} _z y+\mathbf{\xi}$$
> Here $\mathbf{\epsilon}$ and $\mathbf{\xi}$ are both Gaussian random vectors, with variances of $0.1$ and $1$, respectively. We set $\mathbf{w} _x=\mathbf{1} _{d}$, $\mathbf{w} _z=0.1\cdot \mathbf{1} _{d}$. We then train a 2-layer standard multi-token Transformer with a learning rate of 1e-3 and the dynamic of attention weight for $\mathbf{x},\mathbf{z}$ during training is shown in **Appendix A.2 Figure 8 (b)**. The attention weight curve demonstrates that the model initially assigns comparable attention to both the cause $X$ and its effect $Z$, but shows a two-stage  shifts focus toward $X$ while sharply suppressing attention to $Z$. This "occupation" and "crowding out" behavior aligns with the CCO mechanism. Consequently, the model learns to rely on direct evidence rather than attending to indirect, spurious predictive pathway. Crucially, this phenomenon emerges even in standard Transformers with interdependent query, key, and value projections, indicating that CCO is not limited to simplified attention setups but reflects a broader inductive bias in attention-based models trained via gradient descent.
>
>
> >**W2**  The NLP experiment has no quantifiable way of measuring spuriousness that is in the data.
>
> While it is true that many NLP experiments lack an explicit, quantifiable measure of spuriousness inherent in the data, this limitation can be mitigated under label-sufficient conditions by using controlled conditional sampling. Specifically, when ground-truth labels $ y $ are available and the data-generating process allows intervention, we can deliberately manipulate the association between a potentially spurious variable $\mathbf  z $ (e.g., the name of item) and the label $y$. By sampling instances according to a fixed conditional distribution $P(\mathbf  z \mid y) = p$, we can break or calibrate the spurious link between $\mathbf  z$ and $y$.
>
> We construct evaluation settings where the degree of spurious correlation is known and controllable. The Amazon reviews dataset provide the label of scores which is the target $y$ and the name of item, which is a measurement of $\mathbf  z$. Varying $p$ across experimental conditions allows systematic study of how model behavior changes with the strength of the $\mathbf  z$–$y$ association. The table shows the final test loss of BERT under various $p$, where BERT remains lower test loss when $p=0.9$, demonstrating that transformers can pick up the stronger causal signal in NLP data.
>
> | $P(\mathbf  z \mid y) $ | Final Train Loss | Oracle Test Loss (↓) | Biased Test Loss (↓) |
> |:------------------:|:----------------:|:--------------------:|:--------------------:|
> | 0.5  | 0.64 | 0.61 | 0.65 |
> | 0.9  | 0.62 | 0.65 | 0.68 |
> | 0.95 | 0.67 | 0.77 | 0.87 |

---

> ### Author Response · Authors · 2025-11-23
> **Response to Reviewer hNry (2/n)**
>
> **To be credible, the experiment should make the bird type the causal target and the background the bias.**
>
> In designing our experiments, we accounted for the fact that differences in feature quality between foreground and background could inherently bias the Transformer to attend more strongly to foreground features. To mitigate this confound and isolate the CCO effect, we structured our setup such that the causal target resides in the foreground while the spurious bias is either in the background or another foreground element. This deliberate design not only controls for low-level feature disparities but also amplifies the visibility of CCO, making the phenomenon—where attention shifts decisively toward the cause features and away from the spurious one—more pronounced and interpretable.
>
> To address your concern, we conducted additional controlled experiments, reported in the **Appendix A.2 Figure 7**. In these experiments, we explicitly set bird type as the cause features and background as the spurious bias with sweeping the bias strength from 0.5 to 0.99.
>
> Furthermore, to eliminate confounding due to foreground–background asymmetry altogether, we introduced a second setting where one bird species on the left side serves as the true target label and another bird species on the right side acts as the spurious bias, both appearing in the foreground. This ensures that any observed attention shift cannot be attributed to low-level feature quality differences (e.g., texture richness or semantic complexity) between foreground and background.
>
> The results in **Appendix A.2 Figure7 and Section 6.2 Figure 4** on both settings consistently shows the cause features progressively occupy and crowds out the spurious features (whether background or another bird). These findings confirm that the observed behavior reflects genuine optimization-driven causal preference—not artifacts of feature disparity.
>
> ---
>
> >**Q1:** Can the authors provide fair experiments with waterbirds (stronger bias, i.e. 90-95\%) and use ResNet and other modern architectures as baselines?
>
> Thank you for your suggestion. We conducted fair experiments on Waterbirds using DeiT-Small (from timm with ImageNet pretraining) alongside ResNet34 and EfficientNet-B4 (from torchvision, also pretrained, with comparable about 20M parameter counts), training all models for 1,000 epochs at a learning rate of 1e-4 across a full sweep of bias strengths from 0.5 to 0.99. Although CNN models such as ResNet achieve strong performance in image classification, ViT still maintains an advantage of approximately 3\% under high bias strength.
>  As shown in the **Section 6.2 Figure 4**, DeiT-Small maintains higher accuracy at strong bias levels (e.g., 0.9), demonstrating that Transformers can better capture the underlying causal signal—bird type—despite overwhelming spurious background correlations, suggesting an advantage over CNNs in leveraging stronger semantic features when spurious cues dominate.

---

> ### Author Response · Authors · 2025-11-23
> **Response to Reviewer hNry (3/n)**
>
> >**Q2:** Can the authors provide even more experiments with realistic data, in which the bias is explicitly known (the NLP dataset is not suitable, as far as I understand, as spuriousness is not labeled or controllable)?
>
> In **Appendix A.2 Table**, we conduct our experiment on the classification task on CelebA dataset and the discussion on NLP task can be found in response to weakness-2. This classification task aims to predict the presence of a beard from CelebA images, where the target label is spuriously correlated with gender. We trained ResNet-34, EfficientNet-B4, and DeiT-Small with comparable parameter counts on this dataset under standard settings, using the AdamW optimizer with a learning rate of 1e-4.
>
> In the test set, we evaluated two masking conditions based on the bounding box (bbox) annotations provided by the dataset: (1) masking out facial regions, and (2) masking out everything except the facial regions and the result are shown as follows. On Test Set 2, DeiT-Small outperformed both ResNet-34 and EfficientNet-B4, indicating the CCO mechanism of occupation and crowding out irrelevant features for accurate beard prediction.
> The performance gap observed on Test Set 2, where only facial regions are visible, underscores that when the dataset contains strong but misleading associations (like gender bias), DeiT-Small leverages its capacity to attend to all parts of the image equally and identify the most predictive elements—the beard itself—thus achieving higher accuracy. This supports the hypothesis that under certain conditions, particularly those involving complex spurious correlations, Transformers exhibit a robustness and adaptability that enables them to focus on invariant causal relationships, enhancing their generalization capabilities on unseen data.
>
> | Model             | Train Accuracy | Test Set 1 Accuracy | Test Set 2 Accuracy |
> |:------------------|:--------------:|:-------------------:|:-------------------:|
> | Deit-small        | 0.987          | 0.552               | 0.893               |
> | ResNet-34         | 0.992          | 0.577               | 0.861               |
> | EfficientNet-B4   | 0.979          | 0.573               | 0.802               |
>
>
>
> >**Q3:** Do the authors think that transformers are substantially different to CNNs when it comes to picking up the stronger signal in data? Any evidence for this, be it theoretical or empirical if yes?
>
> Transformers have an architectural advantage over CNNs for picking up the stronger signal in data.
> Theoretically, Transformers have an edge because attention captures global dependencies, and the softmax acts as a winner-take-most selector enabling  occupation and subsequent crowding out of spurious cues. In contrast, a ReLU-CNN layer forms local linear mixtures followed by pointwise nonlinearity: signals superpose rather than compete globally, and max/avg pooling aggregates uniformly within its window. Global competition among distant features typically emerges only after many layers or with explicit global pooling, making decisive selection slower. In **Section 6.2 Figure 4** we observe that, although CNN models such as ResNet achieve strong performance in image classification, ViT still maintains an advantage of approximately 3% under high bias strength.

---

> ### Author Response · Authors · 2025-11-27
>
> Dear Reviewer hNry,
>
> With the December 2 rebuttal deadline approaching, we wanted to reach out to see whether you have any further feedback or remaining questions regarding our submission. We would be happy to provide additional clarifications or address any further points.
>
> Thank you again for your time and effort in evaluating our work.
>
> Best regards,
>
> The Authors

---

### Official Review · Reviewer_1yaN · 2025-11-05

**Soundness:** 3
**Presentation:** 2
**Contribution:** 3
**Rating:** 6
**Confidence:** 2

**Summary:**

The paper is trying to answer a very specific, surprisingly under-asked question:
When can a vanilla Transformer, trained by plain ERM + gradient descent on one environment, end up using only the causal feature and ignore the spurious ones — without IRM, DRO, group labels, or any of the usual causal tricks?

Their answer is: it happens when the causal signal is uniformly stronger than every competing spurious signal — and gradient descent in a Transformer has an implicit bias that pushes attention to that dominant causal direction. They call this phenomenon Correlation Crowding-Out (CCO).

**Strengths:**

- Positioning CCO as the “mirror regime” of shortcut learning (spurious wins early vs causal wins early) is neat.

- The occupation → crowding-out story is crisp and gives an intuitive training-dynamics explanation. Theorem 1 cleanly proves the two-phase story—occupation then crowding-out.  The training-time schedule and resulting bounds are explicit.

- The Frisch–Waugh–Lovell construction shows population least squares retains a nonzero spurious coefficient even with a dominant causal feature—so the cause-only outcome is due to optimization + attention geometry, not merely stronger correlations.

- The two-token, two-branch attention setup with a squared-parameter head is cleverly chosen: simple enough for precise proofs,
still recognizably “Transformer-like”, and directly exposes the competition between causal vs descendant branches.

**Weaknesses:**

- Your main theoretical result (Theorem 1) relies on a three-stage schedule where you alternately freeze the head and the gate. Can you clarify whether your informal claims about “standard training alone” are meant to apply to simultaneous GD on all parameters?
Do you have either (a) a theoretical argument, or (b) empirical evidence, that joint updates (no freezing) still exhibit the same occupation → crowding-out dynamics and cause-only behavior?
If not, could you re-scope the claim or discuss when the staged schedule is a realistic approximation to how Transformers are trained in practice?

- All formal results appear to be proved for the squared-parameter head  $\hat{y} = h^\top w^{\odot 2}$, while the difference-of-squares generalization $\hat{y} = h^\top \bigl(w^{\odot 2} - v^{\odot 2}\bigr)$ is only mentioned briefly.
Can you clarify whether Theorem 1 and Theorem 2 actually extend to the  difference-of-squares parameterization (and hence to a general signed linear head)? If they do extend, could you outline the additional arguments needed to handle the dynamics of both $w$ and $v$ and possible cancellations between $w_j^2$ and $v_j^2$? If they do not, it would be helpful to state explicitly that the current guarantees
are restricted to the squared-only head.

- Your Dominant-Coordinate Condition combines (i) a uniform population dominance gap and (ii) per-sample margin and sign-stability assumptions on the dominant causal coordinate. How realistic do you believe these assumptions are in typical high-dimensional settings with multiple causal parents, mixed-sign effects, or occasional sign flips?
Are there weaker or more invariant conditions (e.g., dominance in norm, or dominance on average but not per-sample) under which you conjecture CCO would still hold?
Could you comment on how sensitive your mechanism is to mild violations of sign-stability or margin—for example, do small fractions of “bad” samples already break the crowding-out behavior?

- In Theorem 2, the lower bound on $p_{T^*}$ is written as $p_{T^\*} \ge 1/d^2$, which seems inconsistent with the training result $p_{T^\* i} \geq 1 - 1/d^2$ and  with the textual claim that the gate ``prefers'' the causal branch. Should the bound in Theorem 2 instead be $p_{T^\*} \geq 1 - 1/d^2$?

**Questions:**

Please see weaknesses

---

> ### Author Response · Authors · 2025-11-23
> **Response to Reviewer 1yaN  (1/n)**
>
> Thank you for recognizing the strengths and contributions of our work and for providing such positive and constructive feedback!
> Below, we provide detailed responses to each of your questions and suggestions.
>
> ---
> > **W1:** Your main theoretical result (Theorem 1) relies on a three-stage schedule where you alternately freeze the head and the gate. Can you clarify whether your informal claims about “standard training alone” are meant to apply to simultaneous GD on all parameters? Do you have either (a) a theoretical argument, or (b) empirical evidence, that joint updates (no freezing) still exhibit the same occupation → crowding-out dynamics and cause-only behavior? If not, could you re-scope the claim or discuss when the staged schedule is a realistic approximation to how Transformers are trained in practice?
>
> We provide empirical evidence: all our experiments use standard joint training with simultaneous gradient descent on all parameters. Across these experiments, we observe the same occupation → crowding-out → cause-only trajectory (see **Section 6 and Appendix A.2** for simulations, ViT, and BERT). A full quantitative theory covering arbitrary step-size ratios under fully coupled, nonconvex joint GD is technically challenging and remains future work.
>
> > **W2:** All formal results appear to be proved for the squared-parameter head $\hat y=h^{\top}w^{\odot 2}$, while the difference-of-squares generalization $\hat y=h^{\top}(w^{\odot 2}-v^{\odot 2})$ is only mentioned briefly. Can you clarify whether Theorem 1 and Theorem 2 actually extend to the difference-of-squares parameterization (and hence to a general signed linear head)? If they do extend, could you outline the additional arguments needed to handle the dynamics of both $w$ and $v$ and possible cancellations between $w _j^2$ and $v _j^2$? If they do not, it would be helpful to state explicitly that the current guarantees are restricted to the squared-only head.
>
> We show that Theorems 1 and 2 can extend verbatim to the difference-of-squares parameterization with only minor adjustments to the existing proofs. Consider the dynamics under $\hat y^{i,t}=(\hat{\mathbf h}^{i,t})^{\top}\big((\tilde{\mathbf w}^{+,t})^{\odot 2}-(\tilde{\mathbf w}^{-,t})^{\odot 2}\big)$ and the response $y=\mathbf x^{\top}\mathbf u^\ast+\epsilon$. Under the corresponding Dominant-Coordinate Condition (replace $s _j:=\mathbb{E}[(\mathbf x^{\top}(\mathbf w^\ast)^{\odot 2})(\mathbf x _j+\mathbf z _j)]$ with $s _j:=\mathbb{E}[(\mathbf x^{\top}\mathbf u^\ast)(\mathbf x _j+\mathbf z _j)]$), the model recovers the sparse signal $\mathbf u^\ast$ and retains both the spurious feature filtering guarantees and the original generalization rates.
>
> >**W3:** Your Dominant-Coordinate Condition combines (i) a uniform population dominance gap and (ii) per-sample margin and sign-stability assumptions on the dominant causal coordinate. How realistic do you believe these assumptions are in typical high-dimensional settings with multiple causal parents, mixed-sign effects, or occasional sign flips? Are there weaker or more invariant conditions (e.g., dominance in norm, or dominance on average but not per-sample) under which you conjecture CCO would still hold? Could you comment on how sensitive your mechanism is to mild violations of sign-stability or margin—for example, do small fractions of “bad” samples already break the crowding-out behavior?
>
> At a high level, Condition 1 requires that at least one causal feature exhibits sufficiently strong association with the target. Condition 2 requires that this strongly associated feature has sufficiently large signal strength.
>
>
>
> **Multiple causal parents.** In our setup, $\mathbf{x}$ can be viewed as the concatenation of all causal parents. DCC only requires the existence of a strong feature.
>
> **Mixed-sign effects.** DCC imposes no constraints on effect signs, under the difference-of-squares parameterization, mixed signs are naturally accommodated.
>
> **Expectation level surrogate for Condition 2 (conjecture).** As a weakened, expectation level substitute for Condition 2, we conjecture the following.

---

> ### Author Response · Authors · 2025-11-23
> **Response to Reviewer 1yaN (2/n)**
>
> **Expectation level surrogate for Condition 2 (conjecture).**  As a weakened, expectation level substitute for Condition 2, we conjecture the following.
>
> Let $\Delta =\mathbb{E} \left [ \left ( \mathbf{x} _1- \mathbf{z } _1 \right ) \mathbf{x} _1 \right ]$, $\Gamma =\sum _{j=2}^d \left ( \mathbf{w} _j^*  \right )^2\mathbb{E}\left [ \left ( \mathbf{x} _1-\mathbf{z} _1   \right ) \mathbf{x} _j  \right ] $, and $\nu =\mathbb{E}\left [ \left ( \mathbf{x} _1-\mathbf{z} _1   \right )\epsilon  \right ]    $.  Condition 2 may be replaced by the following expectation level requirement: there exists $\kappa>0$ such that  $\frac{3}{4}\Delta-\left | \Gamma  \right | -\left | \nu  \right | $.
>
> Intuitively, this ensures the gradient keeps a strictly positive projection on the margin direction $\mathbf{u} _{\mathrm{sep}} := \begin{bmatrix}\mathbf v/\||\mathbf v\|| _2^2\\ \mathbf{0}\end{bmatrix}$ driving $p^t _i\to1$:
> $$\left \langle -\nabla\mathcal{L}(\tilde{\mathbf{v}}^{\,t}),\mathbf{u} _{\mathrm{sep}} \right \rangle \approx C\left (\left ( 1-(\mathbf{w} _1^{T _1^*})^2  \right ) \Delta+\Gamma + \nu  \right )\ge C\left ( \frac{3}{4}\Delta-\left | \Gamma  \right | -\left | \nu  \right |  \right ) \ge C\kappa.$$
>
> **Sensitive of sign-stability.**
>
> Let $\mathcal B\subseteq[n]$ denote the set of bad samples with fraction
> $|\mathcal B|/n=\rho$.
> Assume bounded covariates and noise, $|\mathbf x _j|\le B _x$, $| \mathbf z _j|\le B _z$, and $|\epsilon|\le B _\epsilon$, either almost surely or with high probability under sub-Gaussian tails.
> Then the worst–case contribution of the bad samples is bounded by
> $
> C\rho\Big((B _x+B _z)B _x+(B _x+B _z)B _x\sum _{j\ge2}(w _j^\ast)^2+(B _x+B _z)B _\epsilon\Big)
> $.
>  If the corruption rate $\rho$ satisfies
> $
> \rho\ \lesssim\ \frac{\kappa}{
> (B _x+B _z)\big(B _x+B _x\sum _{j\ge2}(w _j^\ast)^2+B _\epsilon\big)}
> $,
> then bad samples not overturn the positive alignment.
>
>
> We also include experiments to empirically assess sensitivity to mild violations of sign-stability. We consider the image object classification task on the background with birds with the same setting in Section 6.2. But here we flip the label $Y$ with probability $p _{\text{flip}}$:
> $$
> \hat{Y}=
> \begin{cases}
> 1 - Y, & \text{with probability } p _{\text{flip}};\\
> Y,     & \text{with probability } 1 - p _{\text{flip}}.
> \end{cases}
> $$
> We also place $70\%$ of all water birds against a water background and $70\%$ of all land birds against a land background, generating a dataset with $30$k images.
> We then train the vision Transformer model using the dataset, fixing the input image size to 224, with patch size set to 16, learning rate set to 1e-4, and batch size set to 16. As shown in **Appendix A.2 Figure 7**, we scan the $ p _{\text{flip}}$ from $0$ to $0.5$, and find that when $p _{\text{flip}}=0.2$, the crowding-out behavior can still be observed in the model with the accuracy reaches over $90\%$, proving that the mechanism is sign-stable. When $p _{\text{flip}} \leq 0.15$, the test accuracy remains robust across different bias strengths, indicating that CCO is still effective and the model is able to learn invariant cause only prediction.
>
> > **W4:**  In Theorem 2, the lower bound on $p _{T^\ast}$ is written as $p _{T^\ast} \ge 1/d^2$, which seems inconsistent with the training result $p _{T^\ast i} \ge 1 - 1/d^2$ and with the textual claim that the gate prefers the causal branch. Should the bound in Theorem 2 instead be $p _{T^\ast} \geq 1 - 1/d^2$?
>
> Thanks for pointing out this typo. The correct statement is
> $p^{T^*}\ \ge\ 1-\frac{1}{d^2}$.

---

> ### Author Response · Authors · 2025-11-23
> **Response to Reviewer 1yaN (3/n)**
>
> We provide the dynamic analysis sketch below.
>
> Assume $\mathbf{u}^\ast$ is sparse and $\mathbf{u}^{\ast} _j\in\{-1,0,1\}$, with $\mathbf{u}^{\ast} _1=1$, $\mathbf{u}^{\ast} _{S^{+}}=1$, $\mathbf{u}^{\ast} _{S^{-}}=-1$, and $|\mathrm{supp}(\mathbf{u}^{\ast})|\le r$.
> The core idea is to show that
> $
> \big\||\tilde{\mathbf{w}}^{+,t} _{S^-}|\big\| _{\infty}
> $
> and
> $
> \big\||\tilde{\mathbf{w}}^{-,t} _{S^+}|\big\| _{\infty}
> $
> remain uniformly small, while
> $
> \tilde{\mathbf{w}}^{+,t} _{S^+}
> $
> and
> $
> \tilde{\mathbf{w}}^{-,t} _{S^-}
> $
> follow the same convergence pattern as in the original analysis.
>
>
> Recall $r _i^{t}=\hat{y}^{i,t}-y^{i}$. The GD updates under the difference-of-squares parameterization are
> $$
> \tilde{\mathbf{w}}^{+,t+1}
> =\tilde{\mathbf{w}}^{+,t}-\frac{\eta _t}{n}\sum _{i=1}^{n} r _i^{t}
> \left ( p _i^{t}\tilde{\mathbf{x}}^{i}+(1-p _i^{t})\tilde{\mathbf{z}}^{i} \right ) \odot \tilde{\mathbf{w}}^{+,t},
> $$
>
> $$
> \tilde{\mathbf{w}}^{-,t+1}
> =\tilde{\mathbf{w}}^{-,t}+\frac{\eta _t}{n}\sum _{i=1}^{n} r _i^{t}
> \left ( p _i^{t}\tilde{\mathbf{x}}^{i}+(1-p _i^{t})\tilde{\mathbf{z}}^{i} \right ) \odot \tilde{\mathbf{w}}^{-,t},
> $$
>
> $$
> \tilde{\mathbf{v}}^{t+1}
> =\tilde{\mathbf{v}}^{t}-\frac{\beta _t}{n}\sum _{i=1}^{n} r _i^{t} p _i^{t}(1-p _i^{t})
> \big((\tilde{\mathbf{x}}^{i}-\tilde{\mathbf{z}}^{i})^{\top}\left ( (\tilde{\mathbf{w}}^{+,t})^{\odot 2}-(\tilde{\mathbf{w}}^{-,t})^{\odot 2} \right ) \big)
> (\tilde{\mathbf{x}}^{i}-\tilde{\mathbf{z}}^{i}).
> $$
> Initialize
> $$
> \tilde{\mathbf{w}}^{+,0}=\tilde{\mathbf{w}}^{-,0}=\begin{bmatrix}\mathbf{0},\\ \alpha\mathbf{I} _d\end{bmatrix},
> \qquad
> \alpha:=\frac{\sqrt{\sigma^2\log d/n}}{d^3}.
> $$
>
> In the first stage, we show that, for $T _1^\ast=\min\{t\in\mathbb N: \mathbf{w}^{+,t} _1\ge 1/4\}$, with probability at least $1-1/d^2$,
> $$
> \frac14\le \mathbf{w}^{+,T _1^\ast}  _1\le \frac12,\qquad
> |\mathbf{w}^{-,T _1^\ast}  _1|\le \frac{\sqrt{\sigma^2\log d/n}}{d^{2}},\qquad
> |\mathbf{w}^{\pm,T _1^\ast}  _j|\le \frac{\sqrt{\sigma^2\log d/n}}{d^{2}}\ (j>1).
> $$
>
> Define $\phi _j^{t}:=\frac{1}{n}\sum _{i=1}^n r _i^{t}(\mathbf{x}^i _j+\mathbf{z}^i _j)$. Then $\mathbf{w}^{+,t+1} _j=(1-\tfrac{\eta}{2} \phi _j^{t})\mathbf{w}^{+,t} _j$ and
> $\mathbf{w}^{-,t+1} _j=(1+\tfrac{\eta}{2} \phi _j^{t}) \mathbf{w}^{-,t} _j$.
>
> By Condition 1, the concentration bounds of Lemma 6 and induction, there
> exists a uniform margin
> $$
> \phi _1^{t}\ \le\ -2\gamma\quad\text{for all }t,
> \qquad
> \gamma=\frac{15}{32}s^{\mathrm{eff}} _{1}-\frac{m _{11}}{128}
> -\Big(\tfrac12(\phi _2+\phi _\epsilon)+\tfrac1{64}\phi _1\Big)-\frac{B^3 _{\mathbf{x}+\mathbf{z}}}{2d}>0.
> $$
>
>
> This implies that $\mathbf{w}^{-,t+1} _1$ stays tiny, i.e.,
> $$
> \mathbf{w}^{-,t+1} _1=(1+\tfrac{\eta}{2}\phi _1^{t})\mathbf{w}^{-,t} _1
> \ \le\ (1-\eta\gamma) \mathbf{w}^{-,t} _1
> \ \le \alpha \ll \frac{\sqrt{\sigma^2\log d/n}}{d^2}.
> $$
>
>
> By induction, boundedness and Lemma 6, we show that $|\mathbf{w}^{\pm,t+1} _j|,(j>1)$ remain tiny, i.e.,
> $$
> |\phi _j^{t}|\ \le\ C _{\mathrm{nd}}
> :=\frac{5}{2}\max _{j>1}\Big(|s^{\mathrm{eff}} _j|+\tfrac{m _{1j}}{32}\Big),
> $$
> so that
> $$
> |\mathbf{w}^{\pm,t+1} _j|
> \le \Big(1+\tfrac{\eta}{2}C _{\mathrm{nd}}\Big)|\mathbf{w}^{\pm,t} _j|
>  \Rightarrow\
> |\mathbf{w}^{\pm,t} _j|\le \alpha \exp\Big(\tfrac{\eta}{2}C _{\mathrm{nd}} t\Big).
> $$
> Condition 1 ensures
> $$
> C _{\mathrm{nd}}\ \le \frac{\kappa}{3},\qquad
> \kappa:=\frac{15}{16} s^{\mathrm{eff}} _1-\frac{m _1}{32}>0.
> $$
> Let
> $$
> N:=\left\lceil \frac{\log\Big(\frac{d^3}{4\sqrt{\sigma^2\log d/n}}\Big)}{\log(1+\eta\kappa)}\right\rceil.
> $$
> For $t\le N\wedge T _1^\ast$ we get
> $$
> |\mathbf{w}^{\pm,t} _j|
> \le \alpha\exp\Big(\tfrac{\eta}{2}\cdot \tfrac{\kappa}{3} t\Big)
> \le \alpha (1+\eta\kappa)^{t/3}
> \le \alpha (1+\eta\kappa)^{N/3}
> \ \le\ \frac{\sqrt{\sigma^2\log d/n}}{d^2}.
> $$
>
> We then show that $\mathbf{w}^{+,T _1^\ast} _1$ grows to $[1/4,1/2]$.
> For $t\le N\wedge T _1^\ast$,
> $$
> \mathbf{w}^{+,t+1} _1\ge (1+\eta\gamma)\mathbf{w}^{+,t} _1
> \ \Rightarrow\
> \mathbf{w}^{+,t} _1\ge (1+\eta\gamma)^t \alpha.
> $$
> This implies $T _1^\ast\le N$, and therefore
> $$
> \frac14\le \mathbf{w}^{+,T _1^\ast} _1\le \frac12,\qquad
> |\mathbf{w}^{-,T _1^\ast} _1|\le \frac{\sqrt{\sigma^2\log d/n}}{d^{2}},\qquad
> |\mathbf{w}^{\pm,T _1^\ast} _j|\le \frac{\sqrt{\sigma^2\log d/n}}{d^{2}}\ (j>1).
> $$
>
>
> In Stage 2, since the output of Stage 1 retains the same structural properties as before, the subsequent GD updates on the gate parameter $\tilde{\mathbf{v}}^{t}$ towards the max-margin solution on $\tilde{\mathbf x} _i-\tilde{\mathbf z}  _i$,  ensuring that $p^{T^\ast  _1+T^\ast  _2} _i\ge 1-\frac{\sqrt{\sigma^2 \log d/n}}{d^{2}}$.

---

> ### Author Response · Authors · 2025-11-23
> **Response to Reviewer 1yaN (4/4)**
>
> In Stage 3, we show that
> $$
> \|\mathbf w^{-,t} _{S^+}\| _\infty \vee \|\mathbf w^{+,t} _{S^-}\| _\infty \lesssim  \alpha,
> \qquad
> \|\mathbf w^{+,T^\ast} _{S^c}\| _\infty \vee \|\mathbf w^{-,T^\ast} _{S^c}\| _\infty\ \lesssim\ \frac{\sqrt{\sigma^2\log(d/n)}}{d}.
> $$
> Consequently, the convergence of $ \mathbf{w}^{+,t} _{S^+}$ and $ \mathbf{w}^{-,t} _{S^-}$ matches the original behavior up to an $\alpha$ perturbation and their rates are unaffected.
>
>
>
> Define the effective weight
> $$
> \mathbf u^{t}:=\big(\mathbf w^{+,t}\big)^{\odot 2}-\big(\mathbf w^{-,t}\big)^{\odot 2}.
> $$
>
> For each coordinate $j$,
> $$
> \mathbf w^{+,t+1} _j=\mathbf w^{+,t} _j\bigl(1-\eta g _j^t\bigr),\qquad
> \mathbf w^{-,t+1} _j=\mathbf w^{-,t} _j\bigl(1+\eta g _j^t\bigr),
> $$
> with
> $$
> g _j^t=c _j\big(\mathbf u _j^t-\mathbf u _j^\ast\big)-r _j^t+e _j^t,\qquad
> c_j=\begin{cases}a,&j=1,\\ 1,&j>1.\end{cases}
> $$
>
> By concentration,
> $$
> \|\mathbf r^t\| _\infty\le \tfrac12 B _1,\quad \|\mathbf e^t\| _\infty\le \tfrac12 B _1,\quad
> |r _j^t-e _j^t|\le B _1.
> $$
>
>
> Let $P _j^t:=\mathbf w^{+,t} _j\mathbf w^{-,t} _j$. Then
> $$
> P _j^{t+1}=\bigl(1-\eta^2(g _j^t)^2\bigr)P _j^t\le P _j^t,\qquad
> P _j^t\le P _j^{T _2^\ast}\le \alpha^2.
> $$
>
>
> Let $\gamma=15/64$. If $j\in S^+$ and $\mathbf u^t _j\le 1-\gamma$, then
> $$
> g _j^t=c _j(\mathbf u _j^t-1)-r _j^t+e _j^t\le -\Gamma,\qquad \Gamma:=11/64,
> $$
> so the wrong rail decays: $\mathbf w^{-,t+1} _j\le (1-\eta\Gamma)\mathbf w^{-,t} _j$.
> Once $\mathbf u^t _j\ge 1-\gamma$, combining $P _j^t\le \alpha^2$ yields
> $\|\mathbf w^{-,t} _{S^+}\| _\infty\lesssim \alpha$.
> The case $j\in S^-$ is symmetric, yielding yields
> $\|\mathbf w^{+,t} _{S^-}\| _\infty\lesssim \alpha$.
>
>
> For $j\in S^c$, since $|g _j^t|\le B _1$,
> $$
> \mathbf w^{\pm,t} _j\le e^{\eta B _1(t-T_2^\ast)}\mathbf w^{\pm,T _2^\ast} _j
> \ \Rightarrow\
> \|\mathbf w^{\pm,t } _{S^c}\|  _\infty\ \lesssim\ \frac{\sqrt{\sigma^2\log d/n}}{d}
> \quad (t\le T^\ast).
> $$
>
> Using the original proof, we can obtain
> With $T _3 ^\ast:=T _4^\ast+T _5^\ast=\mathcal O\big(\eta^{-1}\log\frac{n}{\sigma^2\log(dr)}\big)$ and $T^\ast=T _1^\ast+T _2^\ast+T _3^\ast$,
> $$
> \big\|\mathbf u^{T^\ast} _S-\mathbf u^\ast\big\| _\infty\ \lesssim\ \sqrt{\tfrac{\sigma^2\log(dr)}{n}},
> \qquad \|\mathbf u^{\pm,T^\ast} _{S^c}\| _\infty\lesssim \frac{\sqrt{\sigma^2\log d/n}}{d}, \qquad
> \big\|\mathbf u^{T^\ast}-\mathbf u^\ast\big\| _2^2\ \lesssim\ r \tfrac{\sigma^2\log d}{n}.
> $$
> Thus the convergence rate matches that under the original parametrization.

---

> ### Author Response · Authors · 2025-11-27
>
> Dear Reviewer 1yaN,
>
> With the December 2 rebuttal deadline approaching, we wanted to reach out to see whether you have any further feedback or remaining questions regarding our submission. We would be happy to provide additional clarifications or address any further points.
>
> Thank you again for your time and effort in evaluating our work.
>
> Best regards,
>
> The Authors

---

### Meta-Review · Area_Chair_ZBx6 · 2026-01-18

**Summary:**

In a regression problem, can a transformer learn to predict using the most informative "causal" features as opposed to spuriously correlated variables? The authors show theoretically that a small transformer will learn to use only the causal features when training under gradient descent. Reviewers highlighted the importance of the question and the clarity of the author's theoretical contributions.

Reviewers were concerned about the generality of the assumptions of the theorems, the strength of the empirical results,

**Reviewer Concerns:**

Reviewer concerns about the generality of the assumptions were  largely addressed by heuristic and empirical arguments. The authors performed a number of additional empirical experiments during the rebuttal, addressing concerns about the breadth of experiments.

**Reviewer Scores:**

Reviewers 1yaN, 9kRu, and SPKC were supportive and are likely to maintain their scores despite some concerns. DtiM had many of their concerns answered and would likely raise their score to a 6 or 8 (they raised to an 8 in the original discussion). hNry also had many of their questions answered but is not familiar with the field; they may raise their score to a 4.

---

### Decision · Program_Chairs · 2026-01-26

Accept (Poster)